# Efficient Integrators for Diffusion Generative Models

**Kushagra Pandey** *
Department of Computer Science
University of California, Irvine
pandeyk1@uci.edu

**Maja Rudolph**
Bosch Center for Artificial Intelligence
Maja.Rudolph@us.bosch.com

**Stephan Mandt**
Department of Computer Science
University of California, Irvine
mandt@uci.edu

## ABSTRACT

Diffusion models suffer from slow sample generation at inference time. Therefore, developing a principled framework for fast deterministic/stochastic sampling for a broader class of diffusion models is a promising direction. We propose two complementary frameworks for accelerating sample generation in pre-trained models: *Conjugate Integrators* and *Splitting Integrators*. Conjugate integrators generalize DDIM, mapping the reverse diffusion dynamics to a more amenable space for sampling. In contrast, splitting-based integrators, commonly used in molecular dynamics, reduce the numerical simulation error by cleverly alternating between numerical updates involving different partitions of the drift (and diffusion) components. After extensively studying these methods empirically and theoretically, we present a hybrid method that leads to the best-reported performance for diffusion models in augmented spaces. Applied to Phase Space Langevin Diffusion [Pandey & Mandt, 2023] on CIFAR-10, our *deterministic* and *stochastic* samplers achieve FID scores of 2.11 and 2.36 in only 100 network function evaluations (NFE) as compared to 2.57 and 2.63 for the best-performing baselines, respectively. Our code and model checkpoints will be made publicly available at https://github.com/mandt-lab/PSLD.

## 1 INTRODUCTION

Score-based Generative models (or Diffusion models) (Sohl-Dickstein et al., 2015; Song & Ermon, 2019; Ho et al., 2020; Song et al., 2020) have demonstrated impressive performance on various tasks, such as image and video synthesis (Dhariwal & Nichol, 2021; Ho et al., 2022a; Rombach et al., 2022; Ramesh et al., 2022; Saharia et al., 2022a; Yang et al., 2022; Ho et al., 2022b; Harvey et al., 2022), image super-resolution (Saharia et al., 2022b), and audio and speech synthesis (Chen et al., 2021; Lam et al., 2021).

However, high-quality sample generation in standard diffusion models requires hundreds to thousands of expensive score function evaluations. While there have been recent advances in improving the sampling efficiency (Song et al., 2021; Lu et al., 2022; Zhang & Chen, 2023), most of these efforts have been focused towards a specific family of models that perform diffusion in the data space (Song et al., 2020; Karras et al., 2022). Interestingly, recent work (Dockhorn et al., 2022b; Pandey & Mandt, 2023; Singhal et al., 2023) indicates that performing diffusion in a joint space, where the data space is *augmented* with auxiliary variables, can improve sample quality and likelihood over data-space-only diffusion models. However, with a few exceptions focusing on specific score parameterizations (Zhang et al., 2022), improving the sampling efficiency for augmented diffusion models is still underexplored but a promising avenue for further improvements.

---

*Work partially done during an internship at Bosch Center for Artificial Intelligence

|  | | | | NFE (FID@50k $\downarrow$) | |
|---|---|---|---|---|---|
|  | Method | Description | Diffusion | 50 | 100 |
| Deterministic | (Ours) CSPS-D | Conjugate Splitting-based PSLD Sampler (CSPS) | PSLD | 3.21 | 2.11 |
| | (Ours) CSPS-D (+Pre.) | CSPS-D + Score Network preconditioning | PSLD | 2.65 | 2.24 |
| | DDIM (Song et al., 2021) | Denoising Diffusion Implicit Model | DDPM | 4.67 | 4.16 |
| | DEIS (Zhang & Chen, 2023) | Exponential Integrator with polynomial extrapolation | VP | 2.59 | 2.57 |
| | DPM-Solver-3 (Lu et al., 2022) | Exponential Integrator (order=3) | VP | 2.59 | 2.59 |
| | PNDM (Liu et al., 2022) | Solver for differential equations on manifolds | DDPM | 3.68 | 3.53 |
| | EDM* (Karras et al., 2022) | Heun's method applied to re-scaled diffusion ODE | VP | 3.08 | 3.06 |
| | gDDIM* (Zhang et al., 2022) | Generalized form of DDIM ($q = 2$) | CLD | 3.31 | - |
| | A-DDIM (Bao et al., 2022) | Analytic variance estimation in reverse diffusion | DDPM | 4.04 | 3.55 |
| Stochastic | (Ours) SPS-S | Splitting-based PSLD Sampler (SDE) | PSLD | 2.76 | 2.36 |
| | (Ours) SPS-S (+Pre.) | SPS-D + Score Network Preconditioning | PSLD | 2.74 | 2.47 |
| | SA-Solver (Xue et al., 2023) | Stochastic Adams Solver applied to reverse SDEs | VE | 2.92 | 2.63 |
| | SEEDS-2 (Gonzalez et al., 2023) | Exponential Integrators for SDEs (order=2) | DDPM | 11.10 | 3.19 |
| | EDM (Karras et al., 2022) | Custom stochastic sampler with churn | VP | 3.19 | 2.71 |
| | A-DDPM (Bao et al., 2022) | Analytic variance estimation in reverse diffusion | DDPM | 5.50 | 4.45 |
| | SSCS (Dockhorn et al., 2022b) | Symmetric Splitting CLD Sampler | PSLD | 18.83 | 4.83 |
| | EM (Kloeden & Platen, 1992) | Euler Maruyama SDE sampler | PSLD | 30.81 | 7.83 |

Table 1: Our proposed samplers perform comparably or outperform prior methods for CIFAR-10. Diffusion: (VP,VE) (Song et al., 2020), CLD (Dockhorn et al., 2022b), DDPM (Ho et al., 2020), PSLD (Pandey & Mandt, 2023). To ensure fair comparison, methods indicated with * were evaluated without incorporating additional training tricks. (Extended Results: Fig. 5)

**Problem Statement: Efficient Sampling during Inference.** Our goal is to develop efficient deterministic and stochastic integration schemes that are applicable to sampling from a broader class of diffusion models (for instance, where the data space is augmented with auxiliary variables) and achieve high-fidelity samples, even when the NFE budget is greatly reduced, e.g., from 1000 to 100 or even 50. We evaluate the effectiveness of the proposed samplers in the context of the Phase Space Langevin Diffusion (PSLD) (PSLD) (Pandey & Mandt, 2023) due to its strong empirical performance. However, the presented techniques also apply to other diffusion models, some of which are special cases of PSLD (e.g. Dockhorn et al. (2022b)). We make the following contributions,

- **Conjugate Deterministic Integrators.** These numerical integrators leverage invertible transformations to map the reverse process' deterministic dynamics to a space more suitable for fast sampling. We show that several existing deterministic sampling frameworks like DDIM (Song et al., 2021) and exponential integrators (Lu et al., 2022; Zhang & Chen, 2023; Zhang et al., 2022) are special cases of our framework, allowing us to generalize these methods to generic diffusion models in a principled manner. Moreover, we analyze the proposed framework from the lens of stability analysis and provide a theoretical justification for its effectiveness.

- **Reduced Splitting Integrators.** Taking inspiration from molecular dynamics (Leimkuhler, 2015), we present *Splitting Integrators* for efficient sampling in diffusion models. However, we show that their naive application can be sub-optimal for sampling efficiency. Therefore, based on local error analysis for numerical solvers (Hairer et al., 1993), we present several *improvements* to our naive schemes to achieve improved sample efficiency. We denote the resulting samplers as *Reduced Splitting Integrators*.

- **Conjugate Splitting Integrators.** We combine conjugate integrators with reduced splitting integrators for improved sampling efficiency and denote the resulting samplers as *Conjugate Splitting Integrators*. Our proposed samplers significantly improve sampling efficiency in PSLD. For instance, our best deterministic sampler achieves FID scores of 2.65 and 2.11, while our best stochastic sampler achieves FID scores of 2.74 and 2.36 in 50 and 100 NFEs, respectively, for CIFAR-10 (Krizhevsky, 2009) (See Table 1 for comparisons).

## 2 BACKGROUND

As follows, we provide relevant background on diffusion models and their augmented versions. Diffusion models assume that a continuous-time *forward process* (usually with an affine drift).

$$d\mathbf{z}_t = \boldsymbol{F}_t \mathbf{z}_t \, dt + \boldsymbol{G}_t \, d\mathbf{w}_t, \quad t \in [0, T], \tag{1}$$

with a standard Wiener process $\mathbf{w}_t$, time-dependent matrix $\boldsymbol{F} \colon [0, T] \to \mathbb{R}^{d \times d}$, and diffusion coefficient $\boldsymbol{G} \colon [0, T] \to \mathbb{R}^{d \times d}$, converts data $\mathbf{z}_0 \in \mathbb{R}^d$ into noise. A *reverse* SDE specifies how data is generated from noise (Anderson, 1982; Song et al., 2020),

$$d\mathbf{z}_t = \left[ \boldsymbol{F}_t \mathbf{z}_t - \boldsymbol{G}_t \boldsymbol{G}_t^\top \nabla_{\mathbf{x}_t} \log p_t(\mathbf{z}_t) \right] dt + \boldsymbol{G}_t d\bar{\mathbf{w}}_t, \tag{2}$$

which involves the *score* $\nabla_{\mathbf{z}_t} \log p_t(\mathbf{z}_t)$ of the marginal distribution over $\mathbf{z}_t$ at time $t$. Alternatively, data can be generated from the *Probability-Flow* ODE (Song et al., 2020),

$$d\mathbf{z}_t = \left[ \boldsymbol{F}_t \mathbf{z}_t - \frac{1}{2} \boldsymbol{G}_t \boldsymbol{G}_t^\top \nabla_{\mathbf{z}_t} \log p_t(\mathbf{z}_t) \right] dt. \tag{3}$$

The score is intractable to compute and is approximated using a parametric estimator $\boldsymbol{s}_\theta(\mathbf{z}_t, t)$, trained using denoising score matching (Vincent, 2011; Song & Ermon, 2019; Song et al., 2020). Once the score has been learned, generating new data samples involves sampling noise from the stationary distribution of Eqn. 1 (typically an isotropic Gaussian) and numerically integrating Eqn. 2, resulting in a stochastic sampler, or Eqn. 3 resulting in a deterministic sampler. While most work on efficient sample generation in diffusion models has focused on a limited class of non-augmented diffusion models (Song et al., 2020; Karras et al., 2022), our work is also applicable to a broader class of diffusion models. These two classes of diffusion models are presented next.

**Non-Augmented Diffusions.** Many existing diffusion models are formulated purely in data space, i.e., $\mathbf{z}_t = \mathbf{x}_t \in \mathbb{R}^d$. One popular example is the *Variance Preserving* (VP)-SDE (Song et al., 2020) with $\boldsymbol{F}_t = -\frac{1}{2} \beta_t \boldsymbol{I}_d, \boldsymbol{G}_t = \sqrt{\beta_t} \boldsymbol{I}_d$. Recently, Karras et al. (2022) instead propose a re-scaled process, with $\boldsymbol{F}_t = \boldsymbol{0}_d, \boldsymbol{G}_t = \sqrt{2\dot{\sigma}_t \sigma_t} \boldsymbol{I}_d$, which allows for faster sampling during generation. Here $\beta_t, \sigma_t \in \mathbb{R}$ define the noise schedule in their respective diffusion processes.

**Augmented Diffusions.** For augmented diffusions, the data (or position) space, $\mathbf{x}_t$, is coupled with *auxiliary* (a.k.a momentum) variables, $\mathbf{m}_t$, and diffusion is performed in the joint space. For instance, Pandey & Mandt (2023) propose PSLD, where $\mathbf{z}_t = [\mathbf{x}_t, \mathbf{m}_t]^T \in \mathbb{R}^{2d}$. Moreover,

$$\boldsymbol{F}_t = \left( \frac{\beta}{2} \begin{pmatrix} -\Gamma & M^{-1} \\ -1 & -\nu \end{pmatrix} \otimes \boldsymbol{I}_d \right), \qquad \boldsymbol{G}_t = \left( \begin{pmatrix} \sqrt{\Gamma\beta} & 0 \\ 0 & \sqrt{M\nu\beta} \end{pmatrix} \otimes \boldsymbol{I}_d \right), \tag{4}$$

where $\{\beta, \Gamma, \nu, M^{-1}\} \in \mathbb{R}$ are the SDE hyperparameters. Augmented diffusions have been shown to exhibit better sample quality with a faster generation process (Dockhorn et al., 2022b; Pandey & Mandt, 2023), and better likelihood estimation (Singhal et al., 2023) over their non-augmented counterparts. In this work, we focus on sample quality and, therefore, study the efficient samplers we develop in the PSLD setting.

## 3 Designing efficient Samplers for Generative Diffusions

We present two complementary frameworks for efficient diffusion sampling. We start by discussing *Conjugate Integrators*, a generic framework that maps reverse diffusion dynamics into a more suitable space for efficient deterministic sampling. Next, we discuss *Splitting Integrators*, which alternate between numerical updates for separate components to simulate the reverse diffusion dynamics. Lastly, we unify the benefits of both frameworks and discuss *Conjugate Splitting Integrators*, which enable the generation of high-quality samples, even with a low NFE budget.

### 3.1 Conjugate Integrators for efficient deterministic Sampling

Given a dynamical system (e.g., the ODE in Eqn. 3), the primary intuition behind conjugate integrators is to use invertible transformations to project the current state at time $t$ into another space which is more amenable for numerical integration. The transformation is chosen such that integration can be performed with a relatively larger step size and therefore reaches a solution faster. The resulting dynamics in the projected space can then be inverted to obtain the final solution in the original space. We first define conjugate integrators before deriving a mapping that allows us to use them in diffusion model sampling.

**Definition 3.1** (Conjugate Integrators). Given an ODE: $d\mathbf{z}_t = \boldsymbol{f}(\boldsymbol{z}_t, t)\,dt$, let $\mathcal{G}_h : \mathbf{z}_t \to \mathbf{z}_{t+h}$ denote a numerical integrator map for this ODE with step-size $h > 0$. Furthermore, given a continuous-invertible mapping $\phi : [0, T] \times \mathbb{R}^d \to \mathbb{R}^d$ such that $\hat{\mathbf{z}}_t = \phi_t(\mathbf{z}_t)$, let $\mathcal{H}_h : \hat{\mathbf{z}}_t \to \hat{\mathbf{z}}_{t+h}$ denote a

numerical integrator map for the *transformed* ODE in the projected space. Then the maps $\mathcal{G}_h$ and $\mathcal{H}_h$ are **conjugate** under $\phi$ if,

$$\mathcal{G}_h = \phi_{t+h}^{-1} \circ \mathcal{H}_h \circ \phi_t.$$

We provide an intuitive illustration of conjugate integrators in Fig. 1. Consequently, the iterated maps $\mathcal{G}_h^n$ and $\mathcal{H}_h^n$ (where $n$ denotes the number of iterations) are also conjugate under $\phi$. Next, we design conjugate integrators for efficient deterministic sampling from diffusion models.

**Conjugate Integrators for Diffusion ODEs.** We develop conjugate integrators for solving the probability flow ODE defined in Eqn. 3. In practice, we approximate the actual score by its parametric approximation $s_{\boldsymbol{\theta}}(\mathbf{z}_t, t)$. Following prior work (Karras et al., 2022; Salimans & Ho, 2022; Dockhorn et al., 2022b), we assume the following score network parameterization:

$$s_{\boldsymbol{\theta}}(\mathbf{z}_t, t) = \boldsymbol{C}_{\text{skip}}(t)\mathbf{z}_t + \boldsymbol{C}_{\text{out}}(t)\boldsymbol{\epsilon}_{\boldsymbol{\theta}}(\boldsymbol{C}_{\text{in}}(t)\mathbf{z}_t, C_{\text{noise}}(t)). \quad (5)$$

We restrict the mapping $\phi_t$ in this work to invertible affine transformations such that $\hat{\mathbf{z}}_t = \boldsymbol{A}_t\mathbf{z}_t$. To derive the probability flow ODE in the projected space, we reparameterize $\boldsymbol{A}_t$ in terms of another mapping $\boldsymbol{B} : [0, T] \to \mathbb{R}^d$ and introduce $\boldsymbol{\Phi}_t$ for notational convenience as follows,

Figure 1: Conjugate Integrators (Def. 3.1)

$$\boldsymbol{A}_t = \exp\left(\int_0^t \boldsymbol{B}_s - \boldsymbol{F}_s + \frac{1}{2}\boldsymbol{G}_s\boldsymbol{G}_s^\top\boldsymbol{C}_{\text{skip}}(s)ds\right), \qquad \boldsymbol{\Phi}_t = -\int_0^t \frac{1}{2}\boldsymbol{A}_s\boldsymbol{G}_s\boldsymbol{G}_s^\top\boldsymbol{C}_{\text{out}}(s)ds, \quad (6)$$

where $\exp(.)$ denotes the matrix-exponential, and $\boldsymbol{F}_t$ and $\boldsymbol{G}_t$ are the drift and diffusion coefficients of the underlying forward process (Eqn. 1). The probability flow ODE in the projected space $\hat{\mathbf{z}}_t = \boldsymbol{A}_t\mathbf{z}_t$ can be written in terms of these quantities.

**Theorem 1.** *Let $\mathbf{z}_t$ evolve according to the probability-flow ODE in Eqn. 3 with the score function parameterization given in Eqn. 5. For any mapping $B : [0, T] \times \mathbb{R}^d \to \mathbb{R}^d$ and $\boldsymbol{A}_t$, $\boldsymbol{\Phi}_t$ given by Eqn. 6, the probability flow ODE in the projected space $\hat{\mathbf{z}}_t = \boldsymbol{A}_t\mathbf{z}_t$ is given by*

$$d\hat{\mathbf{z}}_t = \boldsymbol{A}_t\boldsymbol{B}_t\boldsymbol{A}_t^{-1}\hat{\mathbf{z}}_t dt + d\boldsymbol{\Phi}_t\boldsymbol{\epsilon}_{\boldsymbol{\theta}}\left(\boldsymbol{C}_{in}(t)\boldsymbol{A}_t^{-1}\hat{\mathbf{z}}_t, C_{noise}(t)\right). \quad (7)$$

We present the proof in Appendix B.1. Applying an Euler update to the transformed ODE in Eqn. 7 with a step-size $h > 0$ yields the update rule for our proposed conjugate integrator:

$$\hat{\mathbf{z}}_{t-h} = \hat{\mathbf{z}}_t - h\boldsymbol{A}_t\boldsymbol{B}_t\boldsymbol{A}_t^{-1}\hat{\mathbf{z}}_t + (\boldsymbol{\Phi}_{t-h} - \boldsymbol{\Phi}_t)\boldsymbol{\epsilon}_{\boldsymbol{\theta}}\left(\boldsymbol{C}_{\text{in}}(t)\boldsymbol{A}_t^{-1}\hat{\mathbf{z}}_t, C_{\text{noise}}(t)\right). \quad (8)$$

For a given timestep schedule $\{t_i\}$ and a user-specified matrix $\boldsymbol{B}_t$, we present a complete algorithm for the proposed conjugate integrator and some practical considerations, such as computing the coefficients in Eqn. 6 and the invertibility of $\boldsymbol{A}_t$ in Appendix B.6.

Intuitively, projecting the probability-flow ODE dynamics into a different space introduces the matrix $\boldsymbol{B}_t$ as an additional degree of freedom that can be tuned during inference to improve sampling efficiency. In the rest of this section, we demonstrate how certain choices of $\boldsymbol{B}_t$ connect to previous work and how $\boldsymbol{B}_t$ can be chosen to further improve upon prior work.

**Choice of $\boldsymbol{B}_t$ and connections with other integrators.** There has been a lot of recent work in accelerating diffusion models using ODE-based methods like DDIM (Song et al., 2021) and exponential integrators (Zhang & Chen, 2023; Zhang et al., 2022; Lu et al., 2022). We find several theoretical connections between the proposed conjugate integrator in Eqn. 8 and existing deterministic samplers. More specifically, the following theoretical results hold for the choice of $\boldsymbol{B}_t = \boldsymbol{0}$.

**Proposition 1.** *For the VP-SDE (Song et al., 2020), the transformed ODE in Eqn. 7 is equivalent to the DDIM ODE proposed in Song et al. (2021)* (See Appendix B.2 for a proof).

**Proposition 2.** *For the diffusion model formulation considered in Lu et al. (2022), the exponential integrator proposed in DPM-Solver (Lu et al., 2022) is analogous to the numerical integrator in Eqn. 8 (See Appendix B.3 for a proof). More generally, for the forward process in Eqn. 1, the conjugate integrator update in Eqn. 8 is equivalent to applying the exponential integrator proposed in Zhang & Chen (2023) in the original space $\mathbf{z}_t$ (See Appendix B.4 for a proof).*

These theoretical connections allow us to extend methods like DDIM and exponential integrators to novel diffusion models in a principled manner. For an empirical evaluation, we implement DDIM

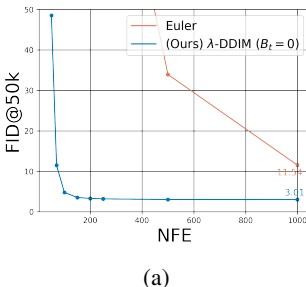 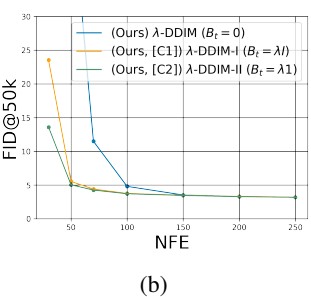 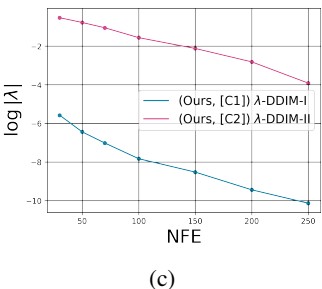

(a)                                      (b)                                      (c)

Figure 2: (Ablation) Conjugate Integrators can significantly improve deterministic sampling efficiency in PSLD for CIFAR-10. a) The Conjugate Integrator proposed in Eqn. 8 ($B_t = 0$) outperforms Euler applied directly to the Prob. Flow ODE. b) Comparison between different choices of $B_t$. c) Impact of the number of diffusion steps on the optimal $\lambda$ value in $\lambda$-DDIM.

for a PSLD model pre-trained on CIFAR-10. We measure sampling efficiency via network function evaluations (network function evaluations (NFE)) and measure sample quality using FID (Heusel et al., 2017). See Appendix E for all implementation details. In Fig. 2a, we find that even with a straightforward choice of $B_t = 0$ (which corresponds to DDIM), the conjugate integrator in Eqn. 8 significantly outperforms Euler applied to the PSLD ODE in the original space, confirming the efficacy of DDIM. We next discuss other choices of $B_t$, which help us generalize beyond exponential integrators and further improve sampling efficiency.

**Beyond Exponential Integrators.** To derive more efficient samplers, we study conjugate integrators with $B_t = \lambda I$ and $B_t = \lambda 1$ where $1$ is a matrix of all ones, and $\lambda$ is a scalar hyperparameter. For a fixed compute budget, we tune $\lambda$ during sampling to optimize for sample quality. We denote the resulting conjugate integrators as $\lambda$-DDIM-I and $\lambda$-DDIM-II, respectively. Empirically, in the context of PSLD, tuning $\lambda$ during sampling can lead to significant improvements in sampling efficiency (see Fig. 2b) over setting $\lambda = 0$ (which corresponds to DDIM or exponential integrators). Moreover, we find that the optimal values of $\lambda$ for both our choices of $B_t$ decrease in magnitude as the sampling budget increases (see Fig. 2c), suggesting that all three schemes are likely to perform similarly for a larger sampling budget. Next, we provide a theoretical justification for improved sample quality for non-zero $\lambda$ values using stability analysis for numerical methods.

**Stability of Conjugate Integrators.** Despite impressive empirical performance, it is unclear why non-zero $\lambda$ values in $\lambda$-DDIM improve sample quality, particularly at large step sizes $h$ (i.e. for a small number of reverse diffusion steps). To this end, we analyze the stability of the conjugate integrator proposed in Eqn. 8 and present the following result:

**Theorem 2.** *Let $U\Lambda U^{-1}$ denote the eigendecomposition of the matrix $\frac{1}{2}G_t G_t^T C_{out}(t)\frac{\partial \epsilon_\theta(C_{in}(t)z_t,t)}{\partial z_t}$. Under invertibility of $A_t$ and certain regularity conditions (as stated in Appendix B.5), the conjugate integrator defined in Eqn. 8 is stable if the eigenvalues $\tilde{\lambda}$ of the matrix $\bar{\Lambda} = \Lambda - U^{-1}B_t U$ satisfy $|1 + h\tilde{\lambda}| \leq 1$. (See Appendix B.5 for a proof)*

**Corollary 1.** *$\lambda$-DDIM-I is stable if $|1 + h(\bar{\lambda} - \lambda)| \leq 1$ where $\bar{\lambda} \in \Lambda$.*

In the context of $\lambda$-DDIM-I, the result in Corollary 1 implies that tuning the hyperparameter $\lambda$ *conditions* the eigenvalues of $\Lambda$ during sampling. This results in a more stable integrator, likely leading to good sample quality even for a large step size $h$. In contrast, setting $\lambda = 0$ disables this conditioning, leading to worse sample quality if the eigenvalues $\bar{\lambda}$ are not already well-conditioned.

**Discussion.** In this section, we introduced Conjugate Integrators for constructing efficient deterministic samplers for diffusion models. In addition to establishing connections with prior work on deterministic sampling, we propose a novel conjugate integrator, $\lambda$-DDIM, that generalizes samplers based on exponential integrators. Lastly, we provide theoretical results that justify the effectiveness of the proposed sampler. However, while we apply the Euler method to the transformed ODE in Eqn. 7, other numerical schemes can also be used. Consequently, our result in Theorem 2 is specific to this case, and we leave deriving similar results for other integrators applied to the transformed ODE in Eqn. 7 as future work. Lastly, while $\lambda$-DDIM-II (Fig. 2b) performs the best, further exploration of better choices of $B_t$ also remains an interesting direction for future work.

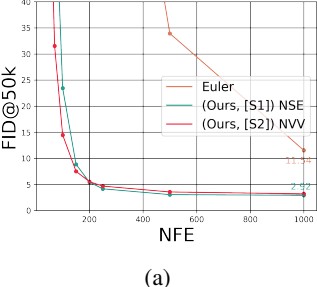 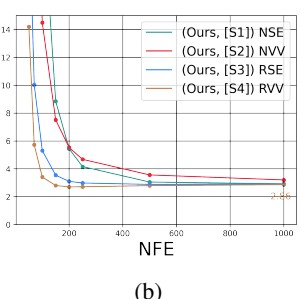 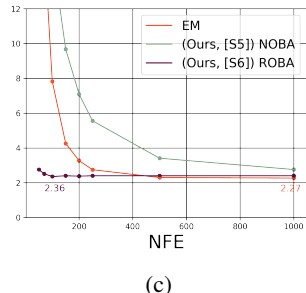

|     |     |     |
| --- | --- | --- |
| (a) | (b) | (c) |

Figure 3: (Ablation) Splitting Integrators significantly improve deterministic/stochastic sampling efficiency in PSLD for CIFAR-10. a) Naive ODE splitting samplers outperform Euler by a large margin. b) Reduced ODE splitting samplers outperform naive schemes. c) Reduced SDE splitting samplers outperform other baselines.

### 3.2 SPLITTING INTEGRATORS FOR FAST ODE AND SDE SAMPLING

We bring another innovation for faster sampling to generative diffusion models. The methods described here are complementary to conjugate integrators, and in Section 3.3, we will study their combined strength. Splitting integrators are commonly used to design symplectic numerical solvers for molecular dynamics systems (Leimkuhler, 2015) which preserve a certain geometric property of the underlying physical system. However, their application for fast diffusion sampling is still underexplored (Dockhorn et al., 2022b). The main intuition behind splitting integrators is to *split* an ODE/SDE into subcomponents which are then *independently solved* numerically (or analytically). The resulting independent updates are then *composed* in a specific order to obtain the final solution. Splitting integrators are particularly suited for augmented diffusion models since they can leverage the split into position and momentum variables for faster sampling. We provide a brief introduction to splitting integrators in Appendix C.1 and refer interested readers to Leimkuhler (2015) for a detailed discussion.

**Setup**: We use the same setup and experimental protocol from Section 3.1 and develop splitting integrators for the PSLD Prob. Flow ODE and Reverse SDE. Though our discussion is primarily focused on PSLD, the idea of splitting is general and can also be applied to other types of diffusion models (Wizadwongsa & Suwajanakorn, 2023; Dockhorn et al., 2022b).

**Deterministic Splitting Integrators.** We choose the following splitting scheme for the PSLD ODE,

$$\begin{pmatrix} d\bar{\mathbf{x}}_t \\ d\bar{\mathbf{m}}_t \end{pmatrix} = \underbrace{\frac{\beta}{2} \begin{pmatrix} \Gamma\bar{\mathbf{x}}_t - M^{-1}\bar{\mathbf{m}}_t + \Gamma \boldsymbol{s}_\theta^x(\bar{\mathbf{z}}_t, T-t) \\ 0 \end{pmatrix} dt}_{A} + \underbrace{\frac{\beta}{2} \begin{pmatrix} 0 \\ \bar{\mathbf{x}}_t + \nu\bar{\mathbf{m}}_t + M\nu \boldsymbol{s}_\theta^m(\bar{\mathbf{z}}_t, T-t) \end{pmatrix} dt}_{B},$$

where $\bar{\mathbf{x}}_t = \mathbf{x}_{T-t}$, $\bar{\mathbf{m}}_t = \mathbf{m}_{T-t}$, $\boldsymbol{s}_\theta^x$ and $\boldsymbol{s}_\theta^m$ denote the score components in the data and momentum space, respectively. Given step size $h$, we denote the Euler updates for the components $A$ and $B$ as $\mathcal{L}_h^A$ and $\mathcal{L}_h^B$ respectively. Consequently, we propose two composition schemes namely, $\mathcal{L}_h^{[BA]} = \mathcal{L}_h^A \circ \mathcal{L}_h^B$ and $\mathcal{L}_h^{[BAB]} = \mathcal{L}_{h/2}^B \circ \mathcal{L}_h^A \circ \mathcal{L}_{h/2}^B$, where $h/2$ denotes an update with half-step. We denote the samplers corresponding to these schemes as **Naive Symplectic Euler (NSE)** and **Naive Velocity Verlet (NVV)**, respectively (see Appendix C.2.1 for exact numerical updates). While the motivation behind the notation "naive" will become clear later, even a direct application of our naive splitting samplers can lead to substantial improvements in sample efficiency over Euler (see Fig. 3a). This is intuitive since, unlike Euler, the proposed naive samplers alternate between updates in the momentum and the position space, thus exploiting the coupling between the data and the momentum variables. We formalize this intuition as the following result.

**Theorem 3.** *Given a step size $h$, the NVV sampler has local truncation errors with orders $\mathcal{O}(\Gamma h^2)$ and $\mathcal{O}(\nu h^2)$ in the position and momentum space, respectively* (See Appendix C.2.4 for proof).

Since the choice of $\Gamma$ in PSLD is usually comparable to the step size $h$ (Pandey & Mandt, 2023), the local truncation error for the NVV sampler in the position space is usually $\mathcal{O}(h^3)$. However, Fig.

3a also suggests that naive splitting schemes exhibit poor sample quality at low NFE budgets. This suggests the need for a deeper insight into the error analysis for the naive schemes.

Therefore, based on local error analysis for ODEs, we propose the following *improvements* to our naive samplers.

- We reuse the score function evaluation between the first consecutive position and the momentum updates in both the NSE and the NVV samplers.
- Next, for NVV, we use the score function evaluation $s_\theta(\mathbf{x}_{t+h}, \mathbf{m}_{t+h/2}, T - (t + h))$ in the last update step instead.

Consequently, we denote the resulting samplers as **Reduced Symplectic Euler (RSE)** and **Reduced Velocity Verlet (RVV)**, respectively (see Appendix C.2.2 for exact numerical updates). Though both the naive and the reduced schemes have the same convergence order (see Appendix C.2.5), the reduced schemes significantly improve PSLD sampling efficiency over their naive counterparts (Fig. 3b). This is because our proposed adjustments serve two benefits: Firstly, the number of NFEs per update step is reduced by one, enabling smaller step sizes for the same sampling budget. This reduces numerical error during sampling. Secondly, our proposed adjustments lead to the cancellation of certain error terms, which is especially helpful for large step sizes during sampling (see Appendix C.2.4 for a theoretical analysis).

**Stochastic Splitting Integrators.** Analogously, we can also apply splitting integrators to the PSLD Reverse SDE. Based on initial experimental results, we use the following splitting scheme.

$$\begin{pmatrix} d\bar{\mathbf{x}}_t \\ d\bar{\mathbf{m}}_t \end{pmatrix} = \underbrace{\frac{\beta}{2} \begin{pmatrix} 2\Gamma\bar{\mathbf{x}}_t - M^{-1}\bar{\mathbf{m}}_t + 2\Gamma s_\theta^x(\bar{\mathbf{z}}_t, t) \\ 0 \end{pmatrix} dt}_{A} + O + \underbrace{\frac{\beta}{2} \begin{pmatrix} 0 \\ \bar{\mathbf{x}}_t + 2\nu\bar{\mathbf{m}}_t + 2M\nu s_\theta^m(\bar{\mathbf{z}}_t, t) \end{pmatrix} dt}_{B} .$$

where $O = \begin{pmatrix} -\frac{\beta\Gamma}{2}\bar{\mathbf{x}}_t dt + \sqrt{\beta\Gamma}d\bar{\mathbf{w}}_t \\ -\frac{\beta\nu}{2}\bar{\mathbf{m}}_t dt + \sqrt{M\nu\beta}d\bar{\mathbf{w}}_t \end{pmatrix}$ represents the Ornstein-Uhlenbeck process in the joint space. Among several possible composition schemes, we found the schemes OBA, BAO, and OBAB to work particularly well. We discuss $\mathcal{L}_h^{[OBA]} = \mathcal{L}_h^A \circ \mathcal{L}_h^B \circ \mathcal{L}_h^O$, which we denote as **Naive OBA (NOBA)**, in more details here and defer all discussion related to other schemes to Appendix C.3. Analogous to the deterministic setting, we propose several adjustments over the naive scheme.

- We reuse the score function evaluation between the position and the momentum updates, which leads to improved sampling efficiency over the naive scheme (Fig. 3c).
- Next, similar to Karras et al. (2022), we introduce a parameter $\lambda_s$ in the position space update for $\mathcal{L}_O$ to control the amount of noise injected in the position space. However, adding a similar parameter in the momentum space led to unstable behavior and, therefore, restricted this adjustment to the position space.

With these adjustments, we denote the resulting sampler as **Reduced OBA (ROBA)** (see Appendix C.3.3 for full numerical updates). Empirically, the ROBA sampler with a tuned $\lambda_s$ outperforms other baselines by a significant margin (see Fig. 3c).

**Discussion.** In this section, we presented Splitting Integrators for constructing efficient deterministic and stochastic samplers for diffusion models. We construct splitting integrators with alternating updates in the position and momentum variables, leading to higher-order integrators. However, a naive application of splitting integrators can be sub-optimal. Consequently, we propose principled adjustments for naive splitting samplers, which lead to significant improvements. However, a more principled theoretical investigation in the role of $\lambda_s$ remains an interesting direction for future work.

### 3.3 COMBINING SPLITTING AND CONJUGATE INTEGRATORS

In the context of Splitting Integrators, so far, we have used Euler for numerically solving each splitting component. However, in principle, each splitting component can also be solved using more efficient numerical schemes like Conjugate Integrators discussed in Section 3.1. We refer to the latter as *Conjugate Splitting Integrators*. For subsequent discussions, we combine the $\lambda$-DDIM-II conjugate integrator proposed in Section 3.1 and the reduced splitting samplers discussed

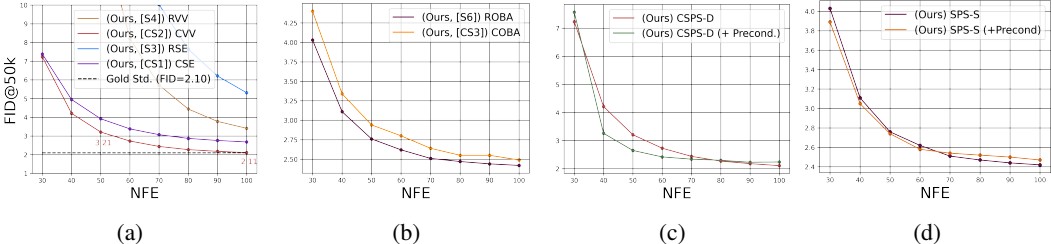

(a)             (b)             (c)             (d)

Figure 4: (Ablation) a) Conjugate-splitting samplers outperform their reduced counterparts for deterministic sampling. b) For stochastic sampling, however, using conjugate-splitting samplers incur a slight degradation in sample quality over the reduced scheme. (c, d) Impact of preconditioning on sample quality for the proposed ODE (Left) and SDE (Right) samplers at low sampling budgets.

in Section 3.2. Consequently, we denote the resulting deterministic samplers as **Conjugate Velocity Verlet (CVV)** and **Conjugate Symplectic Euler (CSE)** corresponding to their reduced counterparts. Similarly, we denote the resulting stochastic sampler as **Conjugate OBA (COBA)**.

**Conjugacy in the position vs. momentum space.** Our initial empirical results indicated that applying conjugacy in the position space yields the most significant gains in sample quality. This might be intuitive since, during reverse diffusion sampling, the dynamics in the position space might be more complex due to a more complex equilibrium distribution. Therefore in this work, we apply conjugacy only in the position space updates (see Appendix D for full update steps).

**Empirical Evaluation.** Fig. 4a illustrates the benefits of using the proposed conjugate-splitting samplers, CVV and CSE, over their corresponding reduced schemes for deterministic sampling on the unconditional CIFAR-10 dataset. Notably, the proposed CVV sampler achieves an FID score of **2.11** within a sampling budget of 100 NFEs which is comparable to the FID score of 2.10 reported in PSLD (Pandey & Mandt, 2023), which requires 242 NFE. For stochastic sampling, we find that applying conjugate integrators to the ROBA sampler slightly degrades sample quality (see Fig. 4b) for CIFAR-10. However, the benefits of the latter are more prominent for higher-resolution datasets (see Section 4), indicating the scalability of the proposed sampler.

## 4 ADDITIONAL EXPERIMENTAL RESULTS

**Ablation Summary.** We summarize our ablation samplers presented in Section 3 in Table 14. In short, we presented Conjugate Integrators in Section 3.1, which enable efficient deterministic sampling in PSLD (Fig. 2). Next, we presented Reduced Splitting Integrators for faster deterministic and stochastic sampling in PSLD (Fig. 3). Lastly, we combined the two frameworks for further gains in sampling efficiency (Fig. 4). *We now present additional quantitative results and comparisons with prior methods for faster deterministic and stochastic sampling.*

**Notation.** For simplicity, we denote our best-performing Reduced Splitting and Conjugate Splitting integrators as *Splitting-based PSLD Sampler (SPS)* and *Conjugate Splitting-based PSLD Sampler (CSPS)*, respectively. Consequently, we refer to the Deterministic RVV and Stochastic ROBA samplers as SPS-D and SPS-S, and their conjugate variants as CSPS-D and CSPS-S, respectively.

**Datasets and Evaluation Metrics.** We use the CIFAR-10, CelebA-64 (Liu et al., 2015) and the AFHQ-v2 (Choi et al., 2020) datasets for comparisons. Unless specified otherwise, we report FID for 50k generated samples for all datasets and quantify sampling efficiency using NFE. We include full experimental details in Appendix E.

**Baselines and setup.** In addition to samplers based on exponential integrators like DDIM (Song et al., 2021), DEIS (Zhang & Chen, 2023) and DPM-Solver (Lu et al., 2022), we compare our best ODE and SDE samplers with PNDM (Liu et al., 2022), EDM (Karras et al., 2022), SA-Solver (Xue et al., 2023) and Analytic DPM (Bao et al., 2022). We provide a brief description of these baselines in Table 1. While the techniques presented in this work generally apply to other types of diffusion models, we compare the empirical performance of our proposed samplers for PSLD with the highlighted baselines for completeness. Lastly, we find that, similar to prior works (Dockhorn

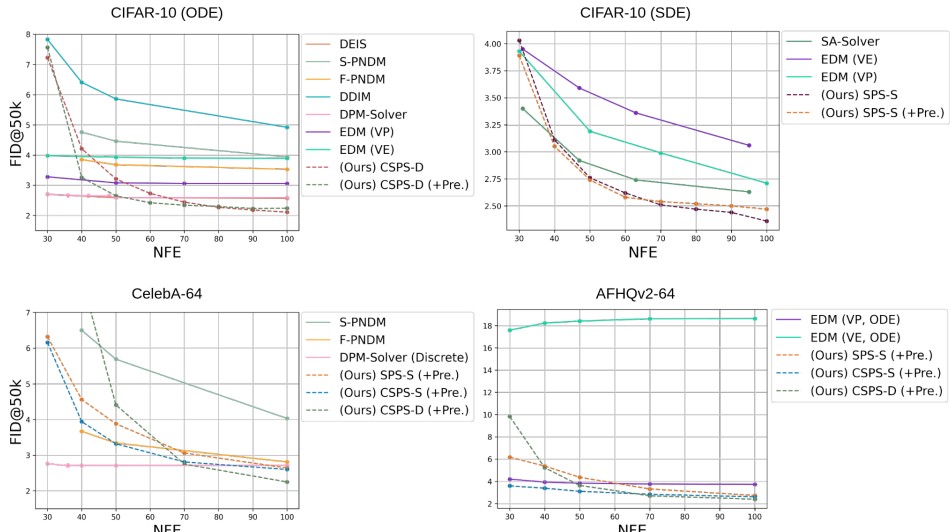

Figure 5: Extended results for Table 1. Our proposed samplers perform comparably or outperform other baselines for similar NFE budgets for the CIFAR-10, CelebA-64, and AFHQv2 datasets.

et al., 2022b; Karras et al., 2022), score network preconditioning leads to better sample quality at low sampling budgets for both deterministic (Fig. 4c) and stochastic sampling (Fig. 4d). For instance, CSPS-D achieves an FID score of 2.65 in NFE=50 with preconditioning as compared to 3.21 without. We provide full technical details for our preconditioning setup in Appendix E.3. Consequently, we report empirical results for our ODE/SDE samplers with and without preconditioning for CIFAR-10 and with preconditioning for other datasets.

**Empirical Observations:** For CIFAR-10, our ODE sampler performs comparably or outperforms all other baselines for NFE $\geq$ 50 (Fig. 5, Top Left). Similarly, our SDE sampler outperforms all other baselines for NFE $\geq$ 40 (Fig. 5, Top Right). We make similar observations for the CelebA-64 and AFHQv2-64 datasets, primarily in the high NFE regime (See Fig. 5, Bottom Left). Therefore, our proposed samplers for PSLD are competitive with recent work. Interestingly, for all datasets, our stochastic sampler achieves better sample quality for low sampling budgets (NFE $<$ 50) as compared to our deterministic sampler. Lastly, in contrast to CIFAR-10, we find that the CSPS-S sampler works better than the SPS-S sampler for the CelebA-64 and AFHQv2-64 datasets, indicating its effectiveness for higher-resolution sampling.

## 5 DISCUSSION

**Contributions.** We have presented two complementary frameworks, *Conjugate* and *Splitting* Integrators, for efficient deterministic and stochastic sampling from a broader class of diffusion models. Furthermore, we combine the two frameworks and propose *Conjugate Splitting Integrators* for further improvements in sampling efficiency in the context of PSLD. The resulting samplers perform comparably with several recent approaches for fast diffusion sampling (see Table 1, Fig. 5).

**Future Directions.** While the framework presented in this work can serve as a good starting point for designing efficient samplers for diffusion models, there are several promising future directions. Firstly, extending our framework of conjugate integrators to design efficient stochastic sampling methods would be interesting. Secondly, our current choice of the core design parameters in conjugate integrators ($\boldsymbol{B}_t$) is mostly heuristical and, therefore, requires further theoretical investigation. In the context of stochastic sampling, firstly, we find that empirically controlling the amount of stochasticity injected during sampling can largely affect sample quality. Therefore, further investigation into the theoretical aspects of optimal noise injection in diffusion model sampling can be an interesting direction for future work. Lastly, we hope that our presentation of reduced splitting integrators for fast deterministic and stochastic sampling in diffusion models can serve as a good initial starting point for further research in exploring more efficient splitting techniques.

ACKNOWLEDGEMENTS

KP acknowledges support from the Bosch Center for Artificial Intelligence and the HPI Research Center in Machine Learning and Data Science at UC Irvine. SM acknowledges support from the National Science Foundation (NSF) under an NSF CAREER Award, award numbers 2003237 and 2007719, by the Department of Energy under grant DE-SC0022331, the IARPA WRIVA program, and by gifts from Qualcomm and Disney.

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

CONTENTS

## A   RELATED WORK

In addition to the recent work based on exponential integrators (Zhang & Chen, 2023; Lu et al., 2022; Zhang et al., 2022; Song et al., 2021), PNDM (Liu et al., 2022) re-casts the sampling process in DDPM (Ho et al., 2020) as numerically solving differential equations on manifolds. Additionally, Karras et al. (2022) highlight and optimize several design choices in diffusion model training (including score network preconditioning, improved network architectures, and improved data augmentation) and sampling (including improved time-discretization schedules), which leads to significant improvements in sample quality during inference. While this is not our primary focus, exploring these choices in the context of other diffusions like PSLD (Pandey & Mandt, 2023) could be an interesting direction for future work. Other works for faster sampling have also focused on using adaptive solvers (Jolicoeur-Martineau et al., 2021a), optimal variance during sampling (Bao et al., 2022), and optimizing timestep schedules (Watson et al., 2021). Though prior works have focused mostly on speeding up deterministic sampling, there have also been some recent advances in speeding up stochastic sampling in diffusion models (Karras et al., 2022; Xue et al., 2023; Gonzalez et al., 2023).

Splitting integrators are extensively used in the design of symplectic integrators in molecular dynamics (Leimkuhler, 2015; Yoshida, 1990; Verlet, 1967; Trotter, 1959). However, their application for efficient sampling in diffusion models is only explored by a few works (Dockhorn et al., 2022b; Wizadwongsa & Suwajanakorn, 2023). In this work, in the context of PSLD, we show the structure in the diffusion model ODE/SDE can be used to design efficient splitting-based samplers. However, as shown in this work, a naive application of splitting integrators can be sub-optimal for sample quality, and careful analysis might be required to design splitting integrators for diffusion models.

Lastly, another line of research for fast diffusion model sampling involves additional training (Song et al., 2023; Dockhorn et al., 2022a; Salimans & Ho, 2022; Meng et al., 2023; Luhman & Luhman, 2021). In contrast, our proposed framework does not require additional training during inference.

## B   CONJUGATE INTEGRATORS FOR FASTER ODE SAMPLING

### B.1   PROOF OF THEOREM 1

We restate the full theorem for completeness.

**Theorem.** *Let* $\mathbf{z}_t$ *evolve according to the probability-flow ODE in Eqn. 3 with the score function parameterization given in Eqn. 5. For any mapping* $B : [0, T] \times \mathbb{R}^d \to \mathbb{R}^d$ *and* $\boldsymbol{A}_t$, $\boldsymbol{\Phi}_t$ *given by Eqn. 6, the probability flow ODE in the projected space* $\hat{\mathbf{z}}_t = \boldsymbol{A}_t \mathbf{z}_t$ *is given by*

$$d\hat{\mathbf{z}}_t = \boldsymbol{A}_t \boldsymbol{B}_t \boldsymbol{A}_t^{-1} \hat{\mathbf{z}}_t dt + d\boldsymbol{\Phi}_t \boldsymbol{\epsilon_\theta} \left( \boldsymbol{C}_{in}(t) \boldsymbol{A}_t^{-1} \hat{\mathbf{z}}_t, C_{noise}(t) \right) \tag{9}$$

The forward process for a diffusion with affine drift can be specified as:

$$d\mathbf{z}_t = \boldsymbol{F}_t \mathbf{z}_t \, dt + \boldsymbol{G}_t \, d\mathbf{w}_t. \tag{10}$$

Consequently, the probability flow ODE corresponding to the process in Eqn. 10 is given by:

$$d\mathbf{z}_t = \left[ \boldsymbol{F}_t \mathbf{z}_t - \frac{1}{2} \boldsymbol{G}_t \boldsymbol{G}_t^\top \boldsymbol{s}_{\boldsymbol{\theta}}(\mathbf{z}_t, t) \right] dt. \tag{11}$$

Furthermore, the score network is parameterized as follows:

$$\boldsymbol{s}_{\boldsymbol{\theta}}(\mathbf{z}_t, t) = \boldsymbol{C}_{\text{skip}}(t) \mathbf{z}_t + \boldsymbol{C}_{\text{out}}(t) \boldsymbol{\epsilon}_{\boldsymbol{\theta}}(\boldsymbol{C}_{\text{in}}(t) \mathbf{z}_t, C_{\text{noise}}(t)) \tag{12}$$

Substituting the score network parameterization in Eqn. 11, we have the following form of the probability flow ODE:

$$\frac{d\mathbf{z}_t}{dt} = \boldsymbol{F}_t \mathbf{z}_t - \frac{1}{2} \boldsymbol{G}_t \boldsymbol{G}_t^\top \left[ \boldsymbol{C}_{\text{skip}}(t) \mathbf{z}_t + \boldsymbol{C}_{\text{out}}(t) \boldsymbol{\epsilon}_{\boldsymbol{\theta}}(\boldsymbol{C}_{\text{in}}(t) \mathbf{z}_t, C_{\text{noise}}(t)) \right] \tag{13}$$

$$= \left[ \boldsymbol{F}_t - \frac{1}{2} \boldsymbol{G}_t \boldsymbol{G}_t^\top \boldsymbol{C}_{\text{skip}}(t) \right] \mathbf{z}_t - \frac{1}{2} \boldsymbol{G}_t \boldsymbol{G}_t^\top \boldsymbol{C}_{\text{out}}(t) \boldsymbol{\epsilon}_{\boldsymbol{\theta}}(\boldsymbol{C}_{\text{in}}(t) \mathbf{z}_t, C_{\text{noise}}(t)) \tag{14}$$

Given an affine transformation which projects the state $\mathbf{z}_t$ to $\hat{\mathbf{z}}_t$,

$$\hat{\mathbf{z}}_t = \boldsymbol{A}_t \mathbf{z}_t \tag{15}$$

Therefore, by the Chain Rule of calculus,

$$\frac{d\hat{\mathbf{z}}_t}{dt} = \frac{d\boldsymbol{A}_t}{dt} \mathbf{z}_t + \boldsymbol{A}_t \frac{d\mathbf{z}_t}{dt} \tag{16}$$

Substituting the ODE in Eqn. 14 in Eqn. 16,

$$\frac{d\hat{\mathbf{z}}_t}{dt} = \frac{d\boldsymbol{A}_t}{dt} \mathbf{z}_t + \boldsymbol{A}_t \left[ \left( \boldsymbol{F}_t - \frac{1}{2} \boldsymbol{G}_t \boldsymbol{G}_t^\top \boldsymbol{C}_{\text{skip}}(t) \right) \mathbf{z}_t - \frac{1}{2} \boldsymbol{G}_t \boldsymbol{G}_t^\top \boldsymbol{C}_{\text{out}}(t) \boldsymbol{\epsilon}_{\boldsymbol{\theta}}(\boldsymbol{C}_{\text{in}}(t) \mathbf{z}_t, C_{\text{noise}}(t)) \right] \tag{17}$$

$$= \left[ \frac{d\boldsymbol{A}_t}{dt} + \boldsymbol{A}_t \left( \boldsymbol{F}_t - \frac{1}{2} \boldsymbol{G}_t \boldsymbol{G}_t^\top \boldsymbol{C}_{\text{skip}}(t) \right) \right] \mathbf{z}_t - \frac{1}{2} \boldsymbol{A}_t \boldsymbol{G}_t \boldsymbol{G}_t^\top \boldsymbol{C}_{\text{out}}(t) \boldsymbol{\epsilon}_{\boldsymbol{\theta}}(\boldsymbol{C}_{\text{in}}(t) \mathbf{z}_t, C_{\text{noise}}(t)) \tag{18}$$

$$= \left[ \frac{d\boldsymbol{A}_t}{dt} + \boldsymbol{A}_t \left( \boldsymbol{F}_t - \frac{1}{2} \boldsymbol{G}_t \boldsymbol{G}_t^\top \boldsymbol{C}_{\text{skip}}(t) \right) \right] \boldsymbol{A}_t^{-1} \hat{\mathbf{z}}_t - \frac{1}{2} \boldsymbol{A}_t \boldsymbol{G}_t \boldsymbol{G}_t^\top \boldsymbol{C}_{\text{out}}(t) \boldsymbol{\epsilon}_{\boldsymbol{\theta}}(\boldsymbol{C}_{\text{in}}(t) \boldsymbol{A}_t^{-1} \hat{\mathbf{z}}_t, C_{\text{noise}}(t)) \tag{19}$$

We further define the matrix coefficients $\boldsymbol{B}_t$ and $\boldsymbol{\Phi}_t$ such that,

$$\frac{d\boldsymbol{A}_t}{dt} + \boldsymbol{A}_t \left( \boldsymbol{F}_t - \frac{1}{2} \boldsymbol{G}_t \boldsymbol{G}_t^\top \boldsymbol{C}_{\text{skip}}(t) \right) = \boldsymbol{A}_t \boldsymbol{B}_t \tag{20}$$

$$\frac{d\boldsymbol{\Phi}_t}{dt} = -\frac{1}{2} \boldsymbol{A}_t \boldsymbol{G}_t \boldsymbol{G}_t^\top \boldsymbol{C}_{\text{out}}(t) \tag{21}$$

which yields the required diffusion ODE in the projected space:

$$\frac{d\hat{\mathbf{z}}_t}{dt} = \boldsymbol{A}_t \boldsymbol{B}_t \boldsymbol{A}_t^{-1} \hat{\mathbf{z}}_t + \frac{d\boldsymbol{\Phi}_t}{dt} \boldsymbol{\epsilon}_{\boldsymbol{\theta}} \left( \boldsymbol{C}_{\text{in}}(t) \boldsymbol{A}_t^{-1} \hat{\mathbf{z}}_t, C_{\text{noise}}(t) \right) \tag{22}$$

### B.2 PROOF OF PROPOSITION 1: CONNECTION WITH DDIM

**Proposition.** *For the VP-SDE (Song et al., 2020), for the choice of $\boldsymbol{B}_t = \boldsymbol{0}$, the transformed ODE in Eqn. 7 corresponds to the DDIM ODE proposed in Song et al. (2021)*

*Proof.* The forward process for the VP-SDE (Song et al., 2020) is given by:

$$d\mathbf{z}_t = -\frac{1}{2}\beta_t\mathbf{z}_t dt + \sqrt{\beta_t}d\mathbf{w}_t \tag{23}$$

where $\beta_t$ determines the noise schedule. This implies $\boldsymbol{F}_t = -\frac{1}{2}\beta_t\boldsymbol{I}_d$ and $\boldsymbol{G}_t = \sqrt{\beta_t}\boldsymbol{I}_d$. Furthermore, the score network in the VP-SDE is often parameterized as $\boldsymbol{s}_{\boldsymbol{\theta}}(\mathbf{z}_t, t) = -\boldsymbol{\epsilon}_{\boldsymbol{\theta}}(\mathbf{z}_t, t)/\sigma_t$ where $\sigma_t^2$ is the variance of the perturbation kernel $p(\mathbf{z}_t|\mathbf{z}_0)$. It follows that for VP-SDE,

$$\boldsymbol{C}_{\text{skip}}(t) = \boldsymbol{0}, \quad \boldsymbol{C}_{\text{out}}(t) = -\frac{1}{\sigma_t}, \quad \boldsymbol{C}_{\text{in}}(t) = \boldsymbol{I}_d, \quad \boldsymbol{C}_{\text{noise}}(t) = t. \tag{24}$$

Setting $\boldsymbol{B}_t = \boldsymbol{0}$, we can determine the coefficients $\boldsymbol{A}_t$ and $\boldsymbol{\Phi}_t$ as follows:

$$\frac{d\boldsymbol{A}_t}{dt} + \boldsymbol{A}_t\left(\boldsymbol{F}_t - \frac{1}{2}\boldsymbol{G}_t\boldsymbol{G}_t^\top\boldsymbol{C}_{\text{skip}}(t)\right) = \boldsymbol{A}_t\boldsymbol{B}_t \quad \Rightarrow \quad \frac{d\boldsymbol{A}_t}{dt} - \frac{1}{2}\beta_t\boldsymbol{A}_t = \boldsymbol{0} \tag{25}$$

$$\boldsymbol{A}_t = \exp\left(\frac{1}{2}\int_0^t \beta_s ds\right)\boldsymbol{I}_d \tag{26}$$

Similarly,

$$\frac{d\boldsymbol{\Phi}_t}{dt} = -\frac{1}{2}\boldsymbol{A}_t\boldsymbol{G}_t\boldsymbol{G}_t^\top\boldsymbol{C}_{\text{out}}(t) = \frac{1}{2}\exp\left(\frac{1}{2}\int_0^t \beta_s ds\right)\frac{\beta_t}{\sigma_t}\boldsymbol{I}_d \tag{27}$$

Since the variance of the perturbation kernel $p(\mathbf{x}_t|\mathbf{x}_0)$ is given by $\sigma_t^2 = \left[1 - \exp\left(-\int_0^t \beta_s ds\right)\right]$, we can reformulate the above ODE as:

$$\frac{d\boldsymbol{\Phi}_t}{dt} = \frac{\beta_t}{2\sigma_t\sqrt{1-\sigma_t^2}}\boldsymbol{I}_d \tag{28}$$

Consequently, the ODE in the transformed space can be specified as:

$$\frac{d\hat{\mathbf{z}}_t}{dt} = \boldsymbol{A}_t\boldsymbol{B}_t\boldsymbol{A}_t^{-1}\hat{\mathbf{z}}_t + \frac{d\boldsymbol{\Phi}_t}{dt}\boldsymbol{\epsilon}_{\boldsymbol{\theta}}\left(\boldsymbol{C}_{\text{in}}(t)\boldsymbol{A}_t^{-1}\hat{\mathbf{z}}_t, \boldsymbol{C}_{\text{noise}}(t)\right) \tag{29}$$

$$= \frac{\beta_t}{2\sigma_t\sqrt{1-\sigma_t^2}}\boldsymbol{\epsilon}_{\boldsymbol{\theta}}\left(\sqrt{1-\sigma_t^2}\hat{\mathbf{z}}_t, t\right) \tag{30}$$

Defining $\gamma_t = \sigma_t/\sqrt{1-\sigma_t^2}$, it can be shown that, $d\gamma_t = \frac{\beta_t}{2\sigma_t\sqrt{1-\sigma_t^2}}dt$. Therefore, reformulating the ODE in Eqn. 30 in terms of $\gamma_t$,

$$\frac{d\hat{\mathbf{z}}_t}{d\gamma_t} = \boldsymbol{\epsilon}_{\boldsymbol{\theta}}\left(\frac{\hat{\mathbf{z}}_t}{\sqrt{1+\gamma_t^2}}, t\right) \tag{31}$$

which is the DDIM ODE proposed in Song et al. (2021). Therefore for the VP-SDE and the choice of $\boldsymbol{B}_t = \boldsymbol{0}$, the proposed conjugate integrator is equivalent to the DDIM integrator. □

### B.3 PROOF OF PROPOSITION 2: CONNECTIONS WITH DPM-SOLVER

**Proposition.** *For the diffusion model formulation considered in Lu et al. (2022), the exponential integrator proposed in DPM-Solver (Lu et al., 2022) is a numerical integrator for the transformed ODE in Eqn. 8 (See Appendix B.3 for a proof)*

*Proof.* For simplicity, we restrict the parameterization of the score estimator to $\boldsymbol{s}_{\boldsymbol{\theta}}(\mathbf{z}_t, t) = -\boldsymbol{L}_t^{-\top}$, where $\boldsymbol{L}_t$ is the Cholesky decomposition of the variance $\Sigma_t$ of the perturbation kernel. This implies,

$$\boldsymbol{C}_{\text{skip}}(t) = \boldsymbol{0}, \quad \boldsymbol{C}_{\text{out}}(t) = -\boldsymbol{L}_t^{-\top}, \quad \boldsymbol{C}_{\text{in}}(t) = \boldsymbol{I}_d, \quad \boldsymbol{C}_{\text{noise}}(t) = t. \tag{32}$$

Furthermore, for the choice of $\boldsymbol{B}_t = \boldsymbol{0}$, the transformed ODE simplifies to,

$$d\hat{\mathbf{z}}_t = d\boldsymbol{\Phi}_t\boldsymbol{\epsilon}_{\boldsymbol{\theta}}\left(\boldsymbol{A}_t^{-1}\hat{\mathbf{z}}_t, t\right) \tag{33}$$

$$d\hat{\mathbf{z}}_t = \frac{1}{2}\boldsymbol{A}_t\boldsymbol{G}_t\boldsymbol{G}_t^\top\boldsymbol{L}_t^{-\top}\boldsymbol{\epsilon}_{\boldsymbol{\theta}}(\boldsymbol{A}_t^{-1}\hat{\mathbf{z}}_t, t) \tag{34}$$

It follows that for any two timepoints $t$ and $s$, we have,

$$\hat{\mathbf{z}}_t = \hat{\mathbf{z}}_s + \frac{1}{2} \int_s^t \boldsymbol{A}_\tau \boldsymbol{G}_\tau \boldsymbol{G}_\tau^\top \boldsymbol{L}_\tau^{-\top} \boldsymbol{\epsilon_\theta}(\boldsymbol{A}_\tau^{-1}\hat{\mathbf{z}}_\tau, \tau)d\tau \tag{35}$$

$$\boldsymbol{A}_t\mathbf{z}_t = \boldsymbol{A}_s\mathbf{z}_s + \frac{1}{2} \int_s^t \boldsymbol{A}_\tau \boldsymbol{G}_\tau \boldsymbol{G}_\tau^\top \boldsymbol{L}_\tau^{-\top} \boldsymbol{\epsilon_\theta}(\boldsymbol{A}_\tau^{-1}\hat{\mathbf{z}}_\tau, \tau)d\tau \tag{36}$$

$$\mathbf{z}_t = \boldsymbol{A}_t^{-1}\boldsymbol{A}_s\mathbf{z}_s + \frac{\boldsymbol{A}_t^{-1}}{2} \int_s^t \boldsymbol{A}_\tau \boldsymbol{G}_\tau \boldsymbol{G}_\tau^\top \boldsymbol{L}_\tau^{-\top} \boldsymbol{\epsilon_\theta}(\boldsymbol{A}_\tau^{-1}\hat{\mathbf{z}}_\tau, \tau)d\tau \tag{37}$$

Moreover, since $\boldsymbol{B}_t = \mathbf{0}$, we have,

$$\frac{d\boldsymbol{A}_t}{dt} + \boldsymbol{A}_t \left( \boldsymbol{F}_t - \frac{1}{2}\boldsymbol{G}_t\boldsymbol{G}_t^\top \boldsymbol{C}_{\text{skip}}(t) \right) = \boldsymbol{A}_t\boldsymbol{B}_t \tag{38}$$

$$\frac{d\boldsymbol{A}_t}{dt} + \boldsymbol{A}_t\boldsymbol{F}_t = \mathbf{0} \Rightarrow \boldsymbol{F}_t = -\boldsymbol{A}_t^{-1}\frac{d\boldsymbol{A}_t}{dt} \tag{39}$$

Next, we consider diffusions of the form,

$$d\mathbf{z}_t = f(t)\mathbf{z}_tdt + g(t)d\mathbf{w}_t \tag{40}$$

where,

$$f(t) = \frac{d\log\alpha_t}{dt}, \qquad g^2(t) = \frac{d\sigma_t^2}{dt} - 2\frac{d\log\alpha_t}{dt}\sigma_t^2. \tag{41}$$

Here $\alpha(t)$ and $\sigma(t)$ are differentiable functions defining the diffusion process's noise schedule. Moreover, for this process, the score is usually parameterized as $s_\theta(\mathbf{z}_t, t) = -\frac{\epsilon_\theta(\mathbf{z}_t,t)}{\sigma_t}$, implying $\boldsymbol{L}_t^{-\top}$ in our parameterization corresponds to $\sigma_t^{-1}$. Moreover, comparing the drift scaling factors in Eqns. 39 and 41, it follows that $\alpha_t = 1/a_t$ where $\boldsymbol{A}_t = a_t\boldsymbol{I}_d$. Therefore, the integrator in Eqn. 37 can be re-written as,

$$\mathbf{z}_t = \frac{\alpha_t}{\alpha_s}\mathbf{z}_s + \frac{\alpha_t}{2} \int_s^t a_\tau \frac{g_\tau^2}{\sigma_t} \boldsymbol{\epsilon_\theta}(\mathbf{z}_\tau, \tau)d\tau \tag{42}$$

Lastly, defining $\lambda_t = \log\frac{\alpha_t}{\sigma_t}$, it can be shown that $g_t^2 = -2\sigma_t^2\frac{d\lambda_t}{dt}$. Substituting this result for $g_t^2$ in Eqn. 42, we get the following integrator,

$$\mathbf{z}_t = \frac{\alpha_t}{\alpha_s}\mathbf{z}_s - \alpha_t \int_s^t \frac{d\lambda_\tau}{d\tau} \frac{\sigma_\tau}{\alpha_\tau} \boldsymbol{\epsilon_\theta}(\mathbf{z}_\tau, \tau)d\tau \tag{43}$$

Applying change of variables in Eqn. 43 from $\tau$ to $\lambda$, we get the exponential integrator in DPM-Solver (Lu et al., 2022) as follows:

$$\mathbf{z}_t = \frac{\alpha_t}{\alpha_s}\mathbf{z}_s - \alpha_t \int_{\lambda_s}^{\lambda_t} e^{-\lambda}\hat{\boldsymbol{\epsilon}}_{\boldsymbol{\theta}}(\hat{\mathbf{z}}_\lambda, \lambda)d\lambda \tag{44}$$

which concludes our proof. $\qquad\qquad\square$

### B.4 PROOF OF PROPOSITION 2: CONNECTIONS WITH DEIS

**Proposition.** *More generally, for any diffusion model as specified in Eqn. 1, the conjugate integrator update in Eqn. 8 is equivalent to applying the exponential integrator proposed in Zhang & Chen (2023) in the original space $\mathbf{z}_t$. Moreover, using polynomial extrapolation in Zhang & Chen (2023) corresponds to using the explicit Adams-Bashforth solver for the transformed ODE in Eqn. 7.*

*Proof.* For simplicity, we restrict the parameterization of the score estimator to $s_\theta(\mathbf{z}_t, t) = -\boldsymbol{L}_t^{-\top}$, where $\boldsymbol{L}_t$ is the Cholesky decomposition of the variance $\Sigma_t$ of the perturbation kernel. This implies,

$$\boldsymbol{C}_{\text{skip}}(t) = \mathbf{0}, \quad \boldsymbol{C}_{\text{out}}(t) = -\boldsymbol{L}_t^{-\top}, \quad \boldsymbol{C}_{\text{in}}(t) = \boldsymbol{I}_d, \quad \boldsymbol{C}_{\text{noise}}(t) = t. \tag{45}$$

Furthermore, for the choice of $\boldsymbol{B}_t = \mathbf{0}$, the simplified transformed ODE can be specified as:

$$d\hat{\mathbf{z}}_t = d\boldsymbol{\Phi}_t \boldsymbol{\epsilon_\theta}\left( \boldsymbol{A}_t^{-1}\hat{\mathbf{z}}_t, t \right), \tag{46}$$

Subsequently, the update rule for the proposed conjugate integrator reduces to the following form:

$$\hat{\mathbf{z}}_{t-h} = \hat{\mathbf{z}}_t + (\boldsymbol{\Phi}_{t-h} - \boldsymbol{\Phi}_t)\boldsymbol{\epsilon}_{\boldsymbol{\theta}}\left(\boldsymbol{A}_t^{-1}\hat{\mathbf{z}}_t, t\right) \tag{47}$$

where,

$$\frac{d\boldsymbol{A}_t}{dt} + \boldsymbol{A}_t\boldsymbol{F}_t = \mathbf{0} \tag{48}$$

$$\boldsymbol{\Phi}_t = \frac{1}{2}\int_0^t \boldsymbol{A}_s\boldsymbol{G}_s\boldsymbol{G}_s^\top\boldsymbol{L}_s^{-\top}\,ds \tag{49}$$

Transforming the update rule in Eqn. 47 back to the original space,

$$\hat{\mathbf{z}}_{t-h} = \hat{\mathbf{z}}_t + (\boldsymbol{\Phi}_{t-h} - \boldsymbol{\Phi}_t)\boldsymbol{\epsilon}_{\boldsymbol{\theta}}\left(\boldsymbol{A}_t^{-1}\hat{\mathbf{z}}_t, t\right) \tag{50}$$

$$\boldsymbol{A}_{t-h}\mathbf{z}_{t-h} = \boldsymbol{A}_t\mathbf{z}_t + (\boldsymbol{\Phi}_{t-h} - \boldsymbol{\Phi}_t)\boldsymbol{\epsilon}_{\boldsymbol{\theta}}\left(\boldsymbol{A}_t^{-1}\hat{\mathbf{z}}_t, t\right) \tag{51}$$

Pre-multiplying with $\boldsymbol{A}_{t-h}^{-1}$ both sides and substituting the value of $\boldsymbol{\Phi}_t$ from Eqn. 49

$$\mathbf{z}_{t-h} = \boldsymbol{A}_{t-h}^{-1}\boldsymbol{A}_t\mathbf{z}_t + \boldsymbol{A}_{t-h}^{-1}(\boldsymbol{\Phi}_{t-h} - \boldsymbol{\Phi}_t)\boldsymbol{\epsilon}_{\boldsymbol{\theta}}\left(\boldsymbol{A}_t^{-1}\hat{\mathbf{z}}_t, t\right) \tag{52}$$

$$= \boldsymbol{A}_{t-h}^{-1}\boldsymbol{A}_t\mathbf{z}_t + \frac{1}{2}\boldsymbol{A}_{t-h}^{-1}\left(\int_0^{t-h}\boldsymbol{A}_s\boldsymbol{G}_s\boldsymbol{G}_s^\top\boldsymbol{L}_s^{-\top}\,ds - \int_0^t\boldsymbol{A}_s\boldsymbol{G}_s\boldsymbol{G}_s^\top\boldsymbol{L}_s^{-\top}\,ds\right)\boldsymbol{\epsilon}_{\boldsymbol{\theta}}\left(\boldsymbol{A}_t^{-1}\hat{\mathbf{z}}_t, t\right) \tag{53}$$

$$\mathbf{z}_{t-h} = \boldsymbol{A}_{t-h}^{-1}\boldsymbol{A}_t\mathbf{z}_t + \frac{1}{2}\boldsymbol{A}_{t-h}^{-1}\left(\int_t^{t-h}\boldsymbol{A}_s\boldsymbol{G}_s\boldsymbol{G}_s^\top\boldsymbol{L}_s^{-\top}\,ds\right)\boldsymbol{\epsilon}_{\boldsymbol{\theta}}\left(\boldsymbol{A}_t^{-1}\hat{\mathbf{z}}_t, t\right) \tag{54}$$

$$= \boldsymbol{A}_{t-h}^{-1}\boldsymbol{A}_t\mathbf{z}_t + \frac{1}{2}\left(\int_t^{t-h}\boldsymbol{A}_{t-h}^{-1}\boldsymbol{A}_s\boldsymbol{G}_s\boldsymbol{G}_s^\top\boldsymbol{L}_s^{-\top}\,ds\right)\boldsymbol{\epsilon}_{\boldsymbol{\theta}}\left(\boldsymbol{A}_t^{-1}\hat{\mathbf{z}}_t, t\right) \tag{55}$$

Defining $\boldsymbol{\psi}(t, s) = \boldsymbol{A}_t^{-1}\boldsymbol{A}_s$, we can rewrite the update rule in Eqn. 55 as follows:

$$\mathbf{z}_{t-h} = \boldsymbol{\psi}(t - h, t)\mathbf{z}_t + \frac{1}{2}\left(\int_t^{t-h}\boldsymbol{\psi}(t - h, s)\boldsymbol{G}_s\boldsymbol{G}_s^\top\boldsymbol{L}_s^{-\top}\,ds\right)\boldsymbol{\epsilon}_{\boldsymbol{\theta}}\left(\boldsymbol{A}_t^{-1}\hat{\mathbf{z}}_t, t\right) \tag{56}$$

$\square$

The update rule in Eqn. 56 is the same as the *exponential integrator* proposed in Zhang & Chen (2023); Zhang et al. (2022). Furthermore, Zhang & Chen (2023) proposes using polynomial extrapolation to speed up the diffusion process further. We next show that using polynomial extrapolation is equivalent to applying the explicit Adams-Bashforth method to the transformed ODE in Eqn. 46.

**Explicit Adams-Bashforth applied to the transformed ODE**: Given the transformed ODE in Eqn. 46, it follows that,

$$\hat{\mathbf{z}}_{t_i} = \hat{\mathbf{z}}_{t_j} + \int_{t_j}^{t_i} d\boldsymbol{\Phi}_s\boldsymbol{\epsilon}_{\boldsymbol{\theta}}\left(\boldsymbol{A}_s^{-1}\hat{\mathbf{z}}_s, s\right) \tag{57}$$

As done in the explicit Adams-Bashforth method, we can approximate the integrand $\boldsymbol{\epsilon}_{\boldsymbol{\theta}}\left(\boldsymbol{A}_s^{-1}\hat{\mathbf{z}}_s, s\right)$ by a polynomial $P_r(s)$ with degree $r$. As an illustration, for $r = 1$, we have $P_1(s) = \boldsymbol{c}_0 + \boldsymbol{c}_1(s - t_j)$, where the coefficients $\boldsymbol{c}_0$ and $\boldsymbol{c}_1$ are specified as,

$$\boldsymbol{c}_0 = \boldsymbol{\epsilon}_{\boldsymbol{\theta}}\left(\boldsymbol{A}_{t_j}^{-1}\hat{\mathbf{z}}_{t_j}, t_j\right), \qquad \boldsymbol{c}_1 = \frac{1}{t_{j-1} - t_j}\left[\boldsymbol{\epsilon}_{\boldsymbol{\theta}}\left(\boldsymbol{A}_{t_{j-1}}^{-1}\hat{\mathbf{z}}_{t_{j-1}}, t_{j-1}\right) - \boldsymbol{\epsilon}_{\boldsymbol{\theta}}\left(\boldsymbol{A}_{t_j}^{-1}\hat{\mathbf{z}}_{t_j}, t_j\right)\right] \tag{58}$$

Therefore we have the polynomial approximation $P_1(s)$ for $\boldsymbol{\epsilon}_{\boldsymbol{\theta}}\left(\boldsymbol{A}_s^{-1}\hat{\mathbf{z}}_s, s\right)$ as,

$$P_1(s) = \boldsymbol{\epsilon}_{\boldsymbol{\theta}}\left(\boldsymbol{A}_{t_j}^{-1}\hat{\mathbf{z}}_{t_j}, t_j\right) + \frac{s - t_j}{t_{j-1} - t_j}\left[\boldsymbol{\epsilon}_{\boldsymbol{\theta}}\left(\boldsymbol{A}_{t_{j-1}}^{-1}\hat{\mathbf{z}}_{t_{j-1}}, t_{j-1}\right) - \boldsymbol{\epsilon}_{\boldsymbol{\theta}}\left(\boldsymbol{A}_{t_j}^{-1}\hat{\mathbf{z}}_{t_j}, t_j\right)\right] \tag{59}$$

$$= \left(\frac{s - t_{j-1}}{t_j - t_{j-1}}\right)\boldsymbol{\epsilon}_{\boldsymbol{\theta}}\left(\boldsymbol{A}_{t_j}^{-1}\hat{\mathbf{z}}_{t_j}, t_j\right) + \left(\frac{s - t_j}{t_{j-1} - t_j}\right)\boldsymbol{\epsilon}_{\boldsymbol{\theta}}\left(\boldsymbol{A}_{t_{j-1}}^{-1}\hat{\mathbf{z}}_{t_{j-1}}, t_{j-1}\right) \tag{60}$$

In the general case, the polynomial $P_r(s)$ can be compactly represented as,

$$P_r(s) = \sum_{k=0}^{r} \boldsymbol{C}_k(s)\boldsymbol{\epsilon_\theta}\left(\boldsymbol{A}_{t_{j-k}}^{-1}\hat{\mathbf{z}}_{t_{j-k}}, t_{j-k}\right), \qquad \boldsymbol{C}_k(s) = \prod_{l\neq k}^{r}\left[\frac{s-t_{j-l}}{t_{j-k}-t_{j-l}}\right] \tag{61}$$

Therefore, replacing the integrand $\boldsymbol{\epsilon_\theta}\left(\boldsymbol{A}_s^{-1}\hat{\mathbf{z}}_s, s\right)$ by its polynomial approximation $P_r(s)$, we have:

$$\hat{\mathbf{z}}_{t_i} = \hat{\mathbf{z}}_{t_j} + \int_{t_j}^{t_i} d\boldsymbol{\Phi}_s P_r(s) \tag{62}$$

$$\hat{\mathbf{z}}_{t_i} = \hat{\mathbf{z}}_{t_j} + \int_{t_j}^{t_i} d\boldsymbol{\Phi}_s \sum_{k=0}^{r} \boldsymbol{C}_k(s)\boldsymbol{\epsilon_\theta}\left(\boldsymbol{A}_{t_{j-k}}^{-1}\hat{\mathbf{z}}_{t_{j-k}}, t_{j-k}\right) \tag{63}$$

$$\hat{\mathbf{z}}_{t_i} = \hat{\mathbf{z}}_{t_j} + \sum_{k=0}^{r}\left[\int_{t_j}^{t_i} d\boldsymbol{\Phi}_s \boldsymbol{C}_k(s)\right]\boldsymbol{\epsilon_\theta}\left(\boldsymbol{A}_{t_{j-k}}^{-1}\hat{\mathbf{z}}_{t_{j-k}}, t_{j-k}\right) \tag{64}$$

$$\hat{\mathbf{z}}_{t_i} = \hat{\mathbf{z}}_{t_j} + \sum_{k=0}^{r}\left[\int_{t_j}^{t_i} \frac{1}{2}\boldsymbol{A}_s\boldsymbol{G}_s\boldsymbol{G}_s^\top\boldsymbol{L}_s^{-\top}\boldsymbol{C}_k(s)ds\right]\boldsymbol{\epsilon_\theta}\left(\boldsymbol{A}_{t_{j-k}}^{-1}\hat{\mathbf{z}}_{t_{j-k}}, t_{j-k}\right) \tag{65}$$

$$\boldsymbol{A}_{t_i}\mathbf{z}_{t_i} = \boldsymbol{A}_{t_j}\mathbf{z}_{t_j} + \sum_{k=0}^{r}\left[\int_{t_j}^{t_i} \frac{1}{2}\boldsymbol{A}_s\boldsymbol{G}_s\boldsymbol{G}_s^\top\boldsymbol{L}_s^{-\top}\boldsymbol{C}_k(s)ds\right]\boldsymbol{\epsilon_\theta}\left(\mathbf{z}_{t_{j-k}}, t_{j-k}\right) \tag{66}$$

$$\mathbf{z}_{t_i} = \boldsymbol{A}_{t_i}^{-1}\boldsymbol{A}_{t_j}\mathbf{z}_{t_j} + \sum_{k=0}^{r}\left[\int_{t_j}^{t_i} \frac{1}{2}\boldsymbol{A}_{t_i}^{-1}\boldsymbol{A}_s\boldsymbol{G}_s\boldsymbol{G}_s^\top\boldsymbol{L}_s^{-\top}\boldsymbol{C}_k(s)ds\right]\boldsymbol{\epsilon_\theta}\left(\mathbf{z}_{t_{j-k}}, t_{j-k}\right) \tag{67}$$

$$\mathbf{z}_{t_i} = \boldsymbol{\psi}(t_i, t_j)\mathbf{z}_{t_j} + \sum_{k=0}^{r}\left[\int_{t_j}^{t_i} \frac{1}{2}\boldsymbol{\psi}(t_i, s)\boldsymbol{G}_s\boldsymbol{G}_s^\top\boldsymbol{L}_s^{-\top}\boldsymbol{C}_k(s)ds\right]\boldsymbol{\epsilon_\theta}\left(\mathbf{z}_{t_{j-k}}, t_{j-k}\right) \tag{68}$$

which is the required exponential integrator with polynomial extrapolation proposed in Zhang & Chen (2023). Therefore, applying Adams-Bashforth in the transformed ODE in Eqn. 46 corresponds to polynomial extrapolation in Zhang & Chen (2023).

### B.5 PROOF OF THEOREM 2

We restate the full statement of Theorem 2 here (with regularity conditions) as follows.

**Theorem.** Let $\mathcal{F}_t$ and $\mathcal{G}_t$ be the flow maps induced by the transformed ODE

$$\frac{d\hat{\mathbf{z}}_t}{dt} = \boldsymbol{A}_t\boldsymbol{B}_t\boldsymbol{A}_t^{-1}\hat{\mathbf{z}}_t + \frac{d\boldsymbol{\Phi}_t}{dt}\boldsymbol{\epsilon_\theta}\left(\boldsymbol{C}_{\text{in}}(t)\boldsymbol{A}_t^{-1}\hat{\mathbf{z}}_t, C_{\text{noise}}(t)\right) \tag{69}$$

and by the conjugate integrator defined as

$$\hat{\mathbf{z}}_{t-h} = \hat{\mathbf{z}}_t - h\boldsymbol{A}_t\boldsymbol{B}_t\boldsymbol{A}_t^{-1}\hat{\mathbf{z}}_t + (\boldsymbol{\Phi}_{t-h} - \boldsymbol{\Phi}_t)\boldsymbol{\epsilon_\theta}\left(\boldsymbol{C}_{\text{in}}(t)\boldsymbol{A}_t^{-1}\hat{\mathbf{z}}_t, C_{\text{noise}}(t)\right) \tag{70}$$

respectively. We define two points, $\hat{\mathbf{z}}(t)$ and $\hat{\mathbf{z}}_t$, sampled from $\mathcal{F}$ and $\mathcal{G}$ respectively at time $t$ such that $\|\hat{\mathbf{z}}(t) - \hat{\mathbf{z}}_t\| < \delta$ for some $\delta > 0$. Furthermore, let $\boldsymbol{U}\boldsymbol{\Lambda}\boldsymbol{U}^{-1}$ denote the eigendecomposition of the matrix $\frac{1}{2}\boldsymbol{G}_t\boldsymbol{G}_t^T\boldsymbol{C}_{\text{out}}(t)\frac{\partial\boldsymbol{\epsilon_\theta}(\boldsymbol{C}_{\text{in}}\mathbf{z}_t, t)}{\partial\mathbf{z}_t}$. The conjugate integrator defined in Eqn. 70 is *stable* if $|1 + h\tilde{\lambda}| \leq 1$, where $\tilde{\lambda}$ denotes the eigenvalues of the matrix $\hat{\Lambda} = \boldsymbol{\Lambda} - \boldsymbol{U}^{-1}\boldsymbol{B}_t\boldsymbol{U}$.

*Proof.* We denote the conjugate integrator numerical update defined in Eqn. 70 by $\mathcal{G}_h$. Therefore, for this integrator to be *stable*, we need to show that,

$$\|\mathcal{G}_h(\hat{\mathbf{z}}(t)) - \mathcal{G}_h(\hat{\mathbf{z}}_t)\| \leq \Delta, \qquad \Delta > 0 \tag{71}$$

i.e., two nearby solution trajectories should not diverge under the application of the numerical update in each step. Next, we compute $\mathcal{G}_h(\hat{\mathbf{z}}(t))$ as follows:

$$\mathcal{G}_h(\hat{\mathbf{z}}(t)) = \hat{\mathbf{z}}(t) - h\boldsymbol{A}_t\boldsymbol{B}_t\boldsymbol{A}_t^{-1}\hat{\mathbf{z}}(t) + (\boldsymbol{\Phi}_{t-h} - \boldsymbol{\Phi}_t)\boldsymbol{\epsilon_\theta}\left(\boldsymbol{C}_{\text{in}}(t)\boldsymbol{A}_t^{-1}\hat{\mathbf{z}}(t), C_{\text{noise}}(t)\right) \tag{72}$$

$$= \hat{\mathbf{z}}(t) - h\boldsymbol{A}_t\boldsymbol{B}_t\boldsymbol{A}_t^{-1}\hat{\mathbf{z}}(t) - h\frac{d\boldsymbol{\Phi}_t}{dt}\boldsymbol{\epsilon_\theta}\left(\boldsymbol{C}_{\text{in}}(t)\boldsymbol{A}_t^{-1}\hat{\mathbf{z}}(t), C_{\text{noise}}(t)\right) + \mathcal{O}(h^2) \tag{73}$$

where we have used the first-order taylor series approximation of $\Phi_{t-h}$ in the above equation. Substituting $\frac{d\Phi_t}{dt} = -\frac{1}{2}A_t G_t G_t^\top C_{\text{out}}(t)$ in the above equation and ignoring the higher order terms $\mathcal{O}(h^2)$, we get,

$$\mathcal{G}_h(\hat{\mathbf{z}}(t)) = \hat{\mathbf{z}}(t) - hA_t B_t A_t^{-1}\hat{\mathbf{z}}(t) + \frac{h}{2}A_t G_t G_t^\top C_{\text{out}}(t)\boldsymbol{\epsilon_\theta}\left(C_{\text{in}}(t)A_t^{-1}\hat{\mathbf{z}}(t), C_{\text{noise}}(t)\right) \quad (74)$$

$$= \hat{\mathbf{z}}(t) - hA_t B_t A_t^{-1}\hat{\mathbf{z}}(t) + \frac{h}{2}A_t G_t G_t^\top C_{\text{out}}(t)\boldsymbol{\epsilon_\theta}\left(C_{\text{in}}(t)\mathbf{z}(t), C_{\text{noise}}(t)\right) \quad (75)$$

Similarly, $\mathcal{G}_h(\hat{\mathbf{z}}_t)$ can be computed as follows:

$$\mathcal{G}_h(\hat{\mathbf{z}}_t) = \hat{\mathbf{z}}_t - hA_t B_t A_t^{-1}\hat{\mathbf{z}}_t + \frac{h}{2}A_t G_t G_t^\top C_{\text{out}}(t)\boldsymbol{\epsilon_\theta}\left(C_{\text{in}}(t)\mathbf{z}_t, C_{\text{noise}}(t)\right) \quad (76)$$

Therefore,

$$\mathcal{G}_h(\hat{\mathbf{z}}(t)) - \mathcal{G}_h(\hat{\mathbf{z}}_t) = [\hat{\mathbf{z}}(t) - \hat{\mathbf{z}}_t] - hA_t B_t A_t^{-1}[\hat{\mathbf{z}}(t) - \hat{\mathbf{z}}_t] + \quad (77)$$

$$\frac{h}{2}A_t G_t G_t^\top C_{\text{out}}(t)\left[\boldsymbol{\epsilon_\theta}\left(C_{\text{in}}(t)\mathbf{z}(t), C_{\text{noise}}(t)\right) - \boldsymbol{\epsilon_\theta}\left(C_{\text{in}}(t)\mathbf{z}_t, C_{\text{noise}}(t)\right)\right] \quad (78)$$

Approximating the term $\boldsymbol{\epsilon_\theta}\left(C_{\text{in}}(t)\mathbf{z}(t), C_{\text{noise}}(t)\right)$ using a first-order taylor series approximation around the point $\boldsymbol{\epsilon_\theta}\left(C_{\text{in}}(t)\mathbf{z}_t, C_{\text{noise}}(t)\right)$ as,

$$\boldsymbol{\epsilon_\theta}\left(C_{\text{in}}(t)\mathbf{z}(t), C_{\text{noise}}(t)\right) = \boldsymbol{\epsilon_\theta}\left(C_{\text{in}}(t)\mathbf{z}_t, C_{\text{noise}}(t)\right) + \nabla_{\mathbf{z}_t}\boldsymbol{\epsilon_\theta}\left(C_{\text{in}}(t)\mathbf{z}_t, C_{\text{noise}}(t)\right)[\mathbf{z}(t) - \mathbf{z}_t] \quad (79)$$

$$= \boldsymbol{\epsilon_\theta}\left(C_{\text{in}}(t)\mathbf{z}_t, C_{\text{noise}}(t)\right) + \nabla_{\mathbf{z}_t}\boldsymbol{\epsilon_\theta}\left(C_{\text{in}}(t)\mathbf{z}_t, C_{\text{noise}}(t)\right)A_t^{-1}[\hat{\mathbf{z}}(t) - \hat{\mathbf{z}}_t] \quad (80)$$

Substituting the first order approximation of $\boldsymbol{\epsilon_\theta}\left(C_{\text{in}}(t)\mathbf{z}(t), C_{\text{noise}}(t)\right)$ in Eqn. 78,

$$\mathcal{G}_h(\hat{\mathbf{z}}(t)) - \mathcal{G}_h(\hat{\mathbf{z}}_t) = \left[I + hR_t\right]\left[\hat{\mathbf{z}}(t) - \hat{\mathbf{z}}_t\right] \quad (81)$$

where we have defined,

$$R_t = \left[\frac{1}{2}A_t G_t G_t^\top C_{\text{out}}(t)\nabla_{\mathbf{z}_t}\boldsymbol{\epsilon_\theta}\left(C_{\text{in}}(t)\mathbf{z}_t, C_{\text{noise}}(t)\right)A_t^{-1} - A_t B_t A_t^{-1}\right] \quad (82)$$

Therefore,

$$\|\mathcal{G}_h(\hat{\mathbf{z}}(t)) - \mathcal{G}_h(\hat{\mathbf{z}}_t)\| = \|(I + hR_t)(\hat{\mathbf{z}}(t) - \hat{\mathbf{z}}_t)\| \quad (83)$$

$$\leq \|I + hR_t\|\|\hat{\mathbf{z}}(t) - \hat{\mathbf{z}}_t\| \quad (84)$$

Since $\|\hat{\mathbf{z}}(t) - \hat{\mathbf{z}}_t\| < \delta$, we need the growth factor $\|I + hR_t\|$ to be bounded, which implies,

$$\rho(I + hR_t) \leq 1 \quad (85)$$

where $\rho$ denotes the spectral radius of a diagonalizable matrix. Furthermore, let,

$$\frac{1}{2}G_t G_t^\top C_{\text{out}}(t)\nabla_{\mathbf{z}_t}\boldsymbol{\epsilon_\theta}\left(C_{\text{in}}(t)\mathbf{z}_t, C_{\text{noise}}(t)\right) = U\Lambda U^{-1} \quad (86)$$

Therefore, we can simplify $R_t$ as,

$$R_t = \left[\frac{1}{2}A_t G_t G_t^\top C_{\text{out}}(t)\nabla_{\mathbf{z}_t}\boldsymbol{\epsilon_\theta}\left(C_{\text{in}}(t)\mathbf{z}_t, C_{\text{noise}}(t)\right)A_t^{-1} - A_t B_t A_t^{-1}\right] \quad (87)$$

$$= A_t\left[\frac{1}{2}G_t G_t^\top C_{\text{out}}(t)\nabla_{\mathbf{z}_t}\boldsymbol{\epsilon_\theta}\left(C_{\text{in}}(t)\mathbf{z}_t, C_{\text{noise}}(t)\right) - B_t\right]A_t^{-1} \quad (88)$$

$$= A_t\left[U\Lambda U^{-1} - B_t\right]A_t^{-1} \quad (89)$$

$$= (A_t U)\underbrace{\left[\Lambda - U^{-1}B_t U\right]}_{=V\tilde{\Lambda}V^{-1}}(A_t U)^{-1} \quad (90)$$

$$= (A_t U V)\tilde{\Lambda}(A_t U V)^{-1} \quad (91)$$

Substituting this simplified expression for $R_t$ in Eqn. 85, it follows that,

$$|1 + h\tilde{\lambda}| \leq 1 \quad (92)$$

where $\tilde{\lambda}$ is an eigenvalue of the matrix $\Lambda - U^{-1}B_t U$ which concludes the proof. $\square$

As a special case, for $B_t = \lambda I_d$, we have $R_t = (A_t U)\left[\Lambda - \lambda I\right](A_t U)^{-1}$. In this case the condition for stability reduces to $|1 + h(\hat{\lambda} - \lambda)| \leq 1$ which concludes the proof for Corollary 1

---

**Algorithm 1** *Conjugate Integrators* (defined in Eqn. 8)

---

**Input:** Trajectory length T, Network function $\boldsymbol{\epsilon}_{\boldsymbol{\theta}}(\boldsymbol{C}_{\text{in}}\mathbf{z}_t, t)$, number of sampling steps $N$, a monotonically decreasing timestep discretization $\{t_i\}_{i=0}^{N}$ spanning the interval $(\epsilon, \text{T})$ and choice of $\boldsymbol{B}_t$.
**Output:** $\mathbf{z}_{\epsilon} = (\mathbf{x}_{\epsilon}, \mathbf{m}_{\epsilon})$

Compute $\{\boldsymbol{A}_{t_i}\}_{i=0}^{N}$ and $\{\boldsymbol{\Phi}_{t_i}\}_{i=0}^{N}$ as in Eqn. 6               ▷ Pre-compute coefficients
$\mathbf{z}_{t_0} \sim p(\mathbf{z}_T)$                     ▷ Draw initial samples from the generative prior
$\hat{\mathbf{z}}_{t_0} = \boldsymbol{A}_{t_0}\mathbf{z}_{t_0}$                               ▷ Transform
**for** $n = 0$ **to** $N - 1$ **do**
    $h = (t_{n+1} - t_n)$                            ▷ Time step differential
    $d\boldsymbol{\Phi}_t = (\boldsymbol{\Phi}_{t_{n+1}} - \boldsymbol{\Phi}_{t_n})$                      ▷ Phi differential
    $\hat{\mathbf{z}}_{t_{n+1}} \leftarrow \hat{\mathbf{z}}_{t_n} + h\boldsymbol{A}_{t_n}\boldsymbol{B}_{t_n}\boldsymbol{A}_{t_n}^{-1}\hat{\mathbf{z}}_{t_n} + d\boldsymbol{\Phi}_t\boldsymbol{\epsilon}_{\boldsymbol{\theta}}(\boldsymbol{C}_{\text{in}}(t_n)\boldsymbol{A}_{t_n}^{-1}\hat{\mathbf{z}}_{t_n}, \boldsymbol{C}_{\text{noise}}(t_n))$    ▷ Update
**end for**
$\mathbf{z}_{t_N} = \boldsymbol{A}_{t_N}^{-1}\hat{\mathbf{z}}_{t_N}$                              ▷ Project to original space

---

### B.6 CONJUGATE INTEGRATORS IN THE WILD

Here, we highlight some practical considerations when implementing Conjugate Integrators. We present a high-level algorithmic implementation for the conjugate integrator defined in Eqn. 8 in Algorithm 1. Next, we discuss several practical aspects, including the invertibility of the transformation $\boldsymbol{A}_t$ and computing the coefficients $\boldsymbol{A}_t$ and $\boldsymbol{\Phi}_t$ as specified in Eqn. 6.

**Invertibility of the transformation $\boldsymbol{A}_t$:** Since we need to transform back the diffusion ODE dynamics from the projected space $\hat{\mathbf{z}}_t$ to the original space $\mathbf{z}_t$, ensuring the invertibility of the transformation $\boldsymbol{A}_t$ is a crucial requirement of conjugate integrators. However, since the expression for $\boldsymbol{A}_t$ is composed of an integral over multiple terms in the matrix exponential, it is non-trivial to guarantee matrix inversion since the matrices $\boldsymbol{B}_t$ and $\boldsymbol{C}_{\text{skip}}$ are user-specified. An alternate choice could be to update the mapping $\boldsymbol{A}_t$ to $\boldsymbol{A}_t + \delta I$ where $\delta > 0$ is a small constant to ensure non-zero eigenvalues at any time t, thus ensuring invertibility. In this work, we set $\delta = 0$ for all experiments since we do not encounter any such instabilities during sampling.

**Computing the Coefficients $\boldsymbol{A}_t$ and $\boldsymbol{\Phi}_t$:** The coefficients $\boldsymbol{A}_t$ and $\boldsymbol{\Phi}_t$ are defined as:

$$\boldsymbol{A}_t = \exp\left(\int_0^t \boldsymbol{B}_s - \boldsymbol{F}_s + \frac{1}{2}\boldsymbol{G}_s\boldsymbol{G}_s^{\top}\boldsymbol{C}_{\text{skip}}(s)ds\right), \qquad \boldsymbol{\Phi}_t = -\int_0^t \frac{1}{2}\boldsymbol{A}_s\boldsymbol{G}_s\boldsymbol{G}_s^{\top}\boldsymbol{C}_{\text{out}}(s)ds \quad (93)$$

where $\exp(.)$ denotes the matrix exponential. For the score parameterization in PSLD (Eqn. 273), these coefficients can be simplified as,

$$\boldsymbol{A}_t = \exp\left(\int_0^t (\boldsymbol{B}_s - \boldsymbol{F}_s)\, ds\right), \qquad \boldsymbol{\Phi}_t = \int_0^t \frac{1}{2}\boldsymbol{A}_s\boldsymbol{G}_s\boldsymbol{G}_s^{\top}\boldsymbol{L}_s^{-\top}ds \qquad (94)$$

For $\lambda$-DDIM, the matrix $\boldsymbol{B}_t$ is time-independent. Similarly, for PSLD, the matrix $\boldsymbol{F}_t$ is also time-independent. Therefore, the coefficient $\boldsymbol{A}_t$ further simplifies to,

$$\boldsymbol{A}_t = \exp\left((\boldsymbol{B} - \boldsymbol{F})\, t\right) \qquad (95)$$

The above matrix exponential can be computed using standard scientific libraries like PyTorch (Paszke et al., 2019) or SciPy (Virtanen et al., 2020). Consequently, the coefficient $\boldsymbol{\Phi}_t$ reduces to the following form,

$$\boldsymbol{\Phi}_t = \int_0^t \frac{1}{2}\exp\left((\boldsymbol{B} - \boldsymbol{F})\, s\right)\boldsymbol{G}_s\boldsymbol{G}_s^{\top}\boldsymbol{L}_s^{-\top}ds \qquad (96)$$

Therefore, at any time $t$, we estimate the coefficient $\boldsymbol{\Phi}_t$ using numerical integration. For a given timestep schedule $\{t_i\}$ during sampling, we precompute the coefficient $\boldsymbol{\Phi}_t$, which can be shared between all generated samples. For numerical integration, we use the `odeint` method from the `torchdiffeq` package (Chen, 2018) with parameters `atol=1e-5`, `rtol=1e-5` and the `RK45` solver (Dormand & Prince, 1980). We set $\boldsymbol{\Phi}_0 = \boldsymbol{0}$ as an initial condition. This is because, for the VP-SDE, $\boldsymbol{\Phi}_t$ corresponds to the noise-to-signal ratio at time $t$. Since we recover the data at time $t = 0$, the noise-to-signal ratio drops to zero. We extend this intuition to multivariate diffusions like PSLD and find this initial condition to work well in practice.

**Time Required for Computing coefficients $\boldsymbol{\Phi}_t$:** Given a set of sampling timepoints $\{t_i\}$, since $\boldsymbol{\Phi}_{t_i}$ is shared between all samples, we only need to compute $\{\boldsymbol{\Phi}_{t_i}\}$ once at the start of sampling. Empirically, for our largest budget of NFE=100 in this work, numerical integration for computing coefficients $\boldsymbol{\Phi}_t$ takes around 20 seconds on our setup, which is very cheap when amortized over a large number of generated samples.

## C  SPLITTING INTEGRATORS FOR FAST ODE/SDE SAMPLING

### C.1  INTRODUCTION TO SPLITTING INTEGRATORS

Here we provide a brief introduction to splitting integrators. For a detailed account of splitting integrators for designing symplectic numerical methods, we refer interested readers to Leimkuhler (2015). As discussed in the main text, the main idea behind splitting integrators is to split the vector field of an ODE or the drift and the diffusion components of an SDE into independent sub-components, which are then solved independently using a numerical scheme (or analytically). The solutions to independent sub-components are then composed in a specific order to obtain the final solution. Thus, three key steps in designing a splitting integrator are **split**, **solve**, and **compose**. We illustrate these steps with an example of a deterministic dynamical system. However, the concept is generic and can be applied to systems with stochastic dynamics as well.

Consider a dynamical system specified by the following ODE:

$$\begin{pmatrix} d\mathbf{x}_t \\ d\mathbf{m}_t \end{pmatrix} = \begin{pmatrix} \boldsymbol{f}(\mathbf{x}_t, \mathbf{m}_t) \\ \boldsymbol{g}(\mathbf{x}_t, \mathbf{m}_t) \end{pmatrix} dt \tag{97}$$

We start by choosing a scheme to split the vector field for the ODE in Eqn. 97. While different types of splitting schemes can be possible, we choose the following scheme for this example,

$$\begin{pmatrix} d\mathbf{x}_t \\ d\mathbf{m}_t \end{pmatrix} = \underbrace{\begin{pmatrix} \boldsymbol{f}(\mathbf{x}_t, \mathbf{m}_t) \\ 0 \end{pmatrix} dt}_{A} + \underbrace{\begin{pmatrix} 0 \\ \boldsymbol{g}(\mathbf{x}_t, \mathbf{m}_t) \end{pmatrix} dt}_{B} \tag{98}$$

where we denote the individual components by $A$ and $B$. Next, we solve each of these components independently, i.e., we compute solutions for the following ODEs independently.

$$\begin{pmatrix} d\mathbf{x}_t \\ d\mathbf{m}_t \end{pmatrix} = \begin{pmatrix} \boldsymbol{f}(\mathbf{x}_t, \mathbf{m}_t) \\ 0 \end{pmatrix} dt, \qquad \begin{pmatrix} d\mathbf{x}_t \\ d\mathbf{m}_t \end{pmatrix} = \begin{pmatrix} 0 \\ \boldsymbol{g}(\mathbf{x}_t, \mathbf{m}_t) \end{pmatrix} dt \tag{99}$$

While any numerical scheme can be used to approximate the solution for the splitting components, we use Euler throughout this work. Therefore, applying an Euler approximation, with a step size $h$, to each of these splitting components yields the solutions $\mathcal{L}_h^A$ and $\mathcal{L}_h^B$, as follows,

$$\mathcal{L}_h^A = \begin{cases} \mathbf{x}_{t+h} = \mathbf{x}_t + h\boldsymbol{f}(\mathbf{x}_t, \mathbf{m}_t) \\ \mathbf{m}_{t+h} = \mathbf{m}_t \end{cases}, \quad \mathcal{L}_h^B = \begin{cases} \mathbf{x}_{t+h} = \mathbf{x}_t \\ \mathbf{m}_{t+h} = \mathbf{m}_t + h\boldsymbol{g}(\mathbf{x}_t, \mathbf{m}_t) \end{cases} \tag{100}$$

In the final step, we compose the solutions to the independent components in a specific order. For instance, for the composition scheme AB, the final solution $\mathcal{L}_h^{[AB]} = \mathcal{L}_h^B \circ \mathcal{L}_h^A$. Therefore,

$$\mathcal{L}_h^{[AB]} = \begin{cases} \mathbf{x}_{t+h} = \mathbf{x}_t + h\boldsymbol{f}(\mathbf{x}_t, \mathbf{m}_t) \\ \mathbf{m}_{t+h} = \mathbf{m}_t + h\boldsymbol{g}(\mathbf{x}_{t+h}, \mathbf{m}_t) \end{cases} \tag{101}$$

is the required solution. It is worth noting that the final solution depends on the chosen composition scheme, and often it is not clear beforehand which composition scheme might work best.

### C.2  DETERMINISTIC SPLITTING INTEGRATORS

We split the Probability Flow ODE for PSLD using the following splitting scheme

$$\begin{pmatrix} d\bar{\mathbf{x}}_t \\ d\bar{\mathbf{m}}_t \end{pmatrix} = \underbrace{\frac{\beta}{2} \begin{pmatrix} \Gamma\bar{\mathbf{x}}_t - M^{-1}\bar{\mathbf{m}}_t + \Gamma\boldsymbol{s}_\theta^x(\bar{\mathbf{z}}_t, T-t) \\ 0 \end{pmatrix} dt}_{A} + \underbrace{\frac{\beta}{2} \begin{pmatrix} 0 \\ \bar{\mathbf{x}}_t + \nu\bar{\mathbf{m}}_t + M\nu\boldsymbol{s}_\theta^m(\bar{\mathbf{z}}_t, T-t) \end{pmatrix} dt}_{B} \tag{102}$$

where $\bar{\mathbf{x}}_t = \mathbf{x}_{T-t}$, $\bar{\mathbf{m}}_t = \mathbf{m}_{T-t}$, $\boldsymbol{s}_\theta^x$ and $\boldsymbol{s}_\theta^m$ denote the score components in the data and momentum space, respectively. In this work, we approximate the numerical update for each split using a simple Euler-based update. Formally, we denote the Euler approximation for the splits $A$ and $B$ by $\mathcal{L}_A$ and $\mathcal{L}_B$, respectively. The corresponding numerical updates for $\mathcal{L}_A$ and $\mathcal{L}_B$ can be specified as:

$$\mathcal{L}_A : \begin{cases} \bar{\mathbf{x}}_{t+h} & = \bar{\mathbf{x}}_t + \frac{h\beta}{2}\Big[\Gamma\bar{\mathbf{x}}_t - M^{-1}\bar{\mathbf{m}}_t + \Gamma\boldsymbol{s}_\theta^x(\bar{\mathbf{x}}_t, \bar{\mathbf{m}}_t, T-t)\Big] \\ \bar{\mathbf{m}}_{t+h} & = \bar{\mathbf{m}}_t \end{cases} \tag{103}$$

$$\mathcal{L}_B : \begin{cases} \bar{\mathbf{x}}_{t+h} & = \bar{\mathbf{x}}_t \\ \bar{\mathbf{m}}_{t+h} & = \bar{\mathbf{m}}_t + \frac{h\beta}{2}\Big[\bar{\mathbf{x}}_t + \nu\bar{\mathbf{m}}_t + M\nu\boldsymbol{s}_\theta^m(\bar{\mathbf{x}}_t, \bar{\mathbf{m}}_t, T-t)\Big] \end{cases} \tag{104}$$

Next, we summarize the exact update equations for all deterministic splitting samplers proposed in this work.

### C.2.1   NAIVE SPLITTING SAMPLERS

We propose the following naive splitting samplers:

**Naive Symplectic Euler (NSE):** In this scheme, for a given step size h, the solutions to the splitting pieces $\mathcal{L}_h^A$ and $\mathcal{L}_h^B$ are composed as $\mathcal{L}_h^{[BA]} = \mathcal{L}_h^A \circ \mathcal{L}_h^B$. Consequently, one numerical update step for this integrator can be defined as,

$$\bar{\mathbf{m}}_{t+h} = \bar{\mathbf{m}}_t + \frac{h\beta}{2}\left[\bar{\mathbf{x}}_t + \nu\bar{\mathbf{m}}_t + M\nu\boldsymbol{s}_\theta^m(\bar{\mathbf{x}}_t, \bar{\mathbf{m}}_t, T-t)\right] \tag{105}$$

$$\bar{\mathbf{x}}_{t+h} = \bar{\mathbf{x}}_t + \frac{h\beta}{2}\left[\Gamma\bar{\mathbf{x}}_t - M^{-1}\bar{\mathbf{m}}_{t+h} + \Gamma\boldsymbol{s}_\theta^x(\bar{\mathbf{x}}_t, \bar{\mathbf{m}}_{t+h}, T-t)\right] \tag{106}$$

Therefore, one update step for the NVV sampler requires **two NFEs**.

**Naive Velocity Verlet (NVV):** In this scheme, for a given step size h, the solutions to the splitting pieces $\mathcal{L}_h^A$ and $\mathcal{L}_h^B$ are composed as $\mathcal{L}_h^{[BAB]} = \mathcal{L}_{h/2}^B \circ \mathcal{L}_h^A \circ \mathcal{L}_{h/2}^B$. Consequently, one numerical update step for this integrator can be defined as

$$\bar{\mathbf{m}}_{t+h/2} = \bar{\mathbf{m}}_t + \frac{h\beta}{4}\left[\bar{\mathbf{x}}_t + \nu\bar{\mathbf{m}}_t + M\nu\boldsymbol{s}_\theta^m(\bar{\mathbf{x}}_t, \bar{\mathbf{m}}_t, T-t)\right] \tag{107}$$

$$\bar{\mathbf{x}}_{t+h} = \bar{\mathbf{x}}_t + \frac{h\beta}{2}\left[\Gamma\bar{\mathbf{x}}_t - M^{-1}\bar{\mathbf{m}}_{t+h/2} + \Gamma\boldsymbol{s}_\theta^x(\bar{\mathbf{x}}_t, \bar{\mathbf{m}}_{t+h/2}, T-t)\right] \tag{108}$$

$$\bar{\mathbf{m}}_{t+h} = \bar{\mathbf{m}}_{t+h/2} + \frac{h\beta}{4}\left[\bar{\mathbf{x}}_{t+h} + \nu\bar{\mathbf{m}}_{t+h/2} + M\nu\boldsymbol{s}_\theta^m(\bar{\mathbf{x}}_{t+h}, \bar{\mathbf{m}}_{t+h/2}, T-t)\right] \tag{109}$$

Therefore, one update step for the NVV sampler requires **three NFEs**.

### C.2.2   REDUCED SPLITTING SAMPLERS

Analogous to the NSE and NVV samplers, we propose the Reduced Symplectic Euler (RSE) and the Reduced Velocity Verlet (RVV) samplers, respectively.

**Reduced Symplectic Euler (RSE):** The numerical updates for this scheme are as follows (the terms in red denote the changes from the NSE scheme),

$$\bar{\mathbf{m}}_{t+h} = \bar{\mathbf{m}}_t + \frac{h\beta}{2}\left[\bar{\mathbf{x}}_t + \nu\bar{\mathbf{m}}_t + M\nu\boldsymbol{s}_\theta^m(\bar{\mathbf{x}}_t, \bar{\mathbf{m}}_t, T-t)\right] \tag{110}$$

$$\bar{\mathbf{x}}_{t+h} = \bar{\mathbf{x}}_t + \frac{h\beta}{2}\left[\Gamma\bar{\mathbf{x}}_t - M^{-1}\bar{\mathbf{m}}_{t+h} + \Gamma\boldsymbol{s}_\theta^x(\bar{\mathbf{x}}_t, \bar{\mathbf{m}}_t, T-t)\right] \tag{111}$$

It is worth noting that the RSE sampler requires only **one NFE** per update step since a single score evaluation is re-used in both the momentum and the position updates.

**Reduced Velocity Verlet (RVV)**: The numerical updates for this scheme are as follows (the terms in blue denote the changes from the NVV scheme),

$$\bar{\mathbf{m}}_{t+h/2} = \bar{\mathbf{m}}_t + \frac{h\beta}{4} \left[ \bar{\mathbf{x}}_t + \nu \bar{\mathbf{m}}_t + M\nu \boldsymbol{s}_\theta^m(\bar{\mathbf{x}}_t, \bar{\mathbf{m}}_t, T-t) \right] \tag{112}$$

$$\bar{\mathbf{x}}_{t+h} = \bar{\mathbf{x}}_t + \frac{h\beta}{2} \left[ \Gamma\bar{\mathbf{x}}_t - M^{-1}\bar{\mathbf{m}}_{t+h/2} + \Gamma\boldsymbol{s}_\theta^x(\bar{\mathbf{x}}_t, \bar{\mathbf{m}}_t, T-t) \right] \tag{113}$$

$$\bar{\mathbf{m}}_{t+h} = \bar{\mathbf{m}}_{t+h/2} + \frac{h\beta}{4} \left[ \bar{\mathbf{x}}_{t+h} + \nu \bar{\mathbf{m}}_{t+h/2} + M\nu \boldsymbol{s}_\theta^m(\bar{\mathbf{x}}_{t+h}, \bar{\mathbf{m}}_{t+h/2}, T-(t+h)) \right] \tag{114}$$

In contrast to the NVV sampler, the RVV sampler requires **two** NFEs per update step. It is worth noting that the reduced schemes require fewer NFEs per update step than their naive counterparts. This implies that for the same compute budget, the reduced schemes use smaller step sizes as compared to the naive schemes. This is one of the reasons for the empirical effectiveness of the reduced schemes as compared to their naive counterparts. Next, we discuss the effectiveness of the reduced samplers from the lens of local error analysis.

### C.2.3 LOCAL ERROR ANALYSIS FOR DETERMINISTIC SPLITTING INTEGRATORS

We now analyze the naive and reduced splitting samplers proposed in this work from the lens of local error analysis for ODE solvers. The probability flow ODE for PSLD is defined as,

$$\begin{pmatrix} d\bar{\mathbf{x}}_t \\ d\bar{\mathbf{m}}_t \end{pmatrix} = \frac{\beta}{2} \begin{pmatrix} \Gamma\bar{\mathbf{x}}_t - M^{-1}\bar{\mathbf{m}}_t + \Gamma\boldsymbol{s}_\theta^x(\bar{\mathbf{z}}_t, T-t) \\ \bar{\mathbf{x}}_t + \nu\bar{\mathbf{m}}_t + M\nu\boldsymbol{s}_\theta^m(\bar{\mathbf{z}}_t, T-t) \end{pmatrix} dt, \qquad t \in [0, T] \tag{115}$$

We denote the proposed numerical schemes by $\mathcal{G}_h$ and the underlying ground-truth flow map for the probability flow ODE as $\mathcal{F}_h$ where $h > 0$ is the step-size for numerical integration. Formally, we analyze the growth of $\bar{e}_{t+h} = e_{T-(t+h)} = \|\bar{\mathbf{z}}(t+h) - \bar{\mathbf{z}}_{t+h}\|$ where $\bar{\mathbf{z}}_{t+h} = \mathbf{z}_{T-(t+h)} = \mathcal{G}_h(\bar{\mathbf{z}}_t)$ and $\bar{\mathbf{z}}(t+h) = \mathbf{z}_{T-(t+h)}\mathcal{F}_h(\bar{\mathbf{z}}(t))$ are the approximated and ground-truth solutions at time $T - (t+h)$. Furthermore,

$$\bar{e}_{t+h} = \|\mathcal{F}_h(\bar{\mathbf{z}}(t)) - \mathcal{G}_h(\bar{\mathbf{z}}_t)\| \tag{116}$$

$$= \|\mathcal{F}_h(\bar{\mathbf{z}}(t)) - \mathcal{G}_h(\bar{\mathbf{z}}(t)) + \mathcal{G}_h(\bar{\mathbf{z}}(t)) - \mathcal{G}_h(\bar{\mathbf{z}}_t)\| \tag{117}$$

$$\leq \|\mathcal{F}_h(\bar{\mathbf{z}}(t)) - \mathcal{G}_h(\bar{\mathbf{z}}(t))\| + \|\mathcal{G}_h(\bar{\mathbf{z}}(t)) - \mathcal{G}_h(\bar{\mathbf{z}}_t)\| \tag{118}$$

The first term on the right-hand side of the above error bound is referred to as the *local truncation error*. Intuitively, it gives an estimate of how much error is introduced by our numerical scheme given the ground truth solution till the previous time step $t$. The second term in the error bound is referred to as the *stability* of the numerical scheme. Intuitively, it gives an estimate of how much divergence is introduced by our numerical scheme given two nearby solution trajectories such that $\|\mathbf{z}(t) - \mathbf{z}_t\| < \delta$. Here, we only deal with the local truncation error in the position and the momentum space. To this end, we first compute the term $\mathcal{F}_h(\mathbf{z}(t))$ using the Taylor-series expansion.

**Computation of $\mathcal{F}_h(\mathbf{z}(t))$**: Using the Taylor-series expansion in the position space, we have,

$$\bar{\mathbf{x}}(t+h) = \bar{\mathbf{x}}(t) + h\frac{d\bar{\mathbf{x}}(t)}{dt} + \frac{h^2}{2}\frac{d^2\bar{\mathbf{x}}(t)}{dt^2} + \mathcal{O}(h^3) \tag{119}$$

$$\bar{\mathbf{m}}(t+h) = \bar{\mathbf{m}}(t) + h\frac{d\bar{\mathbf{m}}(t)}{dt} + \frac{h^2}{2}\frac{d^2\bar{\mathbf{m}}(t)}{dt^2} + \mathcal{O}(h^3) \tag{120}$$

Substituting the values of $\frac{d\bar{\mathbf{x}}(t)}{dt}$ and $\frac{d\bar{\mathbf{m}}(t)}{dt}$ from the PSLD Prob. Flow ODE, it follows that,

$$\mathcal{F}_h(\bar{\mathbf{x}}(t)) = \bar{\mathbf{x}}(t) + \frac{h\beta}{2} \left[ \Gamma\bar{\mathbf{x}}(t) - M^{-1}\bar{\mathbf{m}}(t) + \Gamma\boldsymbol{s}_\theta^x(\bar{\mathbf{z}}(t), T-t) \right] + \tag{121}$$

$$\frac{h^2\beta}{4}\frac{d}{dt} \left[ \Gamma\bar{\mathbf{x}}(t) - M^{-1}\bar{\mathbf{m}}(t) + \Gamma\boldsymbol{s}_\theta^x(\bar{\mathbf{z}}(t), T-t) \right] + \mathcal{O}(h^3) \tag{122}$$

$$\mathcal{F}_h(\bar{\mathbf{m}}(t)) = \bar{\mathbf{m}}(t) + \frac{h\beta}{2} \left[ \bar{\mathbf{x}}(t) + \nu\bar{\mathbf{m}}(t) + M\nu\boldsymbol{s}_\theta^m(\bar{\mathbf{z}}(t), T-t) \right] + \tag{123}$$

$$\frac{h^2\beta}{4}\frac{d}{dt} \left[ \bar{\mathbf{x}}(t) + \nu\bar{\mathbf{m}}(t) + M\nu\boldsymbol{s}_\theta^m(\bar{\mathbf{z}}(t), T-t) \right] + \mathcal{O}(h^3) \tag{124}$$

Next, we analyze the local error for the Naive and Reduced Velocity Verlet samplers while highlighting the justification for the difference in the update rules between the naive and the reduced schemes.

### C.2.4 ERROR ANALYSIS: NAIVE VELOCITY VERLET (NVV)

The NVV sampler has the following update rules:

$$\bar{\mathbf{m}}_{t+h/2} = \bar{\mathbf{m}}_t + \frac{h\beta}{4}\left[\bar{\mathbf{x}}_t + \nu\bar{\mathbf{m}}_t + M\nu s_\theta^m(\bar{\mathbf{x}}_t, \bar{\mathbf{m}}_t, T-t)\right] \tag{125}$$

$$\bar{\mathbf{x}}_{t+h} = \bar{\mathbf{x}}_t + \frac{h\beta}{2}\left[\Gamma\bar{\mathbf{x}}_t - M^{-1}\bar{\mathbf{m}}_{t+h/2} + \Gamma s_\theta^x(\bar{\mathbf{x}}_t, \bar{\mathbf{m}}_{t+h/2}, T-t)\right] \tag{126}$$

$$\bar{\mathbf{m}}_{t+h} = \bar{\mathbf{m}}_{t+h/2} + \frac{h\beta}{4}\left[\bar{\mathbf{x}}_{t+h} + \nu\bar{\mathbf{m}}_{t+h/2} + M\nu s_\theta^m(\bar{\mathbf{x}}_{t+h}, \bar{\mathbf{m}}_{t+h/2}, T-t)\right] \tag{127}$$

We first compute the local truncation error for the NVV sampler in both the position and the momentum space.

**NVV local truncation error in the position space:** From the update equations,

$$\bar{\mathbf{x}}(t+h) = \bar{\mathbf{x}}(t) + \frac{h\beta}{2}\left[\Gamma\bar{\mathbf{x}}(t) - M^{-1}\bar{\mathbf{m}}(t+h/2) + \Gamma s_\theta^x(\bar{\mathbf{x}}(t), \bar{\mathbf{m}}(t+h/2), T-t)\right] \tag{128}$$

$$= \bar{\mathbf{x}}(t) + \frac{h\beta}{2}\left[\Gamma\bar{\mathbf{x}}(t) - M^{-1}\left(\bar{\mathbf{m}}(t) + \frac{h\beta}{4}\left[\bar{\mathbf{x}}(t) + \nu\bar{\mathbf{m}}(t) + M\nu s_\theta^m(\bar{\mathbf{x}}(t), \bar{\mathbf{m}}(t), T-t)\right]\right)\right. \tag{129}$$

$$\left. + \Gamma s_\theta^x(\bar{\mathbf{x}}(t), \bar{\mathbf{m}}(t+h/2), T-t)\right] \tag{130}$$

$$\mathcal{G}_h(\bar{\mathbf{x}}(t)) = \bar{\mathbf{x}}(t) + \frac{h\beta}{2}\left[\Gamma\bar{\mathbf{x}}(t) - M^{-1}\bar{\mathbf{m}}(t) + \Gamma s_\theta^x(\bar{\mathbf{x}}(t), \bar{\mathbf{m}}(t+h/2), T-t)\right] - \tag{131}$$

$$\frac{h^2\beta^2 M^{-1}}{8}\left[\bar{\mathbf{x}}(t) + \nu\bar{\mathbf{m}}(t) + M\nu s_\theta^m(\bar{\mathbf{x}}(t), \bar{\mathbf{m}}(t), T-t)\right] \tag{132}$$

$$\mathcal{G}_h(\bar{\mathbf{x}}(t)) = \bar{\mathbf{x}}(t) + \frac{h\beta}{2}\left[\Gamma\bar{\mathbf{x}}(t) - M^{-1}\bar{\mathbf{m}}(t) + \Gamma s_\theta^x(\bar{\mathbf{x}}(t), \bar{\mathbf{m}}(t+h/2), T-t)\right] - \frac{h^2\beta M^{-1}}{4}\frac{d\bar{\mathbf{m}}(t)}{dt} \tag{133}$$

Therefore, the local truncation error in the position space is given by,

$$\mathcal{F}_h(\bar{\mathbf{x}}(t)) - \mathcal{G}_h(\bar{\mathbf{x}}(t)) = \frac{h\beta\Gamma}{2}\left[s_\theta^x(\bar{\mathbf{x}}(t), \bar{\mathbf{m}}(t), T-t) - s_\theta^x(\bar{\mathbf{x}}(t), \bar{\mathbf{m}}(t+h/2), T-t)\right] + \tag{134}$$

$$\frac{h^2\beta\Gamma}{4}\frac{d}{dt}\left[\bar{\mathbf{x}}(t) + s_\theta^x(\bar{\mathbf{x}}(t), \bar{\mathbf{m}}(t), T-t)\right] \tag{135}$$

We can approximate the term $s_\theta^x(\bar{\mathbf{x}}(t), \bar{\mathbf{m}}(t+h/2), T-t)$ using the Taylor-series expansion as follows,

$$s_\theta^x(\bar{\mathbf{x}}(t), \bar{\mathbf{m}}(t+h/2), T-t) = s_\theta^x(\bar{\mathbf{x}}(t), \bar{\mathbf{m}}(t), T-t) + \frac{\partial s_\theta^x(\bar{\mathbf{x}}(t), \bar{\mathbf{m}}(t), T-t)}{\partial \bar{\mathbf{m}}(t)} \tag{136}$$

$$\left[\bar{\mathbf{m}}(t+h/2) - \bar{\mathbf{m}}(t))\right] + \mathcal{O}(h^2) \tag{137}$$

$$= s_\theta^x(\bar{\mathbf{x}}(t), \bar{\mathbf{m}}(t), T-t) + \frac{\partial s_\theta^x(\bar{\mathbf{x}}(t), \bar{\mathbf{m}}(t), T-t)}{\partial \bar{\mathbf{m}}(t)} \tag{138}$$

$$\left[\frac{h\beta}{4}\left(\bar{\mathbf{x}}(t) + \nu\bar{\mathbf{m}}(t) + M\nu s_\theta^m(\bar{\mathbf{x}}(t), \bar{\mathbf{m}}(t), T-t)\right)\right] + \mathcal{O}(h^2) \tag{139}$$

$$s_\theta^x(\bar{\mathbf{x}}(t), \bar{\mathbf{m}}(t), T-t) - s_\theta^x(\bar{\mathbf{x}}(t), \bar{\mathbf{m}}(t+h/2), T-t) = -\frac{h}{2}\frac{\partial s_\theta^x(\bar{\mathbf{x}}(t), \bar{\mathbf{m}}(t), T-t)}{\partial \bar{\mathbf{m}}(t)}\frac{d\bar{\mathbf{m}}_t}{dt} + \mathcal{O}(h^2) \tag{140}$$

Substituting the above approximation (while ignoring the higher-order terms $\mathcal{O}(h^2)$) in Eqn. 135,

$$\mathcal{F}_h(\bar{\mathbf{x}}(t)) - \mathcal{G}_h(\bar{\mathbf{x}}(t)) = -\frac{h^2\beta\Gamma}{4}\left[\frac{\partial \boldsymbol{s}_\theta^x(\bar{\mathbf{x}}(t), \bar{\mathbf{m}}(t), T-t)}{\partial \bar{\mathbf{m}}(t)}\frac{d\bar{\mathbf{m}}_t}{dt}\right] + \tag{141}$$

$$\frac{h^2\beta\Gamma}{4}\frac{d}{dt}\left[\bar{\mathbf{x}}(t) + \boldsymbol{s}_\theta^x(\bar{\mathbf{x}}(t), \bar{\mathbf{m}}(t), T-t)\right] \tag{142}$$

$$= \frac{h^2\beta\Gamma}{4}\left[\frac{d}{dt}\left(\bar{\mathbf{x}}(t) + \boldsymbol{s}_\theta^x(\bar{\mathbf{x}}(t), \bar{\mathbf{m}}(t), T-t)\right) - \frac{\partial \boldsymbol{s}_\theta^x(\bar{\mathbf{x}}(t), \bar{\mathbf{m}}(t), T-t)}{\partial \bar{\mathbf{m}}(t)}\frac{d\bar{\mathbf{m}}_t}{dt}\right] \tag{143}$$

$$= \frac{h^2\beta\Gamma}{4}\left[\frac{d\bar{\mathbf{x}}(t)}{dt} + \left(\frac{d\boldsymbol{s}_\theta^x(\bar{\mathbf{x}}(t), \bar{\mathbf{m}}(t), T-t)}{dt} - \frac{\partial \boldsymbol{s}_\theta^x(\bar{\mathbf{x}}(t), \bar{\mathbf{m}}(t), T-t)}{\partial \bar{\mathbf{m}}(t)}\frac{d\bar{\mathbf{m}}_t}{dt}\right)\right] \tag{144}$$

From the Chain rule, we have the following result,

$$\frac{d\boldsymbol{s}_\theta^x(\bar{\mathbf{x}}(t), \bar{\mathbf{m}}(t), T-t)}{dt} = \frac{\partial \boldsymbol{s}_\theta^x(\bar{\mathbf{x}}(t), \bar{\mathbf{m}}(t), T-t)}{\partial t} + \frac{\partial \boldsymbol{s}_\theta^x(\bar{\mathbf{x}}(t), \bar{\mathbf{m}}(t), T-t)}{\partial \bar{\mathbf{x}}_t}\frac{d\bar{\mathbf{x}}_t}{dt} + \tag{145}$$

$$\frac{\partial \boldsymbol{s}_\theta^x(\bar{\mathbf{x}}(t), \bar{\mathbf{m}}(t), T-t)}{\partial \bar{\mathbf{m}}_t}\frac{d\bar{\mathbf{m}}_t}{dt} \tag{146}$$

Substituting the above result in Eqn. 144,

$$\mathcal{F}_h(\bar{\mathbf{x}}(t)) - \mathcal{G}_h(\bar{\mathbf{x}}(t)) = \frac{h^2\beta\Gamma}{4}\left[\frac{d\bar{\mathbf{x}}(t)}{dt} + \left(\frac{\partial \boldsymbol{s}_\theta^x(\bar{\mathbf{x}}(t), \bar{\mathbf{m}}(t), T-t)}{\partial \bar{\mathbf{x}}(t)}\frac{d\bar{\mathbf{x}}_t}{dt} + \frac{\partial \boldsymbol{s}_\theta^x(\bar{\mathbf{x}}(t), \bar{\mathbf{m}}(t), T-t)}{\partial t}\right)\right] \tag{147}$$

The above equation implies that,

$$\boxed{\|\mathcal{F}_h(\bar{\mathbf{x}}(t)) - \mathcal{G}_h(\bar{\mathbf{x}}(t))\| \leq \frac{C\beta\Gamma h^2}{4}} \tag{148}$$

Since we choose $\beta = 8$ throughout this work, $\beta/4 = 2$ can be absorbed in the constant $C$. Therefore, the local truncation error for the Naive Velocity Verlet (NVV) is of the order of $\mathcal{O}(\Gamma h^2)$. Since $\Gamma$ is usually small in PSLD (Pandey & Mandt, 2023) (for instance, 0.01 for CIFAR-10 and 0.005 for CelebA-64), its magnitude is comparable or less than $h$ (particularly in the low NFE regime). Therefore, the effective local truncation order for the NVV scheme is of the order of $\mathcal{O}(h^3)$.

Next, we analyze the local truncation error for NVV in the momentum space.

**NVV local truncation error in the momentum space:** From the update equations,

$$\bar{\mathbf{m}}(t+h) = \bar{\mathbf{m}}(t+h/2) + \frac{h\beta}{4}\left[\bar{\mathbf{x}}(t+h) + \nu\bar{\mathbf{m}}(t+h/2) + M\nu\boldsymbol{s}_\theta^m(\bar{\mathbf{x}}(t+h), \bar{\mathbf{m}}(t+h/2), T-t)\right] \tag{149}$$

$$\bar{\mathbf{m}}(t+h) = \bar{\mathbf{m}}(t) + \frac{h\beta}{4}\left[\bar{\mathbf{x}}(t) + \nu\bar{\mathbf{m}}(t) + M\nu\boldsymbol{s}_\theta^m(\bar{\mathbf{x}}(t), \bar{\mathbf{m}}(t), T-t)\right] + \frac{h\beta}{4}\left[\bar{\mathbf{x}}(t+h)\right] + \tag{150}$$

$$\frac{h\beta\nu}{4}\left[\bar{\mathbf{m}}(t+h/2)\right] + \frac{h\beta M\nu}{4}\boldsymbol{s}_\theta^m(\bar{\mathbf{x}}(t+h), \bar{\mathbf{m}}(t+h/2), T-t) \tag{151}$$

$$\bar{\mathbf{m}}(t+h) = \bar{\mathbf{m}}(t) + \frac{h\beta}{4}\left[\bar{\mathbf{x}}(t) + \nu\bar{\mathbf{m}}(t) + M\nu\boldsymbol{s}_\theta^m(\bar{\mathbf{x}}(t), \bar{\mathbf{m}}(t), T-t)\right] + \tag{152}$$

$$\frac{h\beta}{4}\left[\bar{\mathbf{x}}(t) + \frac{h\beta}{2}\left[\Gamma\bar{\mathbf{x}}(t) - M^{-1}\bar{\mathbf{m}}(t+h/2) + \Gamma\boldsymbol{s}_\theta^x(\bar{\mathbf{x}}(t), \bar{\mathbf{m}}(t+h/2), T-t)\right]\right] + \tag{153}$$

$$\frac{h\beta\nu}{4}\left[\bar{\mathbf{m}}(t) + \frac{h\beta}{4}\left[\bar{\mathbf{x}}(t) + \nu\bar{\mathbf{m}}(t) + M\nu\boldsymbol{s}_\theta^m(\bar{\mathbf{x}}(t), \bar{\mathbf{m}}(t), T-t)\right]\right] + \tag{154}$$

$$\frac{h\beta M\nu}{4}\boldsymbol{s}_\theta^m(\bar{\mathbf{x}}(t+h), \bar{\mathbf{m}}(t+h/2), T-t) \tag{155}$$

$$\bar{\mathbf{m}}(t+h) = \bar{\mathbf{m}}(t) + \frac{h\beta}{2}\left[\bar{\mathbf{x}}(t) + \nu\bar{\mathbf{m}}(t) + M\nu\boldsymbol{s}_\theta^m(\bar{\mathbf{x}}(t),\bar{\mathbf{m}}(t),T-t)\right] + \tag{156}$$

$$\frac{h^2\beta}{4}\left[\frac{\beta}{2}\left[\Gamma\bar{\mathbf{x}}(t) - M^{-1}\bar{\mathbf{m}}(t+h/2) + \Gamma\boldsymbol{s}_\theta^x(\bar{\mathbf{x}}(t),\bar{\mathbf{m}}(t+h/2),T-t)\right]\right] + \tag{157}$$

$$\frac{h^2\beta\nu}{8}\underbrace{\left[\frac{\beta}{2}\left[\bar{\mathbf{x}}(t) + \nu\bar{\mathbf{m}}(t) + M\nu\boldsymbol{s}_\theta^m(\bar{\mathbf{x}}(t),\bar{\mathbf{m}}(t),T-t)\right]\right]}_{=\frac{d\bar{\mathbf{m}}(t)}{dt}} + \tag{158}$$

$$\frac{h\beta M\nu}{4}\left[\boldsymbol{s}_\theta^m(\bar{\mathbf{x}}(t+h),\bar{\mathbf{m}}(t+h/2),T-t) - \boldsymbol{s}_\theta^m(\bar{\mathbf{x}}(t),\bar{\mathbf{m}}(t),T-t)\right] \tag{159}$$

$$\bar{\mathbf{m}}(t+h) = \bar{\mathbf{m}}(t) + \frac{h\beta}{2}\left[\bar{\mathbf{x}}(t) + \nu\bar{\mathbf{m}}(t) + M\nu\boldsymbol{s}_\theta^m(\bar{\mathbf{x}}(t),\bar{\mathbf{m}}(t),T-t)\right] + \tag{160}$$

$$\frac{h^2\beta}{4}\left[\frac{\beta}{2}\left(\Gamma\bar{\mathbf{x}}(t) - M^{-1}\left(\bar{\mathbf{m}}(t) + \frac{h\beta}{4}\left[\bar{\mathbf{x}}(t) + \nu\bar{\mathbf{m}}(t) + M\nu\boldsymbol{s}_\theta^m(\bar{\mathbf{x}}(t),\bar{\mathbf{m}}(t),T-t)\right]\right)\right. \tag{161}$$

$$\left.\left. + \Gamma\boldsymbol{s}_\theta^x(\bar{\mathbf{x}}(t),\bar{\mathbf{m}}(t+h/2),T-t)\right)\right] + \frac{h^2\beta\nu}{8}\frac{d\bar{\mathbf{m}}(t)}{dt} + \tag{162}$$

$$\frac{h\beta M\nu}{4}\left[\boldsymbol{s}_\theta^m(\bar{\mathbf{x}}(t+h),\bar{\mathbf{m}}(t+h/2),T-t) - \boldsymbol{s}_\theta^m(\bar{\mathbf{x}}(t),\bar{\mathbf{m}}(t),T-t)\right] \tag{163}$$

$$\bar{\mathbf{m}}(t+h) = \bar{\mathbf{m}}(t) + \frac{h\beta}{2}\left[\bar{\mathbf{x}}(t) + \nu\bar{\mathbf{m}}(t) + M\nu\boldsymbol{s}_\theta^m(\bar{\mathbf{x}}(t),\bar{\mathbf{m}}(t),T-t)\right] + \tag{164}$$

$$\frac{h^2\beta}{4}\left[\underbrace{\frac{\beta}{2}\left(\Gamma\bar{\mathbf{x}}(t) - M^{-1}\bar{\mathbf{m}}(t) + \Gamma\boldsymbol{s}_\theta^x(\bar{\mathbf{x}}(t),\bar{\mathbf{m}}(t),T-t)\right)}_{=\frac{d\bar{\mathbf{x}}_t}{dt}}\right] + \tag{165}$$

$$\frac{h^2\beta^2\Gamma}{8}\left[\boldsymbol{s}_\theta^x(\bar{\mathbf{x}}(t),\bar{\mathbf{m}}(t+h/2),T-t) - \boldsymbol{s}_\theta^x(\bar{\mathbf{x}}(t),\bar{\mathbf{m}}(t),T-t)\right] + \frac{h^2\beta\nu}{8}\frac{d\bar{\mathbf{m}}(t)}{dt} + \tag{166}$$

$$\frac{h\beta M\nu}{4}\left[\boldsymbol{s}_\theta^m(\bar{\mathbf{x}}(t+h),\bar{\mathbf{m}}(t+h/2),T-t) - \boldsymbol{s}_\theta^m(\bar{\mathbf{x}}(t),\bar{\mathbf{m}}(t),T-t)\right] + \mathcal{O}(h^3) \tag{167}$$

Approximating $\boldsymbol{s}_\theta^m(\bar{\mathbf{x}}(t+h),\bar{\mathbf{m}}(t+h/2),T-t)$ around $\boldsymbol{s}_\theta^m(\bar{\mathbf{x}}(t),\bar{\mathbf{m}}(t),T-t)$ using a first-order Taylor series,

$$\boldsymbol{s}_\theta^m(\bar{\mathbf{x}}(t+h),\bar{\mathbf{m}}(t+h/2),T-t) \approx \boldsymbol{s}_\theta^m(\bar{\mathbf{x}}(t),\bar{\mathbf{m}}(t),T-t) + \frac{\partial\boldsymbol{s}_\theta^m(\bar{\mathbf{x}}(t),\bar{\mathbf{m}}(t),T-t)}{\partial\bar{\mathbf{x}}(t)} \tag{168}$$

$$\left[\bar{\mathbf{x}}(t+h) - \bar{\mathbf{x}}(t)\right] + \frac{\partial\boldsymbol{s}_\theta^m(\bar{\mathbf{x}}(t),\bar{\mathbf{m}}(t),T-t)}{\partial\bar{\mathbf{m}}(t)}\left[\bar{\mathbf{m}}(t+h/2) - \bar{\mathbf{m}}(t)\right] \tag{169}$$

$$\boldsymbol{s}_\theta^m(\bar{\mathbf{x}}(t+h),\bar{\mathbf{m}}(t+h/2),T-t) = \boldsymbol{s}_\theta^m(\bar{\mathbf{x}}(t),\bar{\mathbf{m}}(t),T-t) + \frac{\partial\boldsymbol{s}_\theta^m(\bar{\mathbf{x}}(t),\bar{\mathbf{m}}(t),T-t)}{\partial\bar{\mathbf{x}}(t)} \tag{170}$$

$$\left[\frac{h\beta}{2}\left(\Gamma\bar{\mathbf{x}}(t) - M^{-1}\bar{\mathbf{m}}(t) + \Gamma\boldsymbol{s}_\theta^x(\bar{\mathbf{x}}(t),\bar{\mathbf{m}}(t+h/2),T-t)\right)\right] + \tag{171}$$

$$\frac{h}{2}\frac{\partial\boldsymbol{s}_\theta^m(\bar{\mathbf{x}}(t),\bar{\mathbf{m}}(t),T-t)}{\partial\bar{\mathbf{m}}(t)}\frac{d\bar{\mathbf{m}}_t}{dt} \tag{172}$$

$$\boldsymbol{s}_\theta^m(\bar{\mathbf{x}}(t+h),\bar{\mathbf{m}}(t+h/2),T-t) = \boldsymbol{s}_\theta^m(\bar{\mathbf{x}}(t),\bar{\mathbf{m}}(t),T-t) + h\frac{\partial\boldsymbol{s}_\theta^m(\bar{\mathbf{x}}(t),\bar{\mathbf{m}}(t),T-t)}{\partial\bar{\mathbf{x}}(t)}\frac{d\bar{\mathbf{x}}_t}{dt} + \tag{173}$$

$$\frac{h}{2}\frac{\partial\boldsymbol{s}_\theta^m(\bar{\mathbf{x}}(t),\bar{\mathbf{m}}(t),T-t)}{\partial\bar{\mathbf{m}}(t)}\frac{d\bar{\mathbf{m}}_t}{dt} + \frac{h\beta\Gamma}{2}\left[\boldsymbol{s}_\theta^x(\bar{\mathbf{x}}(t),\bar{\mathbf{m}}(t+h/2),T-t) - \right. \tag{174}$$

$$\left.\boldsymbol{s}_\theta^x(\bar{\mathbf{x}}(t),\bar{\mathbf{m}}(t),T-t)\right] \tag{175}$$

Substituting the above results in Eqn. 167, we get the following result,

$$\bar{\mathbf{m}}(t+h) = \bar{\mathbf{m}}(t) + \frac{h\beta}{2}\left[\bar{\mathbf{x}}(t) + \nu\bar{\mathbf{m}}(t) + M\nu\boldsymbol{s}_\theta^m(\bar{\mathbf{x}}(t),\bar{\mathbf{m}}(t),T-t)\right] + \frac{h^2\beta}{4}\left[\frac{d\bar{\mathbf{x}}_t}{dt}\right] + \tag{176}$$

$$\frac{h^2\beta\nu}{8}\frac{d\bar{\mathbf{m}}(t)}{dt} + \frac{h^2\beta M\nu}{4}\left[\frac{\partial\boldsymbol{s}_\theta^m(\bar{\mathbf{x}}(t),\bar{\mathbf{m}}(t),T-t)}{\partial\bar{\mathbf{x}}(t)}\frac{d\bar{\mathbf{x}}_t}{dt} + \frac{1}{2}\frac{\partial\boldsymbol{s}_\theta^m(\bar{\mathbf{x}}(t),\bar{\mathbf{m}}(t),T-t)}{\partial\bar{\mathbf{m}}(t)}\frac{d\bar{\mathbf{m}}_t}{dt}\right] \tag{177}$$

$$+ \frac{h^2\beta^2\Gamma(1+M\nu)}{8}\left[\boldsymbol{s}_\theta^x(\bar{\mathbf{x}}(t),\bar{\mathbf{m}}(t+h/2),T-t) - \boldsymbol{s}_\theta^x(\bar{\mathbf{x}}(t),\bar{\mathbf{m}}(t),T-t)\right] + \mathcal{O}(h^3) \tag{178}$$

Using the multivariate Taylor-series expansion, we approximate $\boldsymbol{s}_\theta^x(\bar{\mathbf{x}}(t),\bar{\mathbf{m}}(t+h/2),T-t)$ around $\boldsymbol{s}_\theta^x(\bar{\mathbf{x}}(t),\bar{\mathbf{m}}(t),T-t)$ using a first-order approximation as follows,

$$\boldsymbol{s}_\theta^x(\bar{\mathbf{x}}(t),\bar{\mathbf{m}}(t+h/2),T-t) \approx \boldsymbol{s}_\theta^x(\bar{\mathbf{x}}(t),\bar{\mathbf{m}}(t),T-t) + \frac{\partial\boldsymbol{s}_\theta^x(\bar{\mathbf{x}}(t),\bar{\mathbf{m}}(t),T-t)}{\partial\bar{\mathbf{m}}(t)}\left[\bar{\mathbf{m}}(t+h/2) - \bar{\mathbf{m}}(t)\right] \tag{179}$$

$$\boldsymbol{s}_\theta^x(\bar{\mathbf{x}}(t),\bar{\mathbf{m}}(t+h/2),T-t) \approx \boldsymbol{s}_\theta^x(\bar{\mathbf{x}}(t),\bar{\mathbf{m}}(t),T-t) + \frac{\partial\boldsymbol{s}_\theta^x(\bar{\mathbf{x}}(t),\bar{\mathbf{m}}(t),T-t)}{\partial\bar{\mathbf{m}}(t)} \tag{180}$$

$$\left[\frac{h\beta}{4}\left[\bar{\mathbf{x}}(t) + \nu\bar{\mathbf{m}}(t) + M\nu\boldsymbol{s}_\theta^m(\bar{\mathbf{x}}(t),\bar{\mathbf{m}}(t),T-t)\right]\right] \tag{181}$$

Substituting the above result in Eqn. 178 and ignoring the higher order terms in $\mathcal{O}(h^3)$, we get,

$$\bar{\mathbf{m}}(t+h) = \bar{\mathbf{m}}(t) + \frac{h\beta}{2}\left[\bar{\mathbf{x}}(t) + \nu\bar{\mathbf{m}}(t) + M\nu\boldsymbol{s}_\theta^m(\bar{\mathbf{x}}(t),\bar{\mathbf{m}}(t),T-t)\right] + \frac{h^2\beta}{4}\left[\frac{d\bar{\mathbf{x}}_t}{dt}\right] + \tag{182}$$

$$\frac{h^2\beta\nu}{8}\frac{d\bar{\mathbf{m}}(t)}{dt} + \frac{h^2\beta M\nu}{4}\left[\frac{\partial\boldsymbol{s}_\theta^m(\bar{\mathbf{x}}(t),\bar{\mathbf{m}}(t),T-t)}{\partial\bar{\mathbf{x}}(t)}\frac{d\bar{\mathbf{x}}_t}{dt} + \frac{1}{2}\frac{\partial\boldsymbol{s}_\theta^m(\bar{\mathbf{x}}(t),\bar{\mathbf{m}}(t),T-t)}{\partial\bar{\mathbf{m}}(t)}\frac{d\bar{\mathbf{m}}_t}{dt}\right] \tag{183}$$

$$\bar{\mathbf{m}}(t+h) = \bar{\mathbf{m}}(t) + \frac{h\beta}{2}\left[\bar{\mathbf{x}}(t) + \nu\bar{\mathbf{m}}(t) + M\nu\boldsymbol{s}_\theta^m(\bar{\mathbf{x}}(t),\bar{\mathbf{m}}(t),T-t)\right] + \frac{h^2\beta}{4}\left[\frac{d\bar{\mathbf{x}}_t}{dt} + \nu\frac{d\bar{\mathbf{m}}_t}{dt} + M\nu\right. \tag{184}$$

$$\underbrace{\left(\frac{\partial\boldsymbol{s}_\theta^m(\bar{\mathbf{x}}(t),\bar{\mathbf{m}}(t),T-t)}{\partial\bar{\mathbf{x}}(t)}\frac{d\bar{\mathbf{x}}_t}{dt} + \frac{\partial\boldsymbol{s}_\theta^m(\bar{\mathbf{x}}(t),\bar{\mathbf{m}}(t),T-t)}{\partial\bar{\mathbf{m}}(t)}\frac{d\bar{\mathbf{m}}_t}{dt} + \frac{\partial\boldsymbol{s}_\theta^m(\bar{\mathbf{x}}(t),\bar{\mathbf{m}}(t),T-t)}{\partial t}\right)}_{=\frac{d}{dt}\boldsymbol{s}_\theta^m(\bar{\mathbf{x}}(t),\bar{\mathbf{m}}(t),T-t)} \tag{185}$$

$$\left. - \frac{\nu}{2}\frac{d\bar{\mathbf{m}}(t)}{dt} - \frac{M\nu}{2}\frac{\partial\boldsymbol{s}_\theta^m(\bar{\mathbf{x}}(t),\bar{\mathbf{m}}(t),T-t)}{\partial\bar{\mathbf{m}}(t)} - M\nu\frac{\partial\boldsymbol{s}_\theta^m(\bar{\mathbf{x}}(t),\bar{\mathbf{m}}(t),T-t)}{\partial t}\right] \tag{186}$$

$$\bar{\mathbf{m}}(t+h) = \bar{\mathbf{m}}(t) + \frac{h\beta}{2}\left[\bar{\mathbf{x}}(t) + \nu\bar{\mathbf{m}}(t) + M\nu\boldsymbol{s}_\theta^m(\bar{\mathbf{x}}(t),\bar{\mathbf{m}}(t),T-t)\right] + \frac{h^2\beta}{4}\left[\frac{d\bar{\mathbf{x}}_t}{dt} + \nu\frac{d\bar{\mathbf{m}}_t}{dt} + M\nu\right. \tag{187}$$

$$\left.\frac{d\boldsymbol{s}_\theta^m(\bar{\mathbf{x}}(t),\bar{\mathbf{m}}(t),T-t)}{dt}\right] - \frac{h^2\beta\nu}{8}\left[\frac{d\bar{\mathbf{m}}(t)}{dt} + M\frac{\partial\boldsymbol{s}_\theta^m(\bar{\mathbf{x}}(t),\bar{\mathbf{m}}(t),T-t)}{\partial\bar{\mathbf{m}}(t)} + \tag{188}$$

$$2M\frac{\partial\boldsymbol{s}_\theta^m(\bar{\mathbf{x}}(t),\bar{\mathbf{m}}(t),T-t)}{\partial t}\right] \tag{189}$$

We can now use the above result to analyze the local truncation error in the momentum space as follows,

$$\mathcal{F}_h(\bar{\mathbf{m}}(t)) - \mathcal{G}_h(\bar{\mathbf{m}}(t)) = \frac{h^2\beta\nu}{8}\left[\frac{d\bar{\mathbf{m}}(t)}{dt} + M\frac{\partial\boldsymbol{s}_\theta^m(\bar{\mathbf{x}}(t),\bar{\mathbf{m}}(t),T-t)}{\partial\bar{\mathbf{m}}(t)} + 2M\frac{\partial\boldsymbol{s}_\theta^m(\bar{\mathbf{x}}(t),\bar{\mathbf{m}}(t),T-t)}{\partial t}\right] \tag{190}$$

The above equation implies that,

$$\|\mathcal{F}_h(\bar{\mathbf{m}}(t)) - \mathcal{G}_h(\bar{\mathbf{m}}(t))\| \le \frac{C\beta\nu h^2}{8} \tag{191}$$

Since we choose $\beta = 8$ throughout this work, $\beta/8 = 1$ can be absorbed in the constant $C$. Therefore, the local truncation error for the Naive Velocity Verlet (NVV) in the momentum space is of the order of $\mathcal{O}(\nu h^2)$.

While the NVV sampler has nice theoretical properties, the local truncation error analysis can be misleading for large step sizes. This is because at low NFE regimes (or with high step sizes $h$), the assumption to ignore error contribution from higher-order terms like $\mathcal{O}(h^3)$ might not be reasonable. In the NVV scheme, we make a similar assumption in Eqns. 135,167 and 178 (when approximating the term in blue). This is the primary motivation for re-using the score function evaluation $\boldsymbol{s}_\theta(\bar{\mathbf{x}}_t, \bar{\mathbf{m}}_t, T - t)$ between consecutive position and momentum updates in the RVV scheme. This design choice has the following advantages:

1. Firstly, re-using the score function evaluation $\boldsymbol{s}_\theta(\bar{\mathbf{x}}_t, \bar{\mathbf{m}}_t, T - t)$ between consecutive position and momentum updates exactly cancels out the term in blue in Eqn. 135 eliminating error contribution from additional terms introduced by approximating $\boldsymbol{s}_\theta(\bar{\mathbf{x}}(t), \bar{\mathbf{m}}(t + h/2), T - t)$. This is especially significant for larger step sizes during sampling.
2. Secondly, re-using a score function evaluation also reduces the number of NFEs per update step from **three** in NVV to **two** in RVV. This allows the use of smaller step sizes during inference for the same compute budget.

Next, we analyze the local truncation error for the RVV sampler.

### C.2.5 ERROR ANALYSIS: REDUCED VELOCITY VERLET (RVV)

The NVV sampler has the following update rules:

$$\bar{\mathbf{m}}_{t+h/2} = \bar{\mathbf{m}}_t + \frac{h\beta}{4}\left[\bar{\mathbf{x}}_t + \nu\bar{\mathbf{m}}_t + M\nu\boldsymbol{s}_\theta^m(\bar{\mathbf{x}}_t, \bar{\mathbf{m}}_t, T - t)\right] \tag{192}$$

$$\bar{\mathbf{x}}_{t+h} = \bar{\mathbf{x}}_t + \frac{h\beta}{2}\left[\Gamma\bar{\mathbf{x}}_t - M^{-1}\bar{\mathbf{m}}_{t+h/2} + \Gamma\boldsymbol{s}_\theta^x(\bar{\mathbf{x}}_t, \bar{\mathbf{m}}_t, T - t)\right] \tag{193}$$

$$\bar{\mathbf{m}}_{t+h} = \bar{\mathbf{m}}_{t+h/2} + \frac{h\beta}{4}\left[\bar{\mathbf{x}}_{t+h} + \nu\bar{\mathbf{m}}_{t+h/2} + M\nu\boldsymbol{s}_\theta^m(\bar{\mathbf{x}}_{t+h}, \bar{\mathbf{m}}_{t+h/2}, T - (t + h))\right] \tag{194}$$

Similar to our analysis for the NVV sampler, we first compute the local truncation error in both the position and the momentum space.

**RVV local truncation error in the position space:** From the update equations,

$$\bar{\mathbf{x}}(t + h) = \bar{\mathbf{x}}(t) + \frac{h\beta}{2}\left[\Gamma\bar{\mathbf{x}}(t) - M^{-1}\bar{\mathbf{m}}(t + h/2) + \Gamma\boldsymbol{s}_\theta^x(\bar{\mathbf{x}}(t), \bar{\mathbf{m}}(t), T - t)\right] \tag{195}$$

$$= \bar{\mathbf{x}}(t) + \frac{h\beta}{2}\left[\Gamma\bar{\mathbf{x}}(t) - M^{-1}\left(\bar{\mathbf{m}}(t) + \frac{h\beta}{4}\left[\bar{\mathbf{x}}(t) + \nu\bar{\mathbf{m}}(t) + M\nu\boldsymbol{s}_\theta^m(\bar{\mathbf{x}}(t), \bar{\mathbf{m}}(t), T - t)\right]\right)\right. \tag{196}$$

$$\left. + \Gamma\boldsymbol{s}_\theta^x(\bar{\mathbf{x}}(t), \bar{\mathbf{m}}(t), T - t)\right] \tag{197}$$

$$\mathcal{G}_h(\bar{\mathbf{x}}(t)) = \bar{\mathbf{x}}(t) + \frac{h\beta}{2}\left[\Gamma\bar{\mathbf{x}}(t) - M^{-1}\bar{\mathbf{m}}(t) + \Gamma\boldsymbol{s}_\theta^x(\bar{\mathbf{x}}(t), \bar{\mathbf{m}}(t), T - t)\right] - \tag{198}$$

$$\frac{h^2\beta^2 M^{-1}}{8}\left[\bar{\mathbf{x}}(t) + \nu\bar{\mathbf{m}}(t) + M\nu\boldsymbol{s}_\theta^m(\bar{\mathbf{x}}(t), \bar{\mathbf{m}}(t), T - t)\right] \tag{199}$$

$$\mathcal{G}_h(\bar{\mathbf{x}}(t)) = \bar{\mathbf{x}}(t) + \frac{h\beta}{2}\left[\Gamma\bar{\mathbf{x}}(t) - M^{-1}\bar{\mathbf{m}}(t) + \Gamma\boldsymbol{s}_\theta^x(\bar{\mathbf{x}}(t), \bar{\mathbf{m}}(t), T - t)\right] - \frac{h^2\beta M^{-1}}{4}\frac{d\bar{\mathbf{m}}(t)}{dt} \tag{200}$$

Therefore, the local truncation error in the position space is given by,

$$\mathcal{F}_h(\bar{\mathbf{x}}(t)) - \mathcal{G}_h(\bar{\mathbf{x}}(t)) = \frac{h^2\beta\Gamma}{4}\frac{d}{dt}\left[\bar{\mathbf{x}}(t) + \boldsymbol{s}_\theta^x(\bar{\mathbf{x}}(t), \bar{\mathbf{m}}(t), T - t)\right] \tag{201}$$

The above equation implies that,

$$\boxed{\|\mathcal{F}_h(\bar{\mathbf{x}}(t)) - \mathcal{G}_h(\bar{\mathbf{x}}(t))\| \leq \frac{\bar{C}\beta\Gamma h^2}{4}} \tag{202}$$

Similar to the NVV case, the local truncation error for RVV is of the order $\mathcal{O}(\Gamma h^2)$. Since $\Gamma$ is usually small in PSLD (Pandey & Mandt, 2023) (for instance, 0.01 for CIFAR-10 and 0.005 for CelebA-64), its magnitude is comparable or less than $h$ (particularly in the low NFE regime). Therefore, the effective local truncation order for the NVV scheme is of the order of $\mathcal{O}(h^3)$.

Next, we analyze the local truncation error for RVV in the momentum space.

**RVV local truncation error in the momentum space:** From the update equations,

$$\bar{\mathbf{m}}(t+h) = \bar{\mathbf{m}}(t+h/2) + \frac{h\beta}{4}\Big[\bar{\mathbf{x}}(t+h) + \nu\bar{\mathbf{m}}(t+h/2) + \tag{203}$$

$$M\nu\boldsymbol{s}_\theta^m(\bar{\mathbf{x}}(t+h), \bar{\mathbf{m}}(t+h/2), T-(t+h))\Big] \tag{204}$$

$$\bar{\mathbf{m}}(t+h) = \bar{\mathbf{m}}(t) + \frac{h\beta}{4}\left[\bar{\mathbf{x}}(t) + \nu\bar{\mathbf{m}}(t) + M\nu\boldsymbol{s}_\theta^m(\bar{\mathbf{x}}(t), \bar{\mathbf{m}}(t), T-t)\right] + \frac{h\beta}{4}\left[\bar{\mathbf{x}}(t+h)\right] + \tag{205}$$

$$\frac{h\beta\nu}{4}\left[\bar{\mathbf{m}}(t+h/2)\right] + \frac{h\beta M\nu}{4}\boldsymbol{s}_\theta^m(\bar{\mathbf{x}}(t+h), \bar{\mathbf{m}}(t+h/2), T-(t+h)) \tag{206}$$

$$\bar{\mathbf{m}}(t+h) = \bar{\mathbf{m}}(t) + \frac{h\beta}{4}\left[\bar{\mathbf{x}}(t) + \nu\bar{\mathbf{m}}(t) + M\nu\boldsymbol{s}_\theta^m(\bar{\mathbf{x}}(t), \bar{\mathbf{m}}(t), T-t)\right] + \tag{207}$$

$$\frac{h\beta}{4}\left[\bar{\mathbf{x}}(t) + \frac{h\beta}{2}\left[\Gamma\bar{\mathbf{x}}(t) - M^{-1}\bar{\mathbf{m}}(t+h/2) + \Gamma\boldsymbol{s}_\theta^x(\bar{\mathbf{x}}(t), \bar{\mathbf{m}}(t), T-t)\right]\right] + \tag{208}$$

$$\frac{h\beta\nu}{4}\left[\bar{\mathbf{m}}(t) + \frac{h\beta}{4}\left[\bar{\mathbf{x}}(t) + \nu\bar{\mathbf{m}}(t) + M\nu\boldsymbol{s}_\theta^m(\bar{\mathbf{x}}(t), \bar{\mathbf{m}}(t), T-t)\right]\right] + \tag{209}$$

$$\frac{h\beta M\nu}{4}\boldsymbol{s}_\theta^m(\bar{\mathbf{x}}(t+h), \bar{\mathbf{m}}(t+h/2), T-(t+h)) \tag{210}$$

$$\bar{\mathbf{m}}(t+h) = \bar{\mathbf{m}}(t) + \frac{h\beta}{2}\left[\bar{\mathbf{x}}(t) + \nu\bar{\mathbf{m}}(t) + M\nu\boldsymbol{s}_\theta^m(\bar{\mathbf{x}}(t), \bar{\mathbf{m}}(t), T-t)\right] + \tag{211}$$

$$\frac{h^2\beta}{4}\left[\frac{\beta}{2}\left[\Gamma\bar{\mathbf{x}}(t) - M^{-1}\bar{\mathbf{m}}(t+h/2) + \Gamma\boldsymbol{s}_\theta^x(\bar{\mathbf{x}}(t), \bar{\mathbf{m}}(t), T-t)\right]\right] + \tag{212}$$

$$\frac{h^2\beta\nu}{8}\underbrace{\left[\frac{\beta}{2}\left[\bar{\mathbf{x}}(t) + \nu\bar{\mathbf{m}}(t) + M\nu\boldsymbol{s}_\theta^m(\bar{\mathbf{x}}(t), \bar{\mathbf{m}}(t), T-t)\right]\right]}_{=\frac{d\bar{\mathbf{m}}(t)}{dt}} + \tag{213}$$

$$\frac{h\beta M\nu}{4}\left[\boldsymbol{s}_\theta^m(\bar{\mathbf{x}}(t+h), \bar{\mathbf{m}}(t+h/2), T-(t+h)) - \boldsymbol{s}_\theta^m(\bar{\mathbf{x}}(t), \bar{\mathbf{m}}(t), T-t)\right] \tag{214}$$

$$\bar{\mathbf{m}}(t+h) = \bar{\mathbf{m}}(t) + \frac{h\beta}{2}\left[\bar{\mathbf{x}}(t) + \nu\bar{\mathbf{m}}(t) + M\nu\boldsymbol{s}_\theta^m(\bar{\mathbf{x}}(t), \bar{\mathbf{m}}(t), T-t)\right] + \tag{215}$$

$$\frac{h^2\beta}{4}\left[\frac{\beta}{2}\left(\Gamma\bar{\mathbf{x}}(t) - M^{-1}\left(\bar{\mathbf{m}}(t) + \frac{h\beta}{4}\left[\bar{\mathbf{x}}(t) + \nu\bar{\mathbf{m}}(t) + M\nu\boldsymbol{s}_\theta^m(\bar{\mathbf{x}}(t), \bar{\mathbf{m}}(t), T-t)\right]\right)\right.\right.$$ 
$$\tag{216}$$

$$\left.\left. + \Gamma\boldsymbol{s}_\theta^x(\bar{\mathbf{x}}(t), \bar{\mathbf{m}}(t), T-t)\right)\right] + \frac{h^2\beta\nu}{8}\frac{d\bar{\mathbf{m}}(t)}{dt} + \tag{217}$$

$$\frac{h\beta M\nu}{4}\left[\boldsymbol{s}_\theta^m(\bar{\mathbf{x}}(t+h), \bar{\mathbf{m}}(t+h/2), T-(t+h)) - \boldsymbol{s}_\theta^m(\bar{\mathbf{x}}(t), \bar{\mathbf{m}}(t), T-t)\right] \tag{218}$$

$$\tag{219}$$

$$\bar{\mathbf{m}}(t+h) = \bar{\mathbf{m}}(t) + \frac{h\beta}{2}\left[\bar{\mathbf{x}}(t) + \nu\bar{\mathbf{m}}(t) + M\nu\boldsymbol{s}_\theta^m(\bar{\mathbf{x}}(t), \bar{\mathbf{m}}(t), T-t)\right] + \tag{220}$$

$$\frac{h^2\beta}{4}\left[\underbrace{\frac{\beta}{2}\left(\Gamma\bar{\mathbf{x}}(t) - M^{-1}\bar{\mathbf{m}}(t) + \Gamma\boldsymbol{s}_\theta^x(\bar{\mathbf{x}}(t), \bar{\mathbf{m}}(t), T-t)\right)}_{=\frac{d\bar{\mathbf{x}}_t}{dt}}\right] + \frac{h^2\beta\nu}{8}\frac{d\bar{\mathbf{m}}(t)}{dt} + \tag{221}$$

$$\frac{h\beta M\nu}{4}\left[\boldsymbol{s}_\theta^m(\bar{\mathbf{x}}(t+h), \bar{\mathbf{m}}(t+h/2), T-t) - \boldsymbol{s}_\theta^m(\bar{\mathbf{x}}(t), \bar{\mathbf{m}}(t), T-(t+h))\right] + \mathcal{O}(h^3) \tag{222}$$

Approximating $\boldsymbol{s}_\theta^m(\bar{\mathbf{x}}(t+h), \bar{\mathbf{m}}(t+h/2), T-(t+h))$ around $\boldsymbol{s}_\theta^m(\bar{\mathbf{x}}(t), \bar{\mathbf{m}}(t), T-t)$ using a first-order Taylor series approximation (Ignoring higher order terms in $\mathcal{O}(h^2)$),

$$\boldsymbol{s}_\theta^m(\bar{\mathbf{x}}(t+h), \bar{\mathbf{m}}(t+h/2), T-(t+h)) \approx \boldsymbol{s}_\theta^m(\bar{\mathbf{x}}(t), \bar{\mathbf{m}}(t), T-t) + h\frac{\partial\boldsymbol{s}_\theta^m(\bar{\mathbf{x}}(t), \bar{\mathbf{m}}(t), T-t)}{\partial t} \tag{223}$$

$$+ \frac{\partial\boldsymbol{s}_\theta^m(\bar{\mathbf{x}}(t), \bar{\mathbf{m}}(t), T-t)}{\partial\bar{\mathbf{x}}(t)}\left[\bar{\mathbf{x}}(t+h) - \bar{\mathbf{x}}(t)\right] \tag{224}$$

$$+ \frac{\partial\boldsymbol{s}_\theta^m(\bar{\mathbf{x}}(t), \bar{\mathbf{m}}(t), T-t)}{\partial\bar{\mathbf{m}}(t)}\left[\bar{\mathbf{m}}(t+h/2) - \bar{\mathbf{m}}(t)\right] \tag{225}$$

$$\boldsymbol{s}_\theta^m(\bar{\mathbf{x}}(t+h), \bar{\mathbf{m}}(t+h/2), T-t) = \boldsymbol{s}_\theta^m(\bar{\mathbf{x}}(t), \bar{\mathbf{m}}(t), T-t) + h\frac{\partial\boldsymbol{s}_\theta^m(\bar{\mathbf{x}}(t), \bar{\mathbf{m}}(t), T-t)}{\partial t} + \tag{226}$$

$$\frac{h}{2}\frac{\partial\boldsymbol{s}_\theta^m(\bar{\mathbf{x}}(t), \bar{\mathbf{m}}(t), T-t)}{\partial\bar{\mathbf{m}}(t)}\frac{d\bar{\mathbf{m}}_t}{dt} + \frac{\partial\boldsymbol{s}_\theta^m(\bar{\mathbf{x}}(t), \bar{\mathbf{m}}(t), T-t)}{\partial\bar{\mathbf{x}}(t)} \tag{227}$$

$$\left[\frac{h\beta}{2}\left(\Gamma\bar{\mathbf{x}}(t) - M^{-1}\bar{\mathbf{m}}(t+h/2) + \Gamma\boldsymbol{s}_\theta^x(\bar{\mathbf{x}}(t), \bar{\mathbf{m}}(t+h/2), T-t)\right)\right] \tag{228}$$

$$\boldsymbol{s}_\theta^m(\bar{\mathbf{x}}(t+h), \bar{\mathbf{m}}(t+h/2), T-t) = \boldsymbol{s}_\theta^m(\bar{\mathbf{x}}(t), \bar{\mathbf{m}}(t), T-t) + h\frac{\partial\boldsymbol{s}_\theta^m(\bar{\mathbf{x}}(t), \bar{\mathbf{m}}(t), T-t)}{\partial\bar{\mathbf{x}}(t)}\frac{d\bar{\mathbf{x}}_t}{dt} + \tag{229}$$

$$\frac{h}{2}\frac{\partial\boldsymbol{s}_\theta^m(\bar{\mathbf{x}}(t), \bar{\mathbf{m}}(t), T-t)}{\partial\bar{\mathbf{m}}(t)}\frac{d\bar{\mathbf{m}}_t}{dt} + h\frac{\partial\boldsymbol{s}_\theta^m(\bar{\mathbf{x}}(t), \bar{\mathbf{m}}(t), T-t)}{\partial t} \tag{230}$$

Substituting the above results in Eqn. 222, we get the following result,

$$\bar{\mathbf{m}}(t+h) = \bar{\mathbf{m}}(t) + \frac{h\beta}{2}\left[\bar{\mathbf{x}}(t) + \nu\bar{\mathbf{m}}(t) + M\nu\boldsymbol{s}_\theta^m(\bar{\mathbf{x}}(t), \bar{\mathbf{m}}(t), T-t)\right] + \frac{h^2\beta}{4}\left[\frac{d\bar{\mathbf{x}}_t}{dt}\right] + \tag{231}$$

$$\frac{h^2\beta\nu}{8}\frac{d\bar{\mathbf{m}}(t)}{dt} + \frac{h^2\beta M\nu}{4}\left[\frac{\partial\boldsymbol{s}_\theta^m(\bar{\mathbf{x}}(t), \bar{\mathbf{m}}(t), T-t)}{\partial\bar{\mathbf{x}}(t)}\frac{d\bar{\mathbf{x}}_t}{dt} + \frac{1}{2}\frac{\partial\boldsymbol{s}_\theta^m(\bar{\mathbf{x}}(t), \bar{\mathbf{m}}(t), T-t)}{\partial\bar{\mathbf{m}}(t)}\frac{d\bar{\mathbf{m}}_t}{dt}\right. \tag{232}$$

$$\left. + h\frac{\partial\boldsymbol{s}_\theta^m(\bar{\mathbf{x}}(t), \bar{\mathbf{m}}(t), T-t)}{\partial t}\right] + \mathcal{O}(h^3) \tag{233}$$

$$\bar{\mathbf{m}}(t+h) = \bar{\mathbf{m}}(t) + \frac{h\beta}{2}\left[\bar{\mathbf{x}}(t) + \nu\bar{\mathbf{m}}(t) + M\nu\boldsymbol{s}_\theta^m(\bar{\mathbf{x}}(t), \bar{\mathbf{m}}(t), T-t)\right] + \frac{h^2\beta}{4}\left[\frac{d\bar{\mathbf{x}}_t}{dt} + \nu\frac{d\bar{\mathbf{m}}_t}{dt} + M\nu\right.$$

$$(234)$$

$$\underbrace{\left(\frac{\partial\boldsymbol{s}_\theta^m(\bar{\mathbf{x}}(t),\bar{\mathbf{m}}(t),T-t)}{\partial\bar{\mathbf{x}}(t)}\frac{d\bar{\mathbf{x}}_t}{dt} + \frac{\partial\boldsymbol{s}_\theta^m(\bar{\mathbf{x}}(t),\bar{\mathbf{m}}(t),T-t)}{\partial\bar{\mathbf{m}}(t)}\frac{d\bar{\mathbf{m}}_t}{dt} + \frac{\partial\boldsymbol{s}_\theta^m(\bar{\mathbf{x}}(t),\bar{\mathbf{m}}(t),T-t)}{\partial t}\right)}_{=\frac{d}{dt}\boldsymbol{s}_\theta^m(\bar{\mathbf{x}}(t),\bar{\mathbf{m}}(t),T-t)}$$

$$(235)$$

$$\left.- \frac{\nu}{2}\frac{d\bar{\mathbf{m}}(t)}{dt} - \frac{M\nu}{2}\frac{\partial\boldsymbol{s}_\theta^m(\bar{\mathbf{x}}(t),\bar{\mathbf{m}}(t),T-t)}{\partial\bar{\mathbf{m}}(t)}\right]$$

$$(236)$$

$$\bar{\mathbf{m}}(t+h) = \bar{\mathbf{m}}(t) + \frac{h\beta}{2}\left[\bar{\mathbf{x}}(t) + \nu\bar{\mathbf{m}}(t) + M\nu\boldsymbol{s}_\theta^m(\bar{\mathbf{x}}(t),\bar{\mathbf{m}}(t),T-t)\right] + \frac{h^2\beta}{4}\left[\frac{d\bar{\mathbf{x}}_t}{dt} + \nu\frac{d\bar{\mathbf{m}}_t}{dt} + M\nu\right.$$

$$(237)$$

$$\left.\frac{d\boldsymbol{s}_\theta^m(\bar{\mathbf{x}}(t),\bar{\mathbf{m}}(t),T-t)}{dt}\right] - \frac{h^2\beta\nu}{8}\left[\frac{d\bar{\mathbf{m}}(t)}{dt} + M\frac{\partial\boldsymbol{s}_\theta^m(\bar{\mathbf{x}}(t),\bar{\mathbf{m}}(t),T-t)}{\partial\bar{\mathbf{m}}(t)}\right] \quad (238)$$

We can now use the above result to analyze the local truncation error in the momentum space as follows,

$$\mathcal{F}_h(\bar{\mathbf{m}}(t)) - \mathcal{G}_h(\bar{\mathbf{m}}(t)) = \frac{h^2\beta\nu}{8}\left[\frac{d\bar{\mathbf{m}}(t)}{dt} + M\frac{\partial\boldsymbol{s}_\theta^m(\bar{\mathbf{x}}(t),\bar{\mathbf{m}}(t),T-t)}{\partial\bar{\mathbf{m}}(t)}\right] \quad (239)$$

The above equation implies that,

$$\boxed{\|\mathcal{F}_h(\bar{\mathbf{m}}(t)) - \mathcal{G}_h(\bar{\mathbf{m}}(t))\| \leq \frac{C\beta\nu h^2}{8}} \quad (240)$$

Similar to the NVV sampler, the scaling factor $\beta/8 = 1$ can be absorbed in the constant $C$. Therefore, the local truncation error for the Reduced Velocity Verlet (RVV) in the momentum space is of the order of $\mathcal{O}(\nu h^2)$.

### C.3 STOCHASTIC SPLITTING INTEGRATORS

We split the Reverse Diffusion SDE for PSLD using the following splitting scheme.

$$\begin{pmatrix}d\bar{\mathbf{x}}_t \\ d\bar{\mathbf{m}}_t\end{pmatrix} = \underbrace{\frac{\beta}{2}\begin{pmatrix}2\Gamma\bar{\mathbf{x}}_t - M^{-1}\bar{\mathbf{m}}_t + 2\Gamma\boldsymbol{s}_\theta^x(\bar{\mathbf{z}}_t, t) \\ 0\end{pmatrix}dt}_{A} + O + \underbrace{\frac{\beta}{2}\begin{pmatrix}0 \\ \bar{\mathbf{x}}_t + 2\nu\bar{\mathbf{m}}_t + 2M\nu\boldsymbol{s}_\theta^m(\bar{\mathbf{z}}_t, t)\end{pmatrix}dt}_{B}$$

$$(241)$$

where $O = \begin{pmatrix}-\frac{\beta\Gamma}{2}\bar{\mathbf{x}}_t dt + \sqrt{\beta\Gamma}d\bar{\mathbf{w}}_t \\ -\frac{\beta\nu}{2}\bar{\mathbf{m}}_t dt + \sqrt{M\nu\beta}d\bar{\mathbf{w}}_t\end{pmatrix}$ is the Ornstein-Uhlenbeck component which injects stochasticity during sampling. Similar to the deterministic case, $\bar{\mathbf{x}}_t = \mathbf{x}_{T-t}$, $\bar{\mathbf{m}}_t = \mathbf{m}_{T-t}$, $\boldsymbol{s}_\theta^x$ and $\boldsymbol{s}_\theta^m$ denote the score components in the data and momentum space, respectively. We approximate the solution for splits $A$ and $B$ using a simple Euler-based numerical approximation. Formally, we denote the Euler approximation for the splits $A$ and $B$ by $\mathcal{L}_A$ and $\mathcal{L}_B$, respectively, with their corresponding numerical updates specified as:

$$\mathcal{L}_A : \begin{cases}\bar{\mathbf{x}}_{t+h} &= \bar{\mathbf{x}}_t + \frac{h\beta}{2}\left[\Gamma\bar{\mathbf{x}}_t - M^{-1}\bar{\mathbf{m}}_t + \Gamma\boldsymbol{s}_\theta^x(\bar{\mathbf{x}}_t, \bar{\mathbf{m}}_t, T-t)\right] \\ \bar{\mathbf{m}}_{t+h} &= \bar{\mathbf{m}}_t\end{cases} \quad (242)$$

$$\mathcal{L}_B : \begin{cases}\bar{\mathbf{x}}_{t+h} &= \bar{\mathbf{x}}_t \\ \bar{\mathbf{m}}_{t+h} &= \bar{\mathbf{m}}_t + \frac{h\beta}{2}\left[\bar{\mathbf{x}}_t + \nu\bar{\mathbf{m}}_t + M\nu\boldsymbol{s}_\theta^m(\bar{\mathbf{x}}_t, \bar{\mathbf{m}}_t, T-t)\right]\end{cases} \quad (243)$$

It is worth noting that the solution to the OU component can be computed analytically:

$$\mathcal{L}_O : \begin{cases}\bar{\mathbf{x}}_{t+h} &= \exp\left(\frac{-h\beta\Gamma}{2}\right)\bar{\mathbf{x}}_t + \sqrt{1 - \exp(-h\beta\Gamma)}\boldsymbol{\epsilon}_x, \quad \boldsymbol{\epsilon}_x \sim \mathcal{N}(\mathbf{0}_d, \boldsymbol{I}_d) \\ \bar{\mathbf{m}}_{t+h} &= \exp\left(\frac{-h\beta\nu}{2}\right)\bar{\mathbf{m}}_t + \sqrt{M}\sqrt{1 - \exp(-h\beta\nu)}\boldsymbol{\epsilon}_m, \quad \boldsymbol{\epsilon}_m \sim \mathcal{N}(\mathbf{0}_d, \boldsymbol{I}_d)\end{cases} \quad (244)$$

Next, we highlight the numerical update equations for the Naive-OBA sampler and the Reduced OBA, BAO, and OBAB samplers.

### C.3.1 NAIVE SPLITTING SAMPLERS

**Naive OBA**: In this scheme, for a given step size h, the solutions to the splitting pieces $\mathcal{L}_h^A$, $\mathcal{L}_h^B$ and $\mathcal{L}_h^O$ are composed as $\mathcal{L}_h^{[OBA]} = \mathcal{L}_h^A \circ \mathcal{L}_h^B \circ \mathcal{L}_h^O$. Consequently, one numerical update step for this integrator can be defined as,

$$\bar{\mathbf{x}}_{t+h} = \exp\left(\frac{-h\beta\Gamma}{2}\right)\bar{\mathbf{x}}_t + \sqrt{1 - \exp(-h\beta\Gamma)}\epsilon_x \tag{245}$$

$$\bar{\mathbf{m}}_{t+h} = \exp\left(\frac{-h\beta\nu}{2}\right)\bar{\mathbf{m}}_t + \sqrt{M}\sqrt{1 - \exp(-h\beta\nu)}\epsilon_m \tag{246}$$

$$\hat{\mathbf{m}}_{t+h} = \bar{\mathbf{m}}_{t+h} + \frac{h\beta}{2}\left[\bar{\mathbf{x}}_{t+h} + 2\nu\bar{\mathbf{m}}_{t+h} + 2M\nu s_\theta^m(\bar{\mathbf{x}}_{t+h}, \bar{\mathbf{m}}_{t+h}, T - t)\right] \tag{247}$$

$$\hat{\mathbf{x}}_{t+h} = \bar{\mathbf{x}}_{t+h} + \frac{h\beta}{2}\left[2\Gamma\bar{\mathbf{x}}_{t+h} - M^{-1}\hat{\mathbf{m}}_{t+h} + 2\Gamma s_\theta^x(\bar{\mathbf{x}}_{t+h}, \hat{\mathbf{m}}_{t+h}, T - t)\right] \tag{248}$$

where $\epsilon_x, \epsilon_m \sim \mathcal{N}(\mathbf{0}_d, \boldsymbol{I}_d)$. Therefore, one update step for Naive OBA requires **two NFEs**.

### C.3.2 EFFECTS OF CONTROLLING STOCHASTICITY

Similar to Karras et al. (2022), we introduce a parameter $\lambda_s$ in the position space update for $\mathcal{L}_O$ to control the amount of noise injected in the position space. More specifically, we modify the numerical update equations for the Ornstein-Uhlenbeck process in the position space as follows:

$$\bar{\mathbf{x}}_{t+h} = \exp\left(\frac{-h\beta\Gamma}{2}\right)\bar{\mathbf{x}}_t + \sqrt{1 - \exp(-\bar{t}\lambda_s\beta\Gamma)}\epsilon_x, \quad \epsilon_x \sim \mathcal{N}(\mathbf{0}_d, \boldsymbol{I}_d) \tag{249}$$

where $\bar{t} = \frac{(T-t)+(T-t-h)}{2}$, i.e., the mid-point for two consecutive time steps during sampling. Adding a similar parameter in the momentum space leads to unstable sampling. We therefore restrict this adjustment to only the position space.

### C.3.3 REDUCED SPLITTING SCHEMES

We obtain the Reduced Splitting schemes by sharing the score function evaluation between the first consecutive position and momentum updates for all samplers. Additionally, for half-step updates (as in the OBAB scheme), we condition the score function with the timestep embedding of $T - (t + h)$ instead of $T - t$. Moreover, we make the adjustments as described in Appendix C.3.2.

**Reduced OBA**: The numerical updates for this scheme are as follows (the terms in red denote the changes from the Naive OBA scheme),

$$\bar{\mathbf{x}}_{t+h} = \exp\left(\frac{-h\beta\Gamma}{2}\right)\bar{\mathbf{x}}_t + \sqrt{1 - \exp(-\bar{t}\lambda_s\beta\Gamma)}\epsilon_x \tag{250}$$

$$\bar{\mathbf{m}}_{t+h} = \exp\left(\frac{-h\beta\nu}{2}\right)\bar{\mathbf{m}}_t + \sqrt{M}\sqrt{1 - \exp(-h\beta\nu)}\epsilon_m \tag{251}$$

$$\hat{\mathbf{m}}_{t+h} = \bar{\mathbf{m}}_{t+h} + \frac{h\beta}{2}\left[\bar{\mathbf{x}}_{t+h} + 2\nu\bar{\mathbf{m}}_{t+h} + 2M\nu s_\theta^m(\bar{\mathbf{x}}_{t+h}, \bar{\mathbf{m}}_{t+h}, T - t)\right] \tag{252}$$

$$\hat{\mathbf{x}}_{t+h} = \bar{\mathbf{x}}_{t+h} + \frac{h\beta}{2}\left[2\Gamma\bar{\mathbf{x}}_{t+h} - M^{-1}\hat{\mathbf{m}}_{t+h} + 2\Gamma s_\theta^x(\bar{\mathbf{x}}_{t+h}, \bar{\mathbf{m}}_{t+h}, T - t)\right] \tag{253}$$

where $\epsilon_x, \epsilon_m \sim \mathcal{N}(\mathbf{0}_d, \boldsymbol{I}_d)$. It is worth noting that Reduced OBA requires only **one NFE** per update step since a single score evaluation is re-used in both the momentum and the position updates.

**Reduced BAO**: The numerical updates for this scheme are as follows,

$$\bar{\mathbf{m}}_{t+h} = \bar{\mathbf{m}}_t + \frac{h\beta}{2} \left[ \bar{\mathbf{x}}_t + 2\nu\bar{\mathbf{m}}_t + 2M\nu s_\theta^m(\bar{\mathbf{x}}_t, \bar{\mathbf{m}}_t, T-t) \right] \tag{254}$$

$$\bar{\mathbf{x}}_{t+h} = \bar{\mathbf{x}}_t + \frac{h\beta}{2} \left[ 2\Gamma\bar{\mathbf{x}}_t - M^{-1}\bar{\mathbf{m}}_{t+h} + 2\Gamma s_\theta^x(\bar{\mathbf{x}}_t, \bar{\mathbf{m}}_t, T-t) \right] \tag{255}$$

$$\hat{\mathbf{x}}_{t+h} = \exp\left( \frac{-h\beta\Gamma}{2} \right)\bar{\mathbf{x}}_{t+h} + \sqrt{1 - \exp\left(-\bar{t}\lambda_s\beta\Gamma\right)}\boldsymbol{\epsilon}_x \tag{256}$$

$$\hat{\mathbf{m}}_{t+h} = \exp\left( \frac{-h\beta\nu}{2} \right)\bar{\mathbf{m}}_{t+h} + \sqrt{M}\sqrt{1 - \exp\left(-h\beta\nu\right)}\boldsymbol{\epsilon}_m \tag{257}$$

where $\boldsymbol{\epsilon}_x, \boldsymbol{\epsilon}_m \sim \mathcal{N}(\mathbf{0}_d, \boldsymbol{I}_d)$. Similar to the Reduced OBA scheme, Reduced BAO also requires only **one NFE** per update step since a single score evaluation is re-used in both the momentum and the position updates.

**Reduced OBAB**: The numerical updates for this scheme are as follows,

$$\bar{\mathbf{x}}_{t+h} = \exp\left( \frac{-h\beta\Gamma}{2} \right)\bar{\mathbf{x}}_t + \sqrt{1 - \exp\left(-\bar{t}\lambda_s\beta\Gamma\right)}\boldsymbol{\epsilon}_x \tag{258}$$

$$\bar{\mathbf{m}}_{t+h} = \exp\left( \frac{-h\beta\nu}{2} \right)\bar{\mathbf{m}}_t + \sqrt{M}\sqrt{1 - \exp\left(-h\beta\nu\right)}\boldsymbol{\epsilon}_m \tag{259}$$

$$\hat{\mathbf{m}}_{t+h/2} = \bar{\mathbf{m}}_{t+h} + \frac{h\beta}{4} \left[ \bar{\mathbf{x}}_{t+h} + 2\nu\bar{\mathbf{m}}_{t+h} + 2M\nu s_\theta^m(\bar{\mathbf{x}}_{t+h}, \bar{\mathbf{m}}_{t+h}, T-t) \right] \tag{260}$$

$$\hat{\mathbf{x}}_{t+h} = \bar{\mathbf{x}}_{t+h} + \frac{h\beta}{2} \left[ 2\Gamma\bar{\mathbf{x}}_{t+h} - M^{-1}\hat{\mathbf{m}}_{t+h/2} + 2\Gamma s_\theta^x(\bar{\mathbf{x}}_{t+h}, \bar{\mathbf{m}}_{t+h}, T-t) \right] \tag{261}$$

$$\hat{\mathbf{m}}_{t+h} = \hat{\mathbf{m}}_{t+h/2} + \frac{h\beta}{4} \left[ \hat{\mathbf{x}}_{t+h} + 2\nu\hat{\mathbf{m}}_{t+h/2} + 2M\nu s_\theta^m(\hat{\mathbf{x}}_{t+h}, \hat{\mathbf{m}}_{t+h/2}, T-(t+h)) \right] \tag{262}$$

where $\boldsymbol{\epsilon}_x, \boldsymbol{\epsilon}_m \sim \mathcal{N}(\mathbf{0}_d, \boldsymbol{I}_d)$. It is worth noting that, in contrast to the Reduced OBA and BAO schemes, Reduced OBAB requires **two NFE** per update step. This is similar to the Reduced Velocity Verlet (RVV) sampler.

## D    CONJUGATE SPLITTING INTEGRATORS

Here, we highlight relevant update equations for the Conjugate Splitting Samplers discussed in Section. 3.3.

### D.1    DETERMINISTIC CONJUGATE SPLITTING SAMPLERS

The splitting scheme for deterministic splitting samplers discussed in Section 3.2 is specified as follows,

$$\begin{pmatrix} d\bar{\mathbf{x}}_t \\ d\bar{\mathbf{m}}_t \end{pmatrix} = \underbrace{\frac{\beta}{2} \begin{pmatrix} \Gamma\bar{\mathbf{x}}_t - M^{-1}\bar{\mathbf{m}}_t + \Gamma s_\theta^x(\bar{\mathbf{z}}_t, T-t) \\ 0 \end{pmatrix} dt}_{A} + \underbrace{\frac{\beta}{2} \begin{pmatrix} 0 \\ \bar{\mathbf{x}}_t + \nu\bar{\mathbf{m}}_t + M\nu s_\theta^m(\bar{\mathbf{z}}_t, T-t) \end{pmatrix} dt}_{B} \tag{263}$$

**Conjugate Integrators applied to Splitting components.** The Splitting component $A$ in the position space can be simplified as follows,

$$\begin{pmatrix} d\bar{\mathbf{x}}_t \\ d\bar{\mathbf{m}}_t \end{pmatrix} = \frac{\beta}{2} \begin{pmatrix} -\Gamma\bar{\mathbf{x}}_t + M^{-1}\bar{\mathbf{m}}_t - \Gamma s_\theta^x(\bar{\mathbf{z}}_t, T-t) \\ 0 \end{pmatrix} d\bar{t} \tag{264}$$

$$= \frac{\beta}{2} \begin{pmatrix} -\Gamma & M^{-1} \\ 0 & 0 \end{pmatrix} \begin{pmatrix} \bar{\mathbf{x}}_t \\ \bar{\mathbf{m}}_t \end{pmatrix} - \frac{\Gamma\beta}{2} \begin{pmatrix} s_\theta^x(\bar{\mathbf{z}}_t, T-t) \\ 0 \end{pmatrix} d\bar{t} \tag{265}$$

where $\bar{t} = T - t$. Moreover, for any time-dependent matrix $\boldsymbol{C}_t$, we denote $\boldsymbol{C}_t^m = \boldsymbol{m} \circ \boldsymbol{C}_t$, where, $\circ$ denotes the Hadamard product of the mask $\boldsymbol{m} = \begin{pmatrix} 1 & 1 \\ 0 & 0 \end{pmatrix}$ with the matrix $\boldsymbol{C}_t$. Therefore, Eqn. 265

---

**Algorithm 2** *Conjugate Symplectic Euler*

---

**Input:** Trajectory length T, Network function $\epsilon_\theta(.,.)$, number of sampling steps $N$, a monotonically decreasing timestep discretization $\{t_i\}_{i=0}^N$ spanning the interval $(\epsilon, T)$ and choice of $B_t$.
**Output:** $z_\epsilon = (x_\epsilon, m_\epsilon)$

Compute $\{\hat{A}_{t_i}\}_{i=0}^N$ and $\{\hat{\Phi}_{t_i}\}_{i=0}^N$ as in Eqn. 267   $\triangleright$ Pre-compute coefficients
$z_{t_0} \sim p(z_T)$   $\triangleright$ Draw initial samples from the generative prior
**for** $n = 0$ **to** $N - 1$ **do**
  Compute $\epsilon_\theta(x_{t_n}, m_{t_n}, t_n)$ and $s_\theta(x_{t_n}, m_{t_n}, t_n)$   $\triangleright$ Compute score
  $h = (t_{n+1} - t_n)$   $\triangleright$ Time step differential

  $m_{t_{n+1}} = m_{t_n} - \frac{h\beta}{2}\Big[x_{t_n} + \nu m_{t_n} + M\nu s_\theta^m(x_{t_n}, m_{t_n}, t_n)\Big]$   $\triangleright$ Momentum Update

  Construct $\tilde{z}_{t_n} = [x_{t_n}, m_{t_{n+1}}]^\top$
  $d\hat{\Phi}_t = (\hat{\Phi}_{t_{n+1}} - \hat{\Phi}_{t_n})$   $\triangleright$ Phi differential
  $\hat{z}_{t_n} = \hat{A}_{t_n}\tilde{z}_{t_n}$   $\triangleright$ Transform
  $\hat{z}_{t_{n+1}} \leftarrow \hat{z}_{t_n} + h\lambda\hat{A}_{t_n}\mathbf{1}\hat{A}_{t_n}^{-1}\hat{z}_{t_n} + d\hat{\Phi}_t\epsilon_\theta(x_{t_n}, m_{t_n}, t_n)$   $\triangleright$ Update
  $x_{t_{n+1},\_} = \hat{A}_{t_{n+1}}^{-1}\hat{z}_{t_{n+1}}$   $\triangleright$ Project to original space and discard momentum
  Construct $z_{t_{n+1}} = [x_{t_{n+1}}, m_{t_{n+1}}]^\top$
**end for**

---

**Algorithm 3** *Conjugate Velocity Verlet*

---

**Input:** Trajectory length T, Network function $\epsilon_\theta(.,.)$, number of sampling steps $N$, a monotonically decreasing timestep discretization $\{t_i\}_{i=0}^N$ spanning the interval $(\epsilon, T)$ and choice of $B_t$.
**Output:** $z_\epsilon = (x_\epsilon, m_\epsilon)$

Compute $\{\hat{A}_{t_i}\}_{i=0}^N$ and $\{\hat{\Phi}_{t_i}\}_{i=0}^N$ as in Eqn. 267   $\triangleright$ Pre-compute coefficients
$z_{t_0} \sim p(z_T)$   $\triangleright$ Draw initial samples from the generative prior
**for** $n = 0$ **to** $N - 1$ **do**
  Compute $\epsilon_\theta(\epsilon_\theta(x_{t_n}, m_{t_n}, t_n))$ and $s_\theta(x_{t_n}, m_{t_n}, t_n)$   $\triangleright$ Compute score
  $h = (t_{n+1} - t_n)$   $\triangleright$ Time step differential

  $\tilde{m}_{t_{n+1}} = m_{t_n} - \frac{h\beta}{4}\Big[x_{t_n} + \nu m_{t_n} + M\nu s_\theta^m(x_{t_n}, m_{t_n}, t_n)\Big]$   $\triangleright$ Momentum Update

  Construct $\tilde{z}_{t_n} = [x_{t_n}, \tilde{m}_{t_{n+1}}]^\top$
  $d\hat{\Phi}_t = (\hat{\Phi}_{t_{n+1}} - \hat{\Phi}_{t_n})$   $\triangleright$ Phi differential
  $\hat{z}_{t_n} = \hat{A}_{t_n}\tilde{z}_{t_n}$   $\triangleright$ Transform
  $\hat{z}_{t_{n+1}} \leftarrow \hat{z}_{t_n} + h\lambda\hat{A}_{t_n}\mathbf{1}\hat{A}_{t_n}^{-1}\hat{z}_{t_n} + d\hat{\Phi}_t\epsilon_\theta(C_{\text{in}}(t_n)z_{t_n}, C_{\text{noise}}(t_n))$   $\triangleright$ Update
  $x_{t_{n+1},\_} = \hat{A}_{t_{n+1}}^{-1}\hat{z}_{t_{n+1}}$   $\triangleright$ Project to original space and discard momentum
  Construct $z_{t_{n+1}} = [x_{t_{n+1}}, \tilde{m}_{t_{n+1}}]^\top$

  Compute $\epsilon_\theta(x_{t_{n+1}}, \tilde{m}_{t_{n+1}}, t_{n+1})$ and $s_\theta(x_{t_{n+1}}, \tilde{m}_{t_{n+1}}, t_{n+1})$   $\triangleright$ Compute score
  $m_{t_{n+1}} = \tilde{m}_{t_{n+1}} - \frac{h\beta}{4}\Big[x_{t_{n+1}} + \nu\tilde{m}_{t_{n+1}} + M\nu s_\theta^m(x_{t_{n+1}}, \tilde{m}_{t_{n+1}}, t_{n+1})\Big]$   $\triangleright$ Momentum Update
**end for**

---

can be simplified as follows,

$$\begin{pmatrix} d\bar{x}_t \\ d\bar{m}_t \end{pmatrix} = \left( F_t^m\bar{z}_t - \frac{1}{2}G_t^m(G_t^\top)^m\Big[C_{\text{skip}}^m(t)\bar{z}_t + C_{\text{out}}^m(t)\epsilon_\theta(C_{\text{in}}(t)z_t, C_{\text{noise}}(t))\Big]\right) d\bar{t} \quad (266)$$

where $F_t$ and $G_t$ are the drift scaling matrix and the diffusion coefficients, respectively. We can then determine the transformed ODE corresponding to the *masked* ODE in Eqn. 266 and perform numerical integration in the projected space. We use the $\lambda$-DDIM-II as our choice of the conjugate integrator and, therefore, set $B_t = \lambda\mathbf{1}$. The coefficients $A_t$ and $\Phi_t$ are defined as,

$$\hat{A}_t = \exp\left(\int_0^t \lambda\mathbf{1} - F_s^m + \frac{1}{2}G_s^m(G_s^\top)^m C_{\text{skip}}^m(s)ds\right), \quad \hat{\Phi}_t = -\int_0^t \frac{1}{2}\hat{A}_s G_s^m(G_s^\top)^m C_{\text{out}}^m(s)ds,$$
$$(267)$$

---

**Algorithm 4** *Conjugate OBA*

---

**Input:** Trajectory length T, Network function $\epsilon_{\boldsymbol{\theta}}(.,.)$, number of sampling steps $N$, a monotonically decreasing timestep discretization $\{t_i\}_{i=0}^N$ spanning the interval $(\epsilon, \text{T})$ and choice of $\boldsymbol{B}_t$.
**Output:** $\mathbf{z}_\epsilon = (\mathbf{x}_\epsilon, \mathbf{m}_\epsilon)$

Compute $\{\hat{\boldsymbol{A}}_{t_i}\}_{i=0}^N$ and $\{\hat{\boldsymbol{\Phi}}_{t_i}\}_{i=0}^N$ as in Eqn. 272                                       ▷ Pre-compute coefficients
$\mathbf{z}_{t_0} \sim p(\mathbf{z}_T)$                                                                                              ▷ Draw initial samples from the generative prior
**for** $n = 0$ **to** $N - 1$ **do**
    $h = (t_{n+1} - t_n)$                                                                                            ▷ Time step differential

    $t' = (t_n + t_{n+1})/2$
    $\tilde{\mathbf{x}}_{t_n} = \exp\left(\frac{h\beta\Gamma}{2}\right)\mathbf{x}_{t_n} + \sqrt{1 - \exp\left(-t'\lambda_s\beta\Gamma\right)}\epsilon_x$             ▷ OU-Update (Position)
    $\tilde{\mathbf{m}}_{t_n} = \exp\left(\frac{h\beta\nu}{2}\right)\mathbf{m}_{t_n} + \sqrt{M}\sqrt{1 - \exp\left(h\beta\nu\right)}\epsilon_m$               ▷ OU-Update (Momentum)

    Compute $\epsilon_{\boldsymbol{\theta}}(\tilde{\mathbf{x}}_{t_n}, \tilde{\mathbf{m}}_{t_n}, t_n)$ and $s_{\boldsymbol{\theta}}(\tilde{\mathbf{x}}_{t_n}, \tilde{\mathbf{m}}_{t_n}, t_n)$                 ▷ Compute score
    $\mathbf{m}_{t_{n+1}} = \tilde{\mathbf{m}}_{t_n} - \frac{h\beta}{2}\left[\tilde{\mathbf{x}}_{t_n} + 2\nu\tilde{\mathbf{m}}_{t_n} + 2M\nu s_{\boldsymbol{\theta}}^m(\tilde{\mathbf{x}}_{t_n}, \tilde{\mathbf{m}}_{t_n}, t_n)\right]$      ▷ Momentum Update

    Construct $\tilde{\mathbf{z}}_{t_n} = [\tilde{\mathbf{x}}_{t_n}, \mathbf{m}_{t_{n+1}}]^\top$
    $d\hat{\boldsymbol{\Phi}}_t = (\hat{\boldsymbol{\Phi}}_{t_{n+1}} - \hat{\boldsymbol{\Phi}}_{t_n})$                                                               ▷ Phi differential
    $\hat{\mathbf{z}}_{t_n} = \hat{\boldsymbol{A}}_{t_n}\tilde{\mathbf{z}}_{t_n}$                                                                    ▷ Transform
    $\hat{\mathbf{z}}_{t_{n+1}} \leftarrow \hat{\mathbf{z}}_{t_n} + h\lambda\hat{\boldsymbol{A}}_{t_n}\mathbf{1}\hat{\boldsymbol{A}}_{t_n}^{-1}\hat{\mathbf{z}}_{t_n} + d\hat{\boldsymbol{\Phi}}_t\epsilon_{\boldsymbol{\theta}}(\tilde{\mathbf{x}}_{t_n}, \tilde{\mathbf{m}}_{t_n}, t_n)$      ▷ Update
    $\mathbf{x}_{t_{n+1}}, \_ = \boldsymbol{A}_{t_{n+1}}^{-1}\hat{\mathbf{z}}_{t_{n+1}}$                                             ▷ Project to original space and discard momentum
    Construct $\mathbf{z}_{t_{n+1}} = [\mathbf{x}_{t_{n+1}}, \mathbf{m}_{t_{n+1}}]^\top$
**end for**

---

Based on this analysis, we provide the numerical update rules for the CSE and CVV samplers in Algorithms 2 and 3, respectively.

## D.2 STOCHASTIC CONJUGATE SPLITTING SAMPLERS

We split the Reverse Diffusion SDE for PSLD using the following splitting scheme.

$$\begin{pmatrix} d\bar{\mathbf{x}}_t \\ d\bar{\mathbf{m}}_t \end{pmatrix} = \underbrace{\frac{\beta}{2}\begin{pmatrix} 2\Gamma\bar{\mathbf{x}}_t - M^{-1}\bar{\mathbf{m}}_t + 2\Gamma s_{\boldsymbol{\theta}}^x(\bar{\mathbf{z}}_t, t) \\ 0 \end{pmatrix}dt}_{A} + O + \underbrace{\frac{\beta}{2}\begin{pmatrix} 0 \\ \bar{\mathbf{x}}_t + 2\nu\bar{\mathbf{m}}_t + 2M\nu s_{\boldsymbol{\theta}}^m(\bar{\mathbf{z}}_t, t) \end{pmatrix}dt}_{B}$$

(268)

Therefore, the splitting component corresponding to the position space is,

$$\begin{pmatrix} d\bar{\mathbf{x}}_t \\ d\bar{\mathbf{m}}_t \end{pmatrix} = \frac{\beta}{2}\begin{pmatrix} -2\Gamma\bar{\mathbf{x}}_t + M^{-1}\bar{\mathbf{m}}_t - 2\Gamma s_{\boldsymbol{\theta}}^x(\bar{\mathbf{z}}_t, T - t) \\ 0 \end{pmatrix}d\bar{t} \tag{269}$$

$$= \frac{\beta}{2}\begin{pmatrix} -2\Gamma & M^{-1} \\ 0 & 0 \end{pmatrix}\begin{pmatrix} \bar{\mathbf{x}}_t \\ \bar{\mathbf{m}}_t \end{pmatrix} - \Gamma\beta\begin{pmatrix} s_{\boldsymbol{\theta}}^x(\bar{\mathbf{z}}_t, T - t) \\ 0 \end{pmatrix}d\bar{t} \tag{270}$$

where $\bar{t} = T - t$. Eqn. 270 can be further simplified as follows,

$$d\bar{\mathbf{z}}_t = \left(\tilde{\boldsymbol{F}}_t\bar{\mathbf{z}}_t - \tilde{\boldsymbol{G}}_t\tilde{\boldsymbol{G}}_t^\top\left[C_{\text{skip}}(t)\bar{\mathbf{z}}_t + C_{\text{out}}(t)\epsilon_{\boldsymbol{\theta}}(C_{\text{in}}(t)\mathbf{z}_t, C_{\text{noise}}(t))\right]\right)d\bar{t} \tag{271}$$

where $\tilde{\boldsymbol{F}}_t = \frac{\beta}{2}\begin{pmatrix} -2\Gamma & M^{-1} \\ 0 & 0 \end{pmatrix}$ and $\tilde{\boldsymbol{G}}_t = \begin{pmatrix} \sqrt{\Gamma\beta} & 0 \\ 0 & 0 \end{pmatrix}$. We use the $\lambda$-DDIM-II as our choice of the conjugate integrator and, therefore, set $\boldsymbol{B}_t = \lambda\mathbf{1}$. The coefficients $\hat{\boldsymbol{A}}_t$ and $\hat{\boldsymbol{\Phi}}_t$ are defined as,

$$\hat{\boldsymbol{A}}_t = \exp\left(\int_0^t \lambda\mathbf{1} - \tilde{\boldsymbol{F}}_s + \tilde{\boldsymbol{G}}_s\tilde{\boldsymbol{G}}_s^\top C_{\text{skip}}^m(s)ds\right), \quad \hat{\boldsymbol{\Phi}}_t = -\int_0^t \hat{\boldsymbol{A}}_s\tilde{\boldsymbol{G}}_s\tilde{\boldsymbol{G}}_s^\top C_{\text{out}}^m(s)ds, \tag{272}$$

where, $C_{\text{skip}}^m(t) = \boldsymbol{m} \circ C_{\text{skip}}(t)$ and $C_{\text{out}}^m(t) = \boldsymbol{m} \circ C_{\text{out}}(t)$ and $\boldsymbol{m} = \begin{pmatrix} 1 & 1 \\ 0 & 0 \end{pmatrix}$ Based on this analysis, we present a complete analysis for the Conjugate OBA sampler in Algorithm 4.

# E  IMPLEMENTATION DETAILS

Here, we present complete implementation details for all the samplers presented in this work.

## E.1  DATASETS AND PREPROCESSING

We use the CIFAR-10 (Krizhevsky, 2009) (50k images), CelebA-64 (downsampled to 64 x 64 resolution, $\approx$ 200k images) (Liu et al., 2015) and the AFHQv2-64 (Choi et al., 2020) (downsampled to 64 x 64 resolution, $\approx$ 15k images) datasets for both quantitative and qualitative analysis. We use the AFHQv2 dataset (downsampled to the 128 x 128 resolution) only for qualitative analysis. During training, all datasets are preprocessed to a numerical range of [-1, 1]. Following prior work, we use random horizontal flips to train all new models across datasets as a data augmentation strategy. During inference, we re-scale all generated samples between the range [0, 1].

## E.2  PRE-TRAINED MODELS

For all ablation results in Section 3 in the main text, we use pre-trained PSLD (Pandey & Mandt, 2023) models for CIFAR-10 with SDE hyperparameters $\Gamma = 0.01$, $\nu = 4.01$ and $\beta = 8.0$. The resulting model consists of approximately 97M parameters. For more details on the score network architecture, refer to Pandey & Mandt (2023). Moreover, pre-trained models from PSLD correspond to the following choices of the design parameters in the score parameterization defined in Eqn. 5,

$$C_{\text{skip}}(t) = \mathbf{0}, \quad C_{\text{out}}(t) = -L_t^{-\top}, \quad C_{\text{in}}(t) = \mathbf{I}, \quad C_{\text{noise}}(t) = t. \tag{273}$$

where $L_t^{-\top}$ is the transposed-inverse of the Cholesky decomposition of the covariance matrix $\Sigma_t$ of the perturbation kernel in PSLD. Most comparison baselines in Section 5 (like DEIS (Zhang & Chen, 2023) and DPM-Solver (Lu et al., 2022)) use the VP-SDE (deep) model, which is around 108M parameters in size. Therefore, our model sizes are comparable with other baselines, making our comparisons fair.

## E.3  SCORE NETWORK PRECONDITIONING

For the score network parameterization discussed in Eqn. 5, we choose,

$$C_{\text{skip}}(t) = \text{diag}(\bar{\Sigma}_t), \quad C_{\text{out}}(t) = -L_t^{-\top}, \quad C_{\text{in}}(t) = \mathbf{I}, \quad C_{\text{noise}}(t) = t.$$

where $L_t$ is the Cholesky factorization of the variance $\Sigma_t$ of the perturbation kernel in PSLD. Similarly, $\bar{\Sigma}_t$ is the variance of the perturbation kernel in PSLD with initial variance $\bar{\Sigma}_{xx}^0 = \sigma_0^2 \mathbf{I}$, $\bar{\Sigma}_{xm}^0 = \mathbf{0}, \bar{\Sigma}_{mm}^0 = M\gamma\mathbf{I}$. For optimal sample quality, we set the weighting scheme $\lambda(t) = \frac{1}{\|C_{\text{out}}\|_2^2}$. We set $\sigma_0^2 = 0.25$ for all experimental analysis. Since this requires newly trained PSLD models, we highlight our score network architectures and training configuration next.

**Score-Network architecture.** Table 2 illustrates our score model architectures for different datasets. We use the NCSN++ architecture (Song et al., 2020) for all newly trained models.

**SDE Hyperparameters**: Similar to Pandey & Mandt (2023), we set $\beta = 8.0$, $M^{-1} = 4$ and $\gamma = 0.04$ for all datasets. For CIFAR-10, we set $\Gamma = 0.01$ and $\nu = 4.01$, corresponding to the best settings in PSLD. Similarly, for CelebA-64 and AFHQv2-64 datasets, we set $\Gamma = 0.005$ and $\nu = 4.005$. Similar to Pandey & Mandt (2023), we add a stabilizing numerical epsilon value of $1e^{-9}$ in the diagonal entries of the Cholesky decomposition of $\Sigma_t$ when sampling from the perturbation kernel $p(\mathbf{z}_t|\mathbf{x}_0)$ during training.

**Training** Table 3 summarizes the different training hyperparameters across datasets. We use the Hybrid Score Matching (HSM) objective during training.

## E.4  EVALUATION

We report FID (Heusel et al., 2017) scores on 50k samples for to assess sample quality. We use the Number of Function Evaluations (NFEs) for assessing sampling efficiency.

**Timestep Selection during Sampling**: We use quadratic striding for timestep discretization proposed in Dockhorn et al. (2022b) during sampling, which ensures more number of score function

| Hyperparameter | CIFAR-10 | CelebA-64 | AFHQv2-64 |
|---|---|---|---|
| Base channels | 128 | 128 | 128 |
| Channel multiplier | [2,2,2] | [1,2,2,2] | [1,2,2,2] |
| # Residual blocks | 8 | 4 | 4 |
| Non-Linearity | Swish | Swish | Swish |
| Attention resolution | [16] | [16] | [16] |
| # Attention heads | 1 | 1 | 1 |
| Dropout | 0.15 | 0.1 | 0.25 |
| FIR (Zhang, 2019) | True | True | True |
| FIR kernel | [1,3,3,1] | [1,3,3,1] | [1,3,3,1] |
| Progressive Input | Residual | Residual | Residual |
| Progressive Combine | Sum | Sum | Sum |
| Embedding type | Fourier | Fourier | Fourier |
| Sigma scaling | False | False | False |
| Model size | 97M | 62M | 62M |

Table 2: Score Network hyperparameters for training Preconditioned PSLD models. $\sigma_0^2$ is set to 0.25 for all datasets

| | CIFAR-10 | CelebA-64 | AFHQv2 |
|---|---|---|---|
| Random Seed | 0 | 0 | 0 |
| # iterations | 1.2M | 1.2M | 400k |
| Optimizer | Adam | Adam | Adam |
| Grad Clip. cutoff | 1.0 | 1.0 | 1.0 |
| Learning rate (LR) | 2e-4 | 2e-4 | 2e-4 |
| LR Warmup steps | 5000 | 5000 | 5000 |
| FP16 | False | False | False |
| EMA Rate | 0.9998 | 0.9998 | 0.9998 |
| Effective Batch size | 128 | 128 | 128 |
| # GPUs | 8 | 8 | 8 |
| Train eps cutoff | 1e-5 | 1e-5 | 1e-5 |

Table 3: Training hyperparameters

evaluations in the lower timestep regime (i.e., $t$, which is close to the data). This kind of timestep selection is particularly useful when the NFE budget is limited. We also explored the timestep discretization proposed in Karras et al. (2022) but noticed a degradation in sample quality.

**Last-Step Denoising**: It is common to add an Euler-based denoising step from a cutoff $\epsilon$ to zero to optimize for sample quality (Song et al., 2020; Dockhorn et al., 2022b; Jolicoeur-Martineau et al., 2021b) at the expense of another sampling step. For deterministic samplers presented in this work, we omit this heuristic due to observed degradation in sample quality. However, for stochastic samplers, we find that using last-step denoising leads to improvements in sample quality (especially when adjusting the amount of stochasticity as discussed in Appendix C.3.2). Formally, we perform the following update as a last denoising step for stochastic samplers:

$$\begin{pmatrix} \mathbf{x}_0 \\ \mathbf{m}_0 \end{pmatrix} = \begin{pmatrix} \mathbf{x}_\epsilon \\ \mathbf{m}_\epsilon \end{pmatrix} + \frac{\beta_t \epsilon}{2} \begin{pmatrix} \Gamma \mathbf{x}_\epsilon - M^{-1}\mathbf{m}_\epsilon + 2\Gamma \boldsymbol{s}_\theta(\mathbf{z}_\epsilon, \epsilon)|_{0:d} \\ \mathbf{x}_\epsilon + \nu \mathbf{m}_\epsilon + 2M\nu \boldsymbol{s}_\theta(\mathbf{z}_\epsilon, \epsilon)|_{d:2d} \end{pmatrix} \tag{274}$$

Similar to PSLD, we set $\epsilon = 1e - 3$ during sampling for all experiments. Though recent works (Lu et al., 2022; Zhang & Chen, 2023) have found lower cutoffs to work better for a certain NFE budget, we leave this exploration in the context of PSLD to future work.

**Evaluation Metrics:** Unless specified otherwise, we report the FID (Heusel et al., 2017) score on 50k samples for assessing sample quality. Similarly, we use the network function evaluations (NFE) to assess sampling efficiency. In practice, we use the `torch-fidelity`(Obukhov et al., 2020) package for computing all FID reported in this work.

| | NFE (FID@50k ↓) | | | | | | | |
|---|---|---|---|---|---|---|---|---|
| Method | 50 | 70 | 100 | 150 | 200 | 250 | 500 | 1000 |
| Euler | 431.74 | 397.51 | 330.18 | 233.28 | 163.13 | 110.68 | 33.93 | 11.54 |
| $\lambda$-DDIM ($\boldsymbol{B}_t = \boldsymbol{0}$) | **48.55** | **11.49** | **4.81** | **3.53** | **3.31** | **3.19** | **3.04** | **3.01** |

Table 4: Extended results for Fig. 2a. $\lambda$-DDIM outperforms baseline Euler when applied to the PSLD Prob. Flow ODE. The choice of $\boldsymbol{B}_t = \boldsymbol{0}$ corresponds to the exponential integrators proposed in Zhang & Chen (2023); Zhang et al. (2022). In this case, Euler fails to generate high-quality samples even with a high compute budget of 1000 NFEs. Values in **bold** indicate the best FID scores for that column.

| NFE | $\lambda$-DDIM ($\boldsymbol{B}_t = \boldsymbol{0}$) | $\lambda$-DDIM-I ($\boldsymbol{B}_t = \lambda\boldsymbol{I}$) | | $\lambda$-DDIM-II ($\boldsymbol{B}_t = \lambda\boldsymbol{1}$) | |
|---|---|---|---|---|---|
| | FID@50k ($\downarrow$) | FID@50k ($\downarrow$) | $\lambda$ | FID@50k ($\downarrow$) | $\lambda$ |
| 30 | 311.08 | 23.53 | -0.0038 | **13.6** | 0.59 |
| 50 | 48.55 | 5.54 | -0.0016 | **5.04** | 0.46 |
| 70 | 11.49 | 4.41 | -0.0009 | **4.26** | 0.35 |
| 100 | 4.81 | 3.76 | -0.0004 | **3.71** | 0.21 |
| 150 | 3.53 | 3.49 | -0.0002 | **3.46** | 0.12 |
| 200 | 3.31 | 3.32 | -0.00008 | **3.28** | 0.06 |
| 250 | 3.19 | 3.21 | -0.00004 | **3.19** | 0.02 |

Table 5: Extended results for Figs. 2b,2c. Comparison between different choices of $\boldsymbol{B}_t$ for the proposed $\lambda$-DDIM sampler. $\lambda$-DDIM with non-zero choices of $\boldsymbol{B}_t$ outperforms baseline choice with $\boldsymbol{B}_t = \boldsymbol{0}$ which suggests that the latter choice can be sub-optimal in certain scenarios. Most gains in sample quality using a non-zero $\boldsymbol{B}_t$ are observed at low sampling budgets (NFE < 70). Values in **bold** indicate the best among the three methods for a particular sampling budget.

| $\lambda$ | 0.7 | 0.6 | 0.5 | **0.46** | 0.4 | 0.3 | 0.2 | 0.1 | 0 |
|---|---|---|---|---|---|---|---|---|---|
| FID@50k $\downarrow$ | 21.51 | 10.68 | 5.53 | **5.04** | 5.77 | 10.5 | 19.54 | 32.61 | 48.55 |

Table 6: Impact of the magnitude of $\lambda$ on CIFAR-10 sample quality for a fixed NFE=50 steps for $\lambda$-DDIM-II. Entries in **bold** indicate the best FID scores and the corresponding $\lambda$ value. Interestingly, increasing $\lambda$ improves sample quality significantly compared to $\lambda = 0$. However, too much increase in $\lambda$ leads to significant degradation in sample quality.

# F  EXTENDED RESULTS

## F.1  EXTENDED RESULTS FOR SECTION 3.1: CONJUGATE INTEGRATORS

We include extended results corresponding to Figs. 2a in Table 4 and for Figs. 2b, 2c in Table 5, respectively.

**Impact of varying $\lambda$ on sample quality.** Additionally, we illustrate the impact of varying $\lambda$ on sample quality for a fixed NFE=50 for $\lambda$-DDIM-II in Table 6. Increasing the value of $\lambda$ leads to significant improvements in sample quality. However, excessively increasing $\lambda$ leads to degraded sample quality. This observation empirically supports our theoretical results in Theorem 2.

## F.2  EXTENDED RESULTS FOR SECTION 3.2: SPLITTING INTEGRATORS

We include extended results corresponding to Figs. 3a, 3b in Table 7 and for Fig. 3c in Table 8, respectively.

**Comparison between different Stochastic Reduced Splitting schemes.** Table 9 compares the performance of different reduced splitting schemes for stochastic sampling.

**Impact of varying $\lambda_s$ on stochastic sampling.** Additionally, we illustrate the impact of varying the parameter $\lambda_s$ on sample quality in the context of the Reduced OBA sampler (See Table 10).

| NFE | Prob Flow ODE | NSE | RSE | NVV | RVV |
|---|---|---|---|---|---|
| 50 | 431.74 | 132.45 | 23.5 | 69.06 | **14.19** |
| 70 | 397.51 | 63.69 | 10.03 | 31.54 | **5.72** |
| 100 | 330.18 | 23.47 | 5.31 | 14.49 | **3.41** |
| 150 | 233.28 | 8.85 | 3.54 | 7.51 | **2.8** |
| 200 | 163.13 | 5.44 | 3.1 | 5.53 | **2.7** |
| 250 | 110.68 | 4.16 | 2.98 | 4.68 | **2.71** |
| 500 | 33.93 | 3.05 | 2.88 | 3.56 | **2.79** |
| 1000 | 11.54 | 2.92 | 2.89 | 3.2 | **2.86** |

Table 7: Extended Results for Figs. 3a, 3b. Comparison between Euler, Naive, and Reduced Splitting samplers applied to the PSLD ODE. Naive schemes improve significantly over Euler, indicating the benefits of splitting. Adjusted schemes improve significantly over naive splitting samplers, highlighting the benefit of our proposed modifications to naive schemes. Values in **bold** highlight the best-performing sampler among all comparison baselines. FID ↓ reported on 50k samples.

| NFE | EM SDE | Naive OBA | Reduced OBA | Reduced OBA ($+\lambda_s$) |
|---|---|---|---|---|
| 50 | 30.81 | 36.87 | 19.96 | **2.76 (1.16)** |
| 70 | 15.63 | 24.23 | 12.71 | **2.51 (0.66)** |
| 100 | 7.83 | 15.18 | 7.68 | **2.42 (0.37)** |
| 150 | 4.26 | 9.68 | 5.21 | **2.40 (0.2)** |
| 200 | 3.27 | 7.09 | 4.06 | **2.38 (0.13)** |
| 250 | 2.75 | 5.56 | 3.63 | **2.40 (0.1)** |
| 500 | 2.3 | 3.41 | 2.74 | - |
| 1000 | 2.27 | 2.76 | 2.45 | - |

Table 8: Extended Results for Figs. 3c. Comparison between EM, Naive, and Reduced OBA samplers applied to the PSLD Reverse SDE. Adjusted schemes combined with the tuned parameter $\lambda_s$ improve stochastic sampling performance significantly. Values in **bold** highlight the best-performing sampler among all comparison baselines. FID ↓ reported on 50k samples.

| NFE | RBAO | | ROBAB | | ROBA | |
|---|---|---|---|---|---|---|
| | $(+\lambda_s)$ | $(-\lambda_s)$ | $(+\lambda_s)$ | $(-\lambda_s)$ | $(+\lambda_s)$ | $(-\lambda_s)$ |
| 30 | 7.83 (1.18) | 26.88 | 21.60 (0.24) | 22.09 | **4.03 (2.72)** | 39.51 |
| 50 | 3.33 (0.7) | 12.96 | 6.86 (0.2) | 6.01 | **2.76 (1.16)** | 19.96 |
| 70 | 2.59 (0.44) | 8.2 | 4.66 (0.16) | 3.6 | **2.51 (0.66)** | 12.71 |
| 100 | 2.65 (0.3) | 5.31 | 3.54 (0.14) | 2.73 | **2.36 (0.37)** | 7.68 |
| 150 | 2.60 (0.18) | 3.87 | 2.96 (0.12) | 2.44 | **2.40 (0.2)** | 5.21 |
| 200 | 2.43 (0.1) | 3.26 | 2.67 (0.1) | **2.27** | 2.38 (0.13) | 4.06 |

Table 9: Comparison between Reduced OBA, BAO, and OBAB schemes. Reduced OBA (with $\lambda_s$ performs the best among all schemes. Values in **bold** highlight the best-performing sampler among all comparison baselines for a given NFE budget. FID ↓ reported on 50k samples. Values in (.) indicate the corresponding $\lambda_s$ for a sampler at a given compute budget.

Increasing the value of $\lambda_s$ leads to significant improvements in sample quality. However, a large $\lambda_s$ degrades sample quality significantly.

## F.3 EXTENDED RESULTS FOR SECTION 3.3: CONJUGATE SPLITTING INTEGRATORS

We include extended results corresponding to Fig. 4a in Table 11 and for Fig. 4b in Table 12.

## F.4 EXTENDED RESULTS: IMPACT OF PRECONDITIONING

We include extended results corresponding to Figs. 4c, 4d in Table 13.

| $\lambda_s$ | 0.1 | 0.4 | 0.8 | 1.0 | **1.16** | 1.2 | 1.4 | 1.6 |
|---|---|---|---|---|---|---|---|---|
| FID@50k ↓ | 24.68 | 14.34 | 5.57 | 3.29 | **2.76** | 2.82 | 4.21 | 7.07 |

Table 10: Impact of the magnitude of $\lambda$ on CIFAR-10 sample quality for a fixed NFE=50 steps for $\lambda$-DDIM-II. Entries in **bold** indicate the best FID scores and the corresponding $\lambda$ value. Interestingly, increasing $\lambda$ improves sample quality significantly compared to $\lambda = 0$. However, too much increase in $\lambda$ leads to significant degradation in sample quality.

| | RVV | RSE | CVV | | CSE | |
|---|---|---|---|---|---|---|
| NFE | FID@50k ↓ | FID@50k↓ | FID@50k ↓ | $\lambda$ | FID@50k ↓ | $\lambda$ |
| 30 | 89.86 | 94.21 | **7.23** | -0.41 | 7.38 | 1.38 |
| 40 | 31.78 | 44.3 | **4.21** | -0.3 | 4.95 | 1.35 |
| 50 | 14.19 | 23.5 | **3.21** | -0.25 | 3.92 | 1.33 |
| 60 | 8.22 | 14.46 | **2.73** | -0.21 | 3.38 | 1.33 |
| 70 | 5.72 | 10.03 | **2.44** | -0.2 | 3.07 | 1.31 |
| 80 | 4.44 | 7.64 | **2.27** | -0.17 | 2.87 | 1.3 |
| 90 | 3.78 | 6.21 | **2.18** | -0.16 | 2.76 | 1.27 |
| 100 | 3.41 | 5.31 | **2.11** | -0.14 | 2.68 | 1.25 |

Table 11: Extended Results for Fig. 4a. Comparison between Reduced Splitting samplers and Conjugate Splitting samplers applied to the PSLD Prob. flow ODE. Conjugate Splitting samplers largely outperform their reduced counterparts by a significant margin.

| | Reduced OBA | | Conjugate OBA | | |
|---|---|---|---|---|---|
| NFE | FID@50k ↓ | $\lambda_s$ | FID@50k↓ | $\lambda$ | $\lambda_s$ |
| 30 | **4.03** | 2.72 | 4.4 | -0.3 | 2.72 |
| 40 | **3.11** | 1.7 | 3.34 | -0.2 | 1.7 |
| 50 | **2.76** | 1.16 | 2.94 | -0.1 | 1.16 |
| 60 | **2.62** | 0.84 | 2.8 | -0.1 | 0.84 |
| 70 | **2.51** | 0.66 | 2.64 | -0.1 | 0.66 |
| 80 | **2.47** | 0.53 | 2.55 | -0.1 | 0.53 |
| 90 | **2.44** | 0.43 | 2.55 | -0.1 | 0.43 |
| 100 | **2.36** | 0.37 | 2.49 | -0.1 | 0.37 |

Table 12: Extended Results for Fig. 4b. Comparison between Reduced OBA and Conjugate OBA stochastic samplers. Both samplers share churn values for a given sampling budget. For CIFAR-10, Conjugate OBA slightly degrades sample quality over Reduced OBA.

## F.5    EXTENDED RESULTS FOR SECTION 4: STATE-OF-THE-ART RESULTS

We include extended results corresponding to Fig. 5 in Tables 13, 15, 16 for CIFAR-10, CelebA-64 and the AFHQv2-64 datasets, respectively. We include qualitative samples from our samplers used for state-of-the-art comparisons in Figs. 6-11

| | CSPS-D (+Pre.) | | CSPS-D | | SPS-S (+Pre.) | | SPS-S | |
|---|---|---|---|---|---|---|---|---|
| NFE | FID@50k ↓ | $\lambda$ | FID@50k ↓ | $\lambda$ | FID@50k ↓ | $\lambda_s$ | FID@50k ↓ | $\lambda_s$ |
| 30 | 7.57 | -0.42 | **7.23** | -0.41 | **3.89** | 2.65 | 4.03 | 2.72 |
| 40 | **3.26** | -0.31 | 4.21 | -0.3 | **3.05** | 1.7 | 3.11 | 1.7 |
| 50 | **2.65** | -0.25 | 3.21 | -0.25 | **2.74** | 1.15 | 2.76 | 1.16 |
| 60 | **2.42** | -0.22 | 2.73 | -0.21 | **2.58** | 0.86 | 2.62 | 0.84 |
| 70 | **2.34** | -0.2 | 2.44 | -0.2 | 2.54 | 0.66 | **2.51** | 0.66 |
| 80 | 2.3 | -0.18 | **2.27** | -0.17 | 2.52 | 0.53 | **2.47** | 0.53 |
| 90 | 2.23 | -0.16 | **2.18** | -0.16 | 2.5 | 0.45 | **2.44** | 0.43 |
| 100 | 2.24 | -0.14 | **2.11** | -0.14 | 2.47 | 0.38 | **2.36** | 0.37 |

Table 13: Extended Results for Figs. 4c, 4d. Impact of score network preconditioning on sampler performance. Preconditioning improves sample quality for a low sampling budget but slightly degrades sample quality for a higher budget. Values in **bold** indicate the best-performing sampler with/without preconditioning.

| | Ablation | Description | Type | NPU | FID@50k ↓ (NFE=50) | FID@50k ↓ (NFE=100) |
|---|---|---|---|---|---|---|
| Conjugate (Sec. 3.1) | [C1] $\lambda$-DDIM-I | Conjugate Integrator with choice $B_t = I$ | D | 1 | 5.54 | 3.76 |
| | [C2] $\lambda$-DDIM-II | Conjugate Integrator with choice $B_t = 1$ | D | 1 | 5.04 | 3.71 |
| Splitting (Sec 3.2) | [S1] NSE | Naive Symplectic Euler | D | 2 | 132.45 | 23.47 |
| | [S2] NVV | Naive Velocity Verlet | D | 3 | 69.06 | 14.49 |
| | [S3] RSE | Reduced Symplectic Euler ([S1] + adjustments) | D | 1 | 23.5 | 5.31 |
| | [S4] RVV | Reduced Velocity Verlet ([S2] + adjustments) | D | 2 | 14.19 | 3.41 |
| | [S5] NOBA | Naive OBA | S | 2 | 36.87 | 15.18 |
| | [S6] ROBA | Reduced OBA ([S5] + adjustments) | S | 1 | 2.76 | 2.36 |
| Conjugate Splitting (Sec 3.3) | [CS1] CSE | Conjugate Symplectic Euler ([S3] + [C2]) | D | 1 | 3.92 | 2.68 |
| | [CS2] CVV | Conjugate Velocity Verlet ([S4] + [C2]) | D | 2 | 3.21 | 2.11 |
| | [CS3] COBA | Conjugate OBA ([S6] + [C2]) | S | 1 | 2.94 | 2.49 |

Table 14: Overview of our ablation samplers on the CIFAR-10 dataset for PSLD diffusion. NPU: NFE per numerical update, D: Deterministic, S: Stochastic

| | CSPS-D (+Pre.) | | CSPS-S (+Pre.) | | SPS-S (+Pre.) | |
|---|---|---|---|---|---|---|
| NFE | FID@50k ↓ | $\lambda$ | FID@50k ↓ | $\lambda_s$ | FID@50k ↓ | $\lambda_s$ |
| 30 | 25.75 | -0.5 | 6.16 | 0.85 | 6.32 | 3.92 |
| 40 | 7.97 | -0.4 | 3.94 | 0.8 | 4.56 | 2.6 |
| 50 | 4.41 | -0.33 | 3.32 | 0.7 | 3.88 | 1.8 |
| 70 | 2.75 | -0.23 | 2.81 | 0.5 | 3.06 | 1 |
| 100 | 2.25 | -0.13 | 2.6 | 0.3 | 2.64 | 0.55 |

Table 15: State-of-the-art results for CelebA-64.

| | CSPS-D (+Pre.) | | CSPS-S (+Pre.) | | SPS-S (+Pre.) | |
|---|---|---|---|---|---|---|
| NFE | FID@50k ↓ | $\lambda$ | FID@50k ↓ | $\lambda_s$ | FID@50k ↓ | $\lambda_s$ |
| 30 | 9.83 | -0.15 | 3.59 | 2.8 | 6.17 | 1.5 |
| 40 | 5.22 | -0.1 | 3.38 | 2.8 | 5.37 | 1.35 |
| 50 | 3.63 | -0.07 | 3.1 | 2 | 4.36 | 1.15 |
| 70 | 2.7 | -0.04 | 2.83 | 1 | 3.31 | 0.8 |
| 100 | 2.39 | -0.02 | 2.61 | 0.2 | 2.73 | 0.5 |

Table 16: State-of-the-art results for AFHQv2-64.

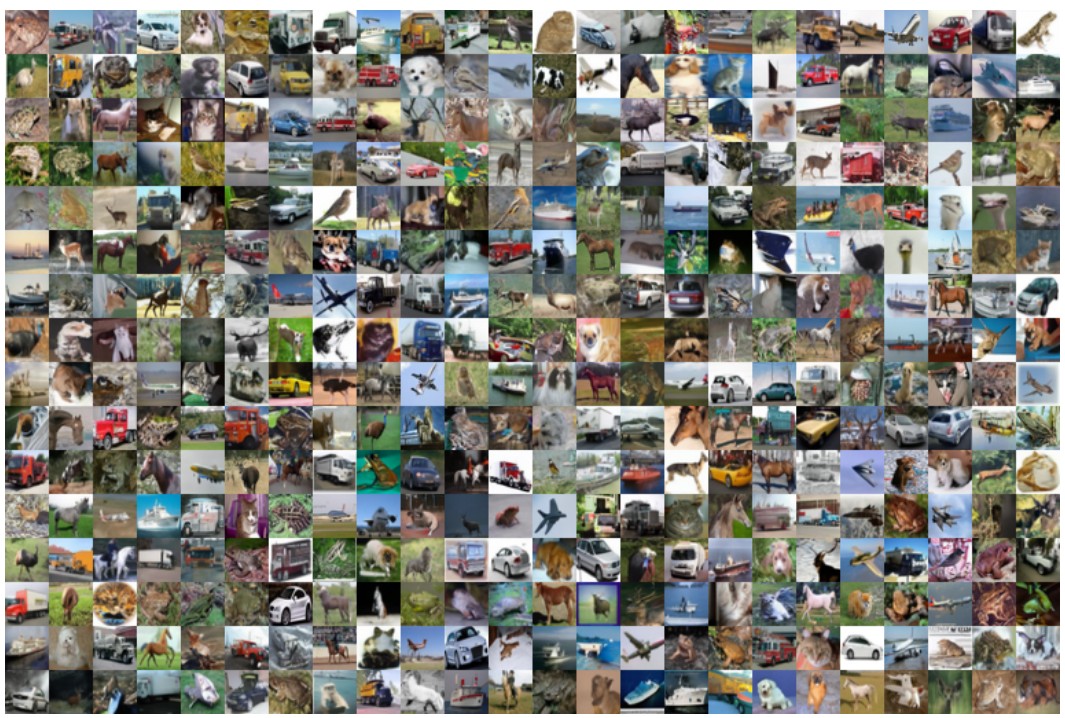

FID = 2.65 (NFE=50),  Sampler: CSPS-D (+Pre.)

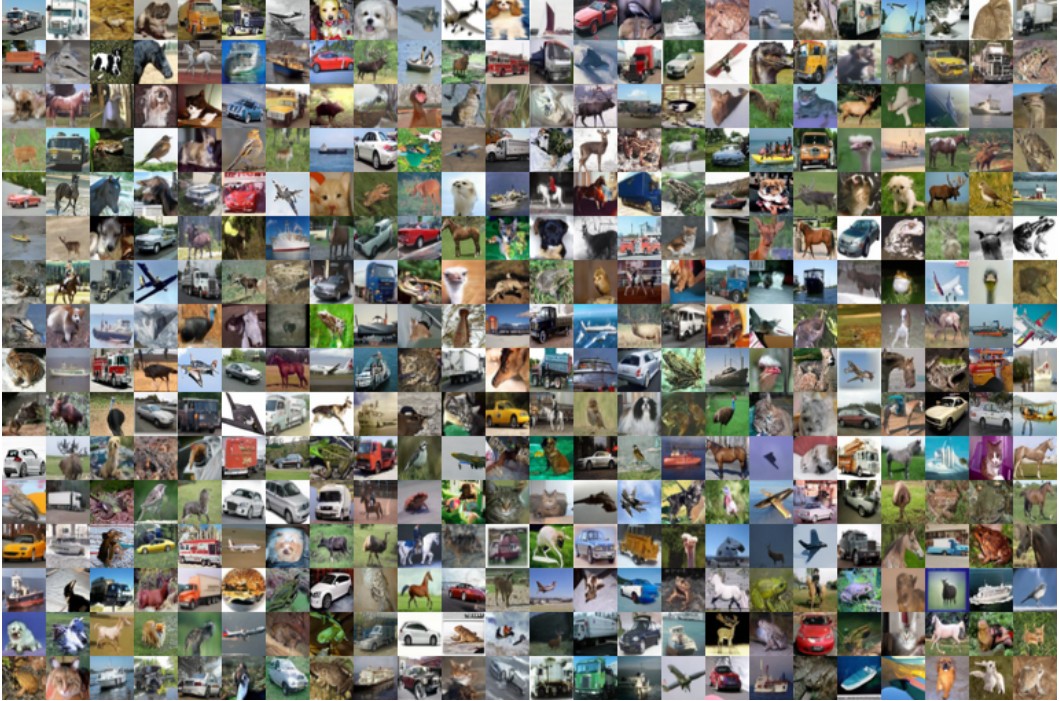

FID = 2.11 (NFE=100),  Sampler: CSPS-D

Figure 6: Random CIFAR-10 samples generated using our deterministic samplers

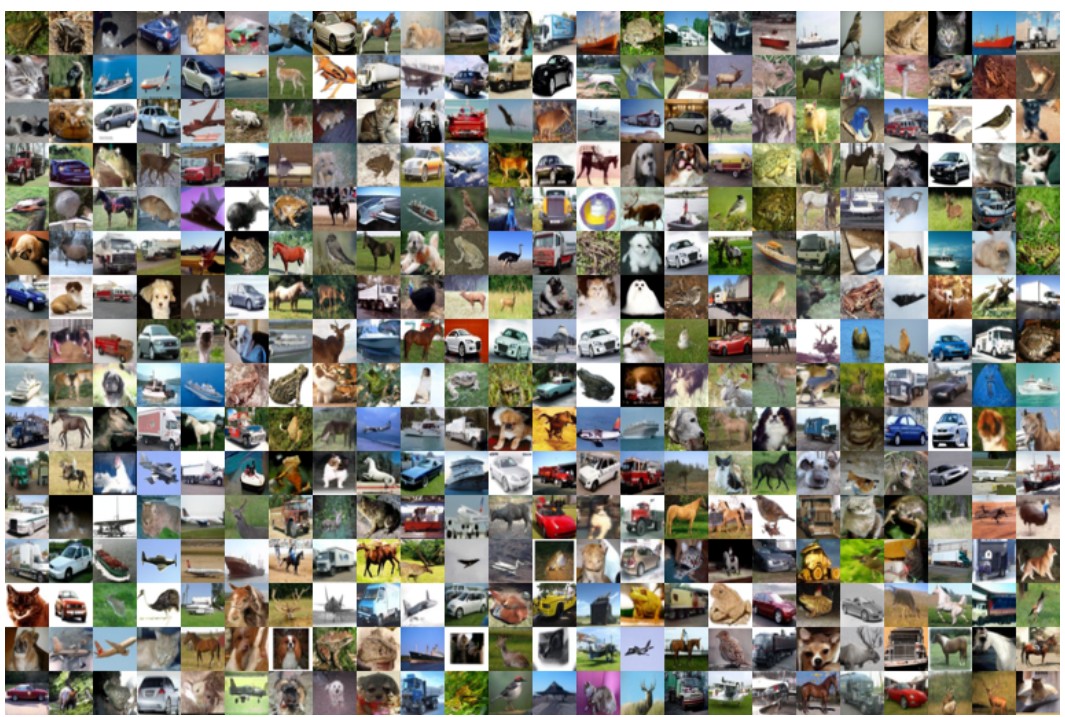

FID = 2.74 (NFE=50),  Sampler: SPS-S (+Pre.)

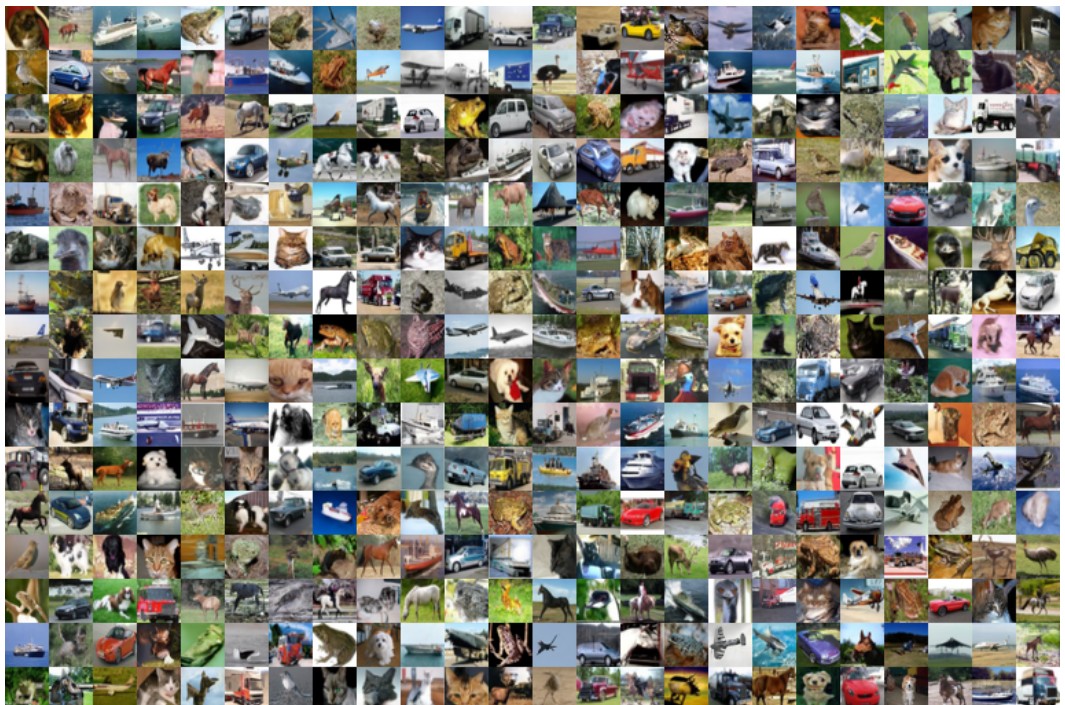

FID = 2.36 (NFE=100),  Sampler: SPS-S

Figure 7: Random CIFAR-10 samples generated using our stochastic samplers

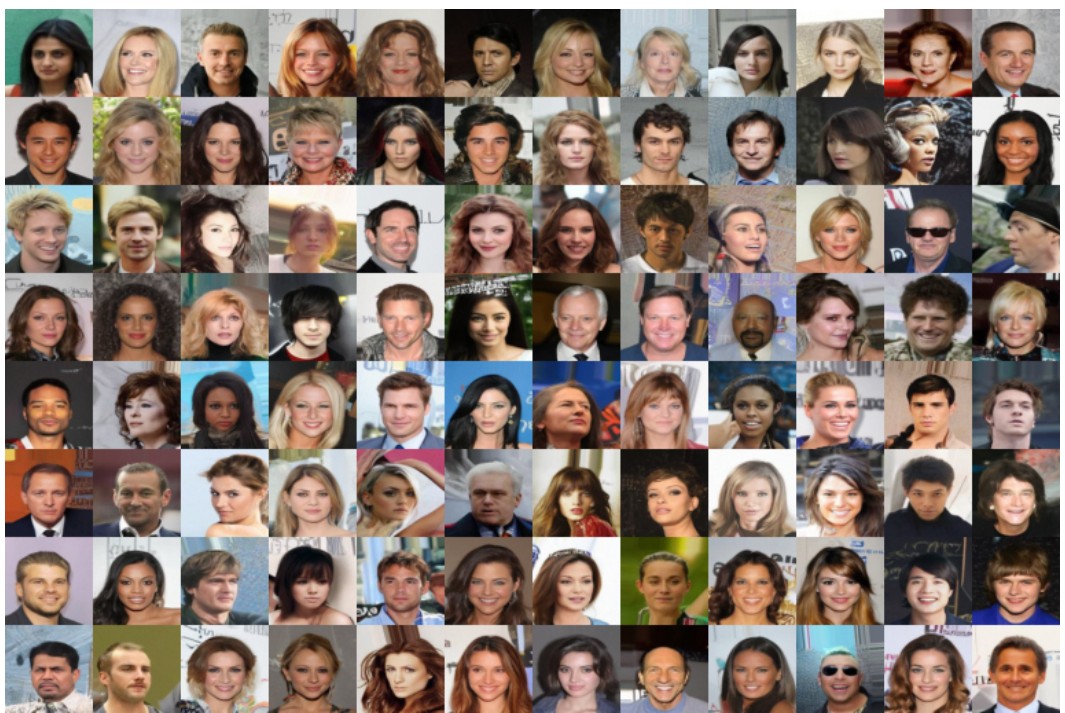

FID = 4.41 (NFE=50),  Sampler: CSPS-D (+Pre.)

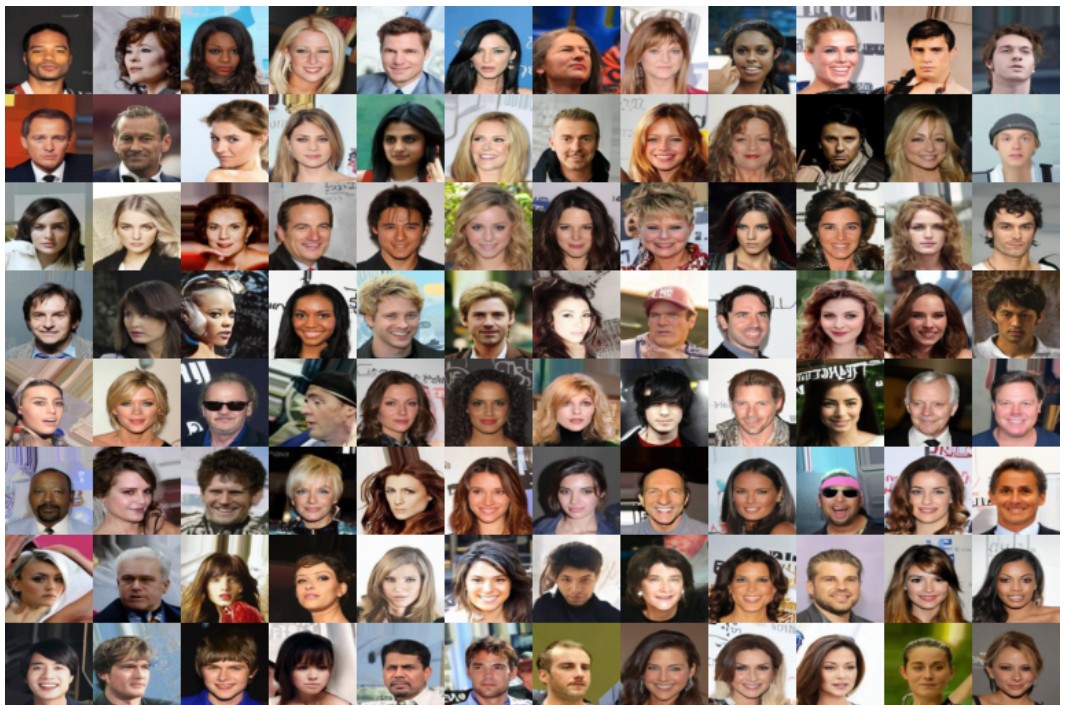

FID = 2.25 (NFE=100),  Sampler: CSPS-D (+Pre.)

Figure 8: Random CelebA-64 samples generated using our deterministic samplers

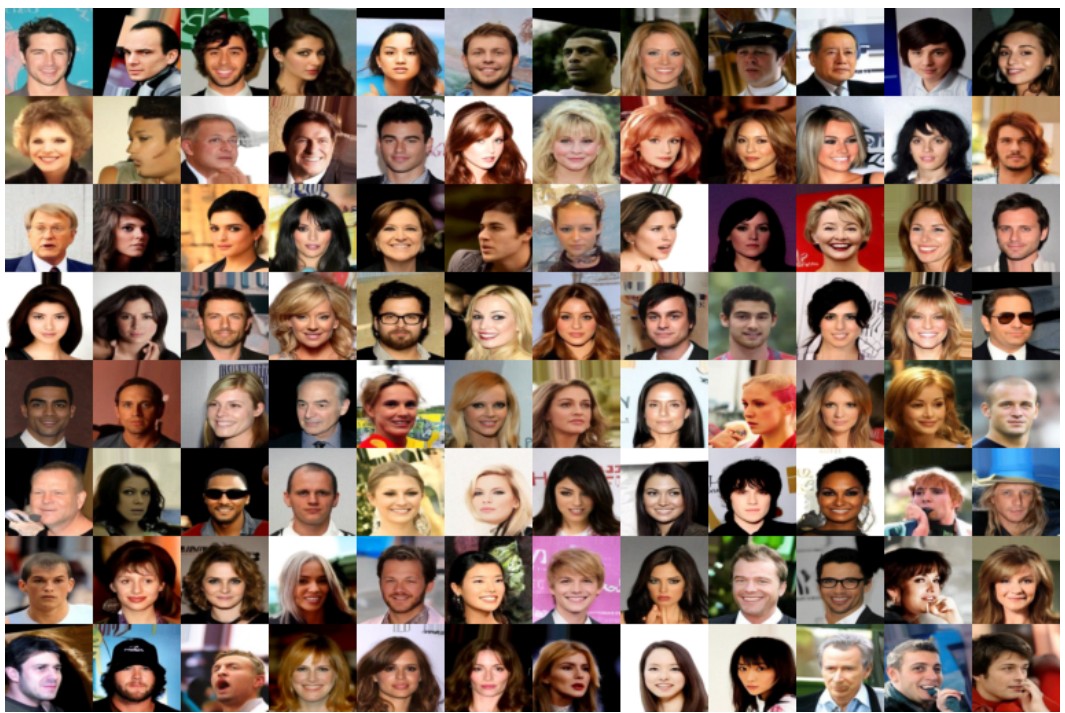

FID = 3.32 (NFE=50), Sampler: CSPS-S (+Pre.)

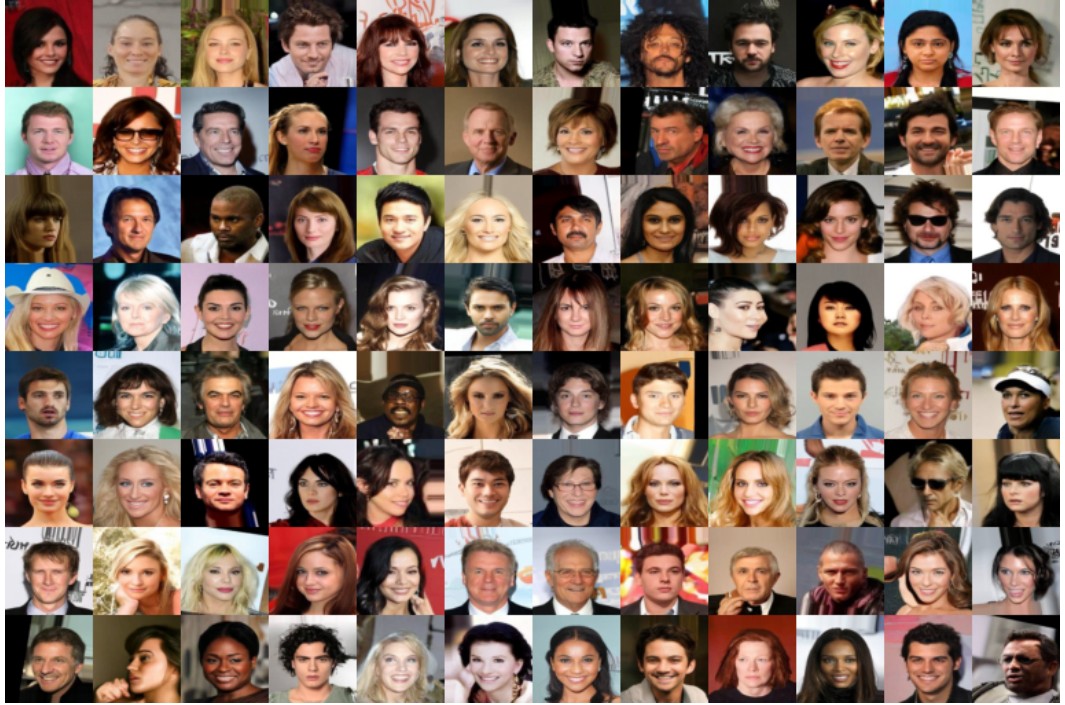

FID = 2.6 (NFE=100), Sampler: CSPS-S (+Pre.)

Figure 9: Random CelebA-64 samples generated using our stochastic samplers

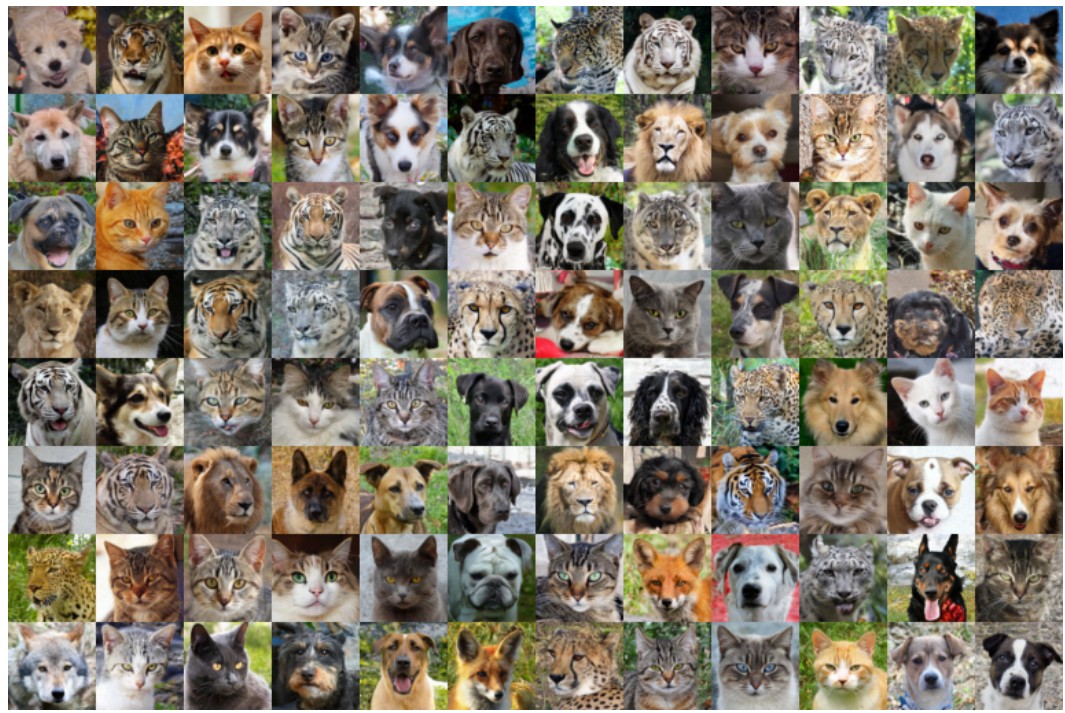

FID = 3.63 (NFE=50),  Sampler: CSPS-D (+Pre.)

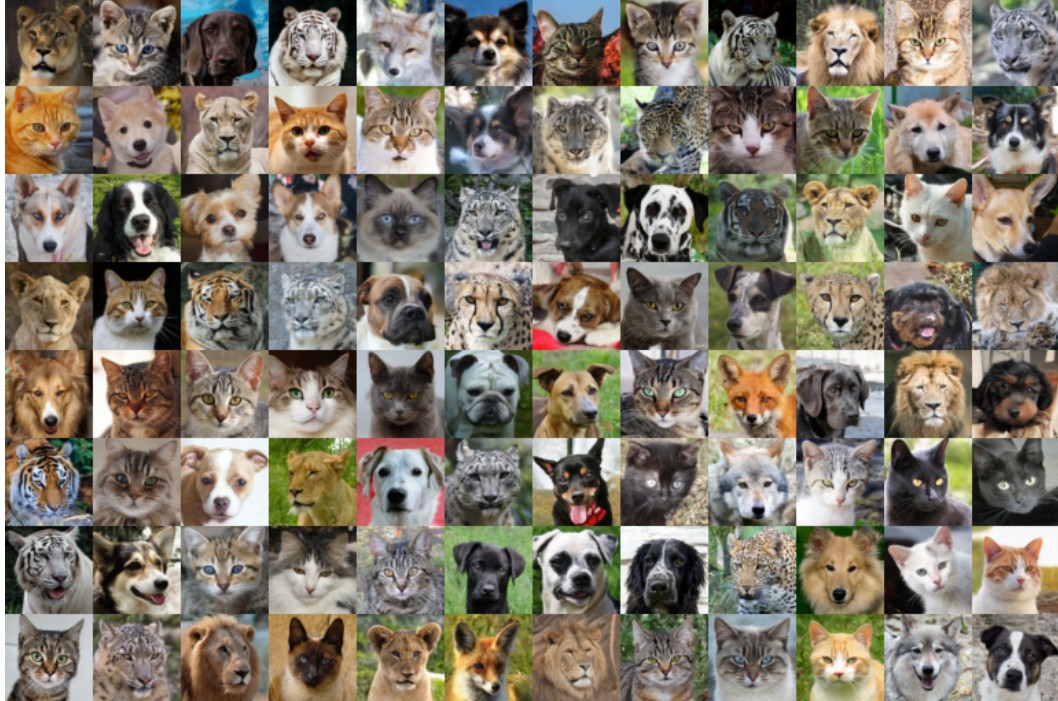

FID = 2.39 (NFE=100),  Sampler: CSPS-D (+Pre.)

Figure 10: Random AFHQv2-64 samples generated using our deterministic samplers

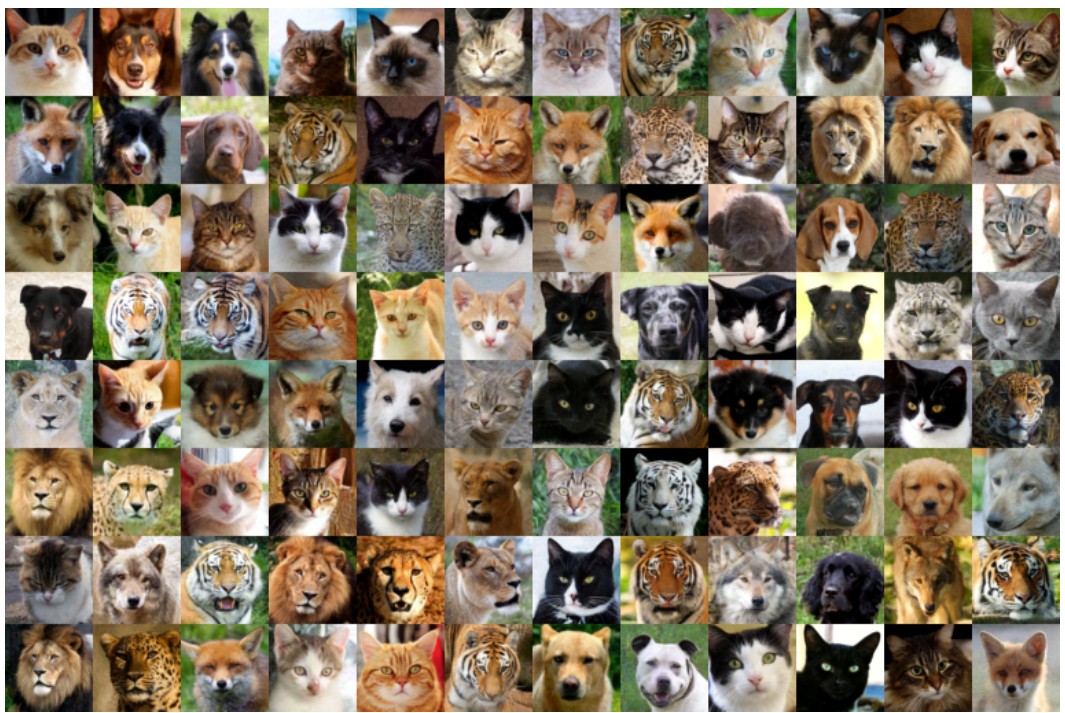

FID = 3.1 (NFE=50), Sampler: CSPS-S (+Pre.)

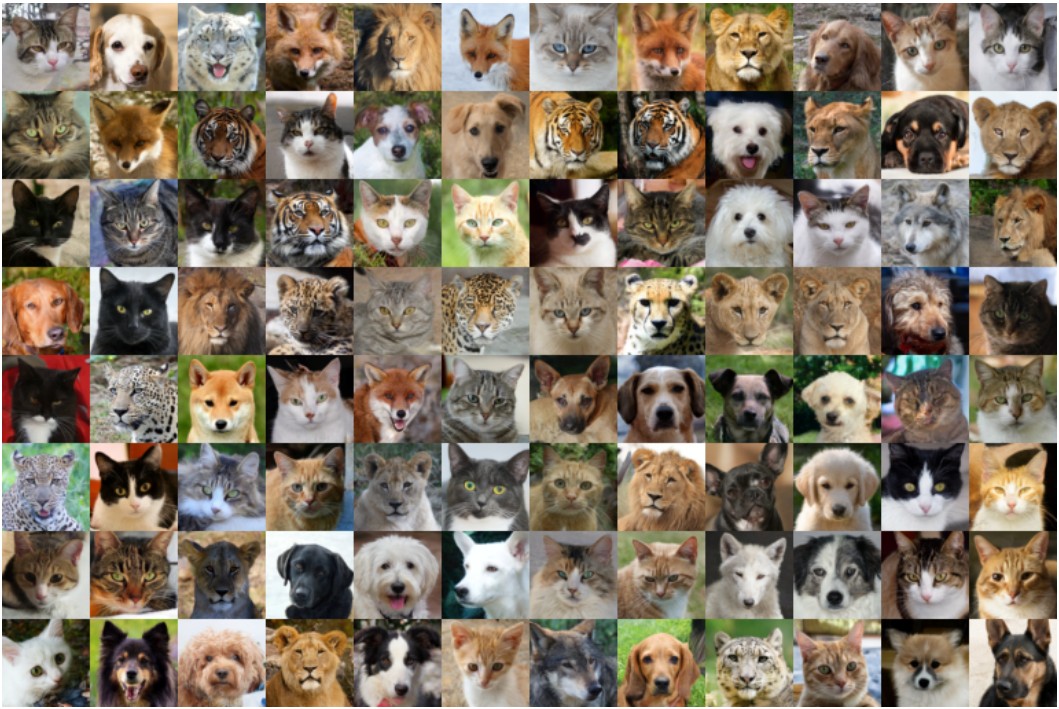

FID = 2.61 (NFE=100), Sampler: CSPS-S (+Pre.)

Figure 11: Random AFHQv2-64 samples generated using our stochastic samplers

