# OpenReview forum: "Efficient Integrators for Diffusion Generative Models"
_ICLR.cc/2024/Conference — ICLR 2024 poster_

### Official Review · Reviewer_8Lyn · 2023-10-25

**Soundness:** 3 good
**Presentation:** 2 fair
**Contribution:** 2 fair
**Rating:** 6
**Confidence:** 4

**Summary:**

This paper...
- proposes conjugate integrators and splitting integrators for accelerating diffusion sampling,
- introduces practical changes to splitting integrators for efficient sampling ("reduced" samplers),
- combines conjugate and splitting integrators to achieve competitive performance in CIFAR-10, CelebA-64, and AFHQ-64 sampling.

**Strengths:**

- Studies conjugate and splitting integrators, which were relatively unexplored in diffusion sampling.
- Proposed samplers generalize previous diffusion integrators, such as DDIM.
- Clearly explains the advantages of conjugate and splitting integrators, and how they contribute in orthogonal ways.

**Weaknesses:**

While the paper has clear strengths, I think the paper needs a major revision. Specifically, I am inclined to give "reject" for the following reasons:

**Weakness 1 : Difficult to figure out the "main" contribution**
- There are too many proposed integrators, and it is difficult to figure out which one is / ones are the "main" contribution of the paper. More specifically, it is difficult to see in which situation we should prefer one integrator over the other. To me, it seems like the authors are taking a very unclear stance on their "main" sampler : the authors are proposing a whole arsenal of samplers, and picking results which happened to beat some baselines results under very specific dataset+sampler+hyper-parameter combinations.
- For instance, what is the rationale behind choosing $B_t = \lambda I$ and $B_t = \lambda 1$? When should we prefer DDIM-I over DDIM-II and vice versa? While there is a theorem on DDIM-I, I don't see any theorem on DDIM-II, which could clarify differences between DDIM-I and DDIM-II.
- Also, readers could expect conjugate+splitting integrators to out-perform conjugate/splitting integrators, as they combine the best of both worlds. However, we don't see this trend in Table 2, where we see Reduced OBA out-performing Conjguate OBA. The authors hypothesize "this might be due to a sub-optimal choice of $B_t$" -- I think this kind of explanation only confuses the readers, as it does not provide any guide on when we should and should not use conjugate splitting.
- I think one factor that makes this paper confusing is the lack of theoretical results -- the authors rely mostly on numerical results to judge the performance of integrators. A theorem comparing, e.g., truncation error of proposed samplers would greatly improve the strength of the paper.

**Weakness 2 : Incomplete / questionable baseline results**
- The authors claim if $B_t = 0$, conjugate integrator is equivalent to DDIM. But, if we see Figure 2 (a), FID for DDIM ($\lambda$-DDIM with $B_t = 0$) is too poor, compared to results in the original DDIM paper. For instance, in the original DDIM paper, DDIM achives FID 13.36 with NFE=10 while in Figure 2 (a), we see FID > 50.
- Result for EDM + pre-conditioning is missing in Figure 5. Why do the authors add pre-conditioning for their methods, but not for EDM? Is it because EDM + pre-conditioning out-performs the proposed integrators? For instance, EDM + pre-conditioning achieves 1.97 FID with NFE=35, which beats the best result in this paper, 2.11 FID with NFE=100.

**Weakness 3 : Incomplete evaluation**
- Results on higher-dimensional data is missing. I would like to see additional results on $\geq 512$ resolution images.
- Results on conditional generation is missing. I would like to see additional results on e.g., class-conditional and text-conditional generation.

**Questions:**

- The relative behavior of stochastic (SPS-S, CSPS-S) and deterministic samplers (CSPS-D) is un-intuitive. Common knowledge is that deterministic samplers outperform stochastic samplers in the low-NFE regime, and vice versa in the high-NFE regime. But, we observe a reverse trend in Figure 5 bottom. Can the authors clarify why this happens?

---

> ### Author Response · Authors · 2023-11-20
> **Response to Reviewer 8Lyn (Part 1/n)**
>
> We thank the reviewer for their insight into our work and respond to each question below:
>
> >**Weakness 1 : Difficult to figure out the "main" contribution**
>
>
>
> > There are too many proposed integrators, and it is difficult to figure out which one is / ones are the "main" contribution of the paper. More specifically, it is difficult to see in which situation we should prefer one integrator over the other. To me, it seems like the authors are taking a very unclear stance on their "main" sampler : the authors are proposing a whole arsenal of samplers, and picking results which happened to beat some baselines results under very specific dataset+sampler+hyper-parameter combinations.
>
> Response: We address this question in different parts
>
> >There are too many proposed integrators, and it is difficult to figure out which one is / ones are the "main" contribution of the paper.
>
> As highlighted in Section 4 (Notation), the main samplers that we recommend using are the Conjugate-Splitting based PSLD samplers (CSPS) and Splitting based PSLD samplers (SPS). We understand that the samplers specified in Table 2 might suggest a lot of contributions. However, we would like to emphasize that Table 2 lists our ablation samplers, the discussion of which is necessary to construct our main samplers, which build on top of these samplers discussed in Table 2.
>
> >More specifically, it is difficult to see in which situation we should prefer one integrator over the other
>
> From Figure 5, we make the following observations. For deterministic sampling, our CSPS samplers always perform better than SPS samplers. For stochastic sampling, except CIFAR-10, the CSPS integrators always perform better than the corresponding SPS samplers. Therefore, for most practical applications, the proposed CSPS samplers should be preferred.
>
> > the authors are proposing a whole arsenal of samplers, and picking results which happened to beat some baselines results under very specific dataset+sampler+hyper-parameter combinations.
>
> We would like to point out that we do not perform any type of cherry-picking when comparing different sampler baselines and our proposed samplers. On the contrary, we perform an extensive comparison across multiple sampling budgets for different datasets in Figure 5, which clearly illustrates when our deterministic and stochastic samplers perform comparably or outperform different baselines. Moreover, our choice of datasets is standard in the diffusion model literature [1,2,3,4]. Regarding hyperparameters, we tune the value of \lambda for CSPS samplers during inference since Theorem 2 suggests that a specific value of \lambda is unlikely to work across multiple integration step sizes h (i.e., for different sampling budgets in terms of NFEs) and is thus a core requirement for CSPS samplers. Moreover, since we evaluate other baselines based on the scores provided in their respective papers, it is safe to assume that our comparison baselines have also been optimized for best performance.
>
>
> > For instance, what is the rationale behind choosing Bt=λI and Bt=λ1? When should we prefer DDIM-I over DDIM-II and vice versa? While there is a theorem on DDIM-I, I don't see any theorem on DDIM-II, which could clarify differences between DDIM-I and DDIM-II.
>
> Response:  We thank the reviewer for this question. The main rationale behind choosing $B_t$ as in $\lambda$-DDIM is mostly simplicity. Our current choice of $B_t =\lambda I$ assumes that the value of $\lambda$ is independent of time $t$, which might be overly simplistic since Theorem 2 suggests that at any time $t$, the eigenvalues $\tilde{\lambda}$ must be bounded for a stable conjugate integrator, therefore, implying a time-varying $\lambda$ might be a better choice. However, we make this simplistic choice because a time-independent lambda is more straightforward to tune during inference. We had the same rationale when choosing $B_t = \lambda 1$.
>
>
> >While there is a theorem on DDIM-I, I don't see any theorem on DDIM-II, which could clarify differences between DDIM-I and DDIM-II.
>
>
> We would like to point out that for the choice of $B_t = \lambda I$, the result in theorem 2 simplifies, resulting in Corollary-1. However, similar simplifications are not apparent for other choices of $B_t = \lambda 1$, and Theorem 2 holds in general for other choices of $B_t$. We acknowledge that further theoretical investigation of different choices of $B_t$ could clarify differences between the two formulations and could be an interesting future direction. We already state this point as an interesting direction for future work in the conclusion.

---

> > ### Author Response · Authors · 2023-11-20
> > **Response to Reviewer 8Lyn (Part 2/n)**
> >
> > >Also, readers could expect conjugate+splitting integrators to out-perform conjugate/splitting integrators, as they combine the best of both worlds. However, we don't see this trend in Table 2, where we see Reduced OBA out-performing Conjguate OBA. The authors hypothesize "this might be due to a sub-optimal choice of Bt" -- I think this kind of explanation only confuses the readers, as it does not provide any guide on when we should and should not use conjugate splitting.
> >
> >
> > While we see this trend for CIFAR-10 in Table 2, we would also like to highlight that for the CelebA64 and the AFHQv2 datasets, CSPS-S (a.k.a Conjugate OBA) outperforms SPS-S (a.k.a Reduced OBA) (see Figure 5 for comparisons).
> >
> >
> > >The authors hypothesize "this might be due to a sub-optimal choice of Bt" -- I think this kind of explanation only confuses the readers, as it does not provide any guide on when we should and should not use conjugate splitting.
> >
> >
> > We would like to point out that the performance of CSPS (and related conjugate integrators) depends significantly on the choice of $B_t$. The choice of optimal $B_t$ is unlikely to be the same for all practical datasets, which is the point in the main text. We will elaborate on this fact in more details in our final revision. In practice, we find that our two considered choices perform well across most of our experimental datasets. Therefore, in general, for both deterministic and stochastic samplers, we recommend using Conjugate Splitting based samplers.
> >
> >
> > >I think one factor that makes this paper confusing is the lack of theoretical results -- the authors rely mostly on numerical results to judge the performance of integrators. A theorem comparing, e.g., truncation error of proposed samplers would greatly improve the strength of the paper.
> >
> >
> > We would like to highlight that we provide an extensive truncation error analysis for deterministic naive and reduced splitting integrators in Theorem 3 (with corresponding proofs in Appendix C.2.3). In fact, we arrive at the design choices for the reduced splitting integrators based on truncation error analysis for both deterministic and stochastic samplers.
> >
> >
> > >**Weakness 2 : Incomplete / questionable baseline results**
> >
> >
> >
> > > The authors claim if Bt=0, conjugate integrator is equivalent to DDIM. But, if we see Figure 2 (a), FID for DDIM (λ-DDIM with Bt=0) is too poor, compared to results in the original DDIM paper. For instance, in the original DDIM paper, DDIM achives FID 13.36 with NFE=10 while in Figure 2 (a), we see FID > 50.
> >
> > Response: We think there is a misunderstanding here. The results in Fig. 2(a) are for DDIM (which is equivalent to our conjugate integrator with $B_t=0$) applied to the PSLD ODE, which is an augmented diffusion model (i.e., data space coupled with some auxiliary variables). In contrast, in the original DDIM paper, the sampler is developed for DDPM, which is a non-augmented diffusion model and is different in design than PSLD. This result also demonstrates an important aspect of DDIM (and related solvers like DEIS and DPM-Solver) that solvers developed for non-augmented diffusion models like VP-SDE/DDPM may not generalize directly to other types of diffusion models. Therefore one of our main contributions in this work is to develop a framework for designing solvers for a broader class of diffusion models like PSLD.
> >
> > > Result for EDM + pre-conditioning is missing in Figure 5. Why do the authors add pre-conditioning for their methods, but not for EDM? Is it because EDM + pre-conditioning out-performs the proposed integrators? For instance, EDM + pre-conditioning achieves 1.97 FID with NFE=35, which beats the best result in this paper, 2.11 FID with NFE=100.
> >
> > Response:
> >
> >
> > >For instance, EDM + pre-conditioning achieves 1.97 FID with NFE=35, which beats the best result in this paper, 2.11 FID with NFE=100.
> >
> >
> > We believe this claim is inaccurate. We include Table 2 in the EDM paper (for CIFAR-10) below for reference:
> >
> >
> > |  Training Configuration                | VP (Unconditional)     | VE (Unconditional)     |
> > |----------------------------------------|------------------------|------------------------|
> > | A - Baseline (VP/VE + EDM Sampler)     | 3.01                   | 3.77                   |
> > | B + Adjust hyperparameters             | 2.51                   | 2.94                   |
> > | C + Redistribute Capacity              | 2.31                   | 2.83                   |
> > | D + Our preconditioning                | 2.29                   | 3.10                   |
> > | E + Our loss function                  | 2.05                   | 1.99                   |
> > | F + Non-leaky augmentation             | 1.97                   | 1.98                   |
> > | NFE                                    | 35                     | 35                     |
> >
> > **Table 2**: Evaluation of training improvements in EDM

---

> > > ### Author Response · Authors · 2023-11-20
> > > **Response to Reviewer 8Lyn (Part 3/n)**
> > >
> > > As can be observed, the FID score of 1.97 in NFE=35 for unconditional CIFAR-10 is achieved after using a number of tricks during training, including extensive hyperparameter tuning, preconditioning, customized loss functions, and an extensive data augmentation pipeline. We emphasize that we do not explore any of these choices (except preconditioning) for improved PSLD training for the following reasons:
> > >
> > >
> > >
> > > 1. Firstly, our comparison baselines like DEIS, DPM-Solver, etc., were evaluated on a vanilla VP-SDE model, and hence tuning such choices for our method would likely result in unfair comparisons.
> > > 2. Secondly, tuning for all these design choices is computationally expensive.
> > >
> > >     Therefore, the claim that EDM + preconditioning achieves a FID score of 1.97 which beats the best result in our paper, is inaccurate since the comparison is not fair in the first place.
> > >
> > >
> > >     Result for EDM + pre-conditioning is missing in Figure 5. Why do the authors add pre-conditioning for their methods, but not for EDM? Is it because EDM + pre-conditioning out-performs the proposed integrators?
> > >
> > >
> > >     To incorporate preconditioning in EDM while maintaining fair comparison with other methods, we trained an unconditional VP-SDE baseline model (Config A in the table above) + EDM’s preconditioning on CIFAR-10 (with 1/C_out as the loss weighting) using their EDM’s official implementation (the hyperparameters during training were set to the baseline parameters in Table 7 in the EDM paper). During sampling, we use the recommended settings in the EDM sampler, i.e., their discretization, choice of scaling and schedule, and Heun’s solver since these are core aspects of their sampler. We compare the sample quality results for different solver budgets below:
> > >
> > >
> > > |  Model                 | 30       | 50       | 70       | 100      |
> > > |------------------------|----------|----------|----------|----------|
> > > | EDM (w/o precond.)     | 3.18     | 3.08     | 3.06     | 3.06     |
> > > | EDM (+precond)         | 5.26     | 5.12     | 5.13     | 5.12     |
> > >
> > > **Table 3**: Comparison between EDM (Config A) with and without EDM specific preconditioning on the CIFAR-10 dataset. All FID scores are reported on 50k samples using the optimal deterministic sampler proposed in EDM.
> > >
> > >
> > > Interestingly, training VP-SDE with the preconditioning proposed in EDM leads to worse performance than without preconditioning. In any case, our proposed samplers outperform both these baselines for CIFAR-10.
> > >
> > > >**Weakness 3 : Incomplete evaluation**
> > >
> > >
> > >
> > > > Results on higher-dimensional data is missing. I would like to see additional results on ≥512 resolution images.
> > >
> > > Response: Since we work with a broader and more recent class of diffusion models like PSLD, the pre-trained models available for these augmented diffusion models are primarily trained on small-scale datasets like CIFAR-10 and CelebA-64. Therefore for additional results on >=512 resolution images, we would need to train a new model from scratch. We would like to emphasize that training such high-resolution augmented models is difficult for two reasons:
> > >
> > > 1. Firstly, training such high-res diffusion models is another active area of research (e.g., DALL-Ev2, Imagen, StableDiffusion) which we don’t consider in this work.
> > > 2. Secondly, training such large-scale models is extremely compute-intensive, which is prohibitive for us. For instance, for training on ImageNet at 64x64 resolution, the authors in EDM train the diffusion model for two weeks on a setup with 32 GPUs which is computationally expensive. Training models with resolutions >512 scales proportionately.
> > >
> > > > Results on conditional generation is missing. I would like to see additional results on e.g., class-conditional and text-conditional generation.
> > >
> > > Response: As mentioned in our previous response, training text conditional models is computationally prohibitive. We acknowledge that investigating the efficacy of proposed solvers for these tasks could be an interesting direction for future work in the context of augmented diffusion models. However, most of these tasks are beyond the scope of this work.

---

> > > > ### Author Response · Authors · 2023-11-20
> > > > **Response to Reviewer 8Lyn (Part 4/n)**
> > > >
> > > > **Questions**:
> > > >
> > > > >The relative behavior of stochastic (SPS-S, CSPS-S) and deterministic samplers (CSPS-D) is un-intuitive. Common knowledge is that deterministic samplers outperform stochastic samplers in the low-NFE regime, and vice versa in the high-NFE regime. But, we observe a reverse trend in Figure 5 bottom. Can the authors clarify why this happens?
> > > >
> > > > We thank the reviewer for this question. In the context of non-augmented diffusion models (like VP-SDE, VE-SDE, etc.), indeed, deterministic samplers outperform stochastic samplers in the low-NFE regime and vice-versa in the high-NFE regime. However, for PSLD (which is an augmented diffusion model), this trend does not seem to hold (at least in the context of samplers presented in this work). We hypothesize that this might be an effect of noise injection from two sources during stochastic sampling. Firstly, we explicitly inject stochasticity in the position space controlled with the parameter \lambda_s (as part of our design choices adopted for reduced splitting integrators). Secondly, noise is also injected into the data space due to coupling with momentum variables during sampling. This might lead the sampler to escape regions of local minima during sampling. Note that such noise injection does not happen for deterministic sampling, which might lead to slower convergence to good samples (especially pronounced in higher resolution datasets like CelebA-64 and AFHQv2.) However, a detailed theoretical investigation of this observation would be an interesting direction for future work.
> > > >
> > > > **References**:
> > > >
> > > > [1] gDDIM: Generalized denoising diffusion implicit models
> > > >
> > > > [2] DPM-Solver: A Fast ODE Solver for Diffusion Probabilistic Model Sampling in Around 10 Steps
> > > >
> > > > [3] Denoising Diffusion Implicit Models
> > > >
> > > > [4] Fast Sampling of Diffusion Models with Exponential Integrator

---

> > > > > ### Comment · Reviewer_8Lyn · 2023-11-21
> > > > >
> > > > > Thank you for the detailed response -- it addresses most of my concerns and misunderstanding regarding this paper. I have raised the score to marginal accept. Please make the promised changes to the main text.

---

> > > > > > ### Author Response · Authors · 2023-11-23
> > > > > > **Author Response**
> > > > > >
> > > > > > We thank the reviewer for adjusting their rating.  We will make sure to include the EDM (+Precond) baseline for all the datasets in our final draft (some of the models on AFHQv2 and CelebaA-64 are still under training, and since the discussion period ends soon, we plan to incorporate all comparisons at once in our final revision.)

---

### Official Review · Reviewer_dkXq · 2023-10-26

**Soundness:** 4 excellent
**Presentation:** 4 excellent
**Contribution:** 4 excellent
**Rating:** 8
**Confidence:** 3

**Summary:**

The paper proposes two frameworks for improving integrators in generative diffusion models, conjugate integrators and splitting integrators, and it proposes a hybrid method combining the two. The integrators are developed from previous use in physics simulation, e.g. molecular dynamics, and they are applicable to augmented diffusion models, for example when including momentum. The authors provide intution behind the integrators and investigate theoretical properties. Finally, they demonstrate experimentally the power of the integrators.

**Strengths:**

- very clear and well-written paper
- very clever application of numerical integration methodology for generative diffusion models
- potentially high impact in improving results for fixed computational budgets

**Weaknesses:**

no apparent weaknesses

**Questions:**

no questions

---

> ### Author Response · Authors · 2023-11-20
> **Response to Reviewer dkXq**
>
> We thank the reviewer for their positive comments. We believe that the frameworks proposed in this paper for fast sampling in diffusion models can be a fruitful starting point for further research in this area.

---

> > ### Comment · Reviewer_dkXq · 2023-11-22
> > **Response**
> >
> > Thank you for the response.

---

### Official Review · Reviewer_eNRC · 2023-10-30

**Soundness:** 2 fair
**Presentation:** 3 good
**Contribution:** 3 good
**Rating:** 6
**Confidence:** 2

**Summary:**

The authors propose two complementary frameworks for accelerating sample generation in pre-trained models: 'Conjugate Integrators' and 'Splitting Integrators'. Conjugate integrators generalize DDIM, mapping the reverse diffusion dynamics to a more amenable space for
sampling. In contrast, splitting-based integrators reduce the numerical simulation error by alternating between numerical updates involving the data and auxiliary variables. The authors test these approaches as well as combinations of these methods on different benchmark datasets.

**Strengths:**

- The theory is interesting and opens up many potential paths for future investigations.
- For some datasets and low-medium number of total integration steps the aforementioned hybrid model shows excellent generative capabilites and outperform many well-known state of the art methods.
- The writing is clear and the paper is well structured.

**Weaknesses:**

- Combinations between datasets and methods are not consistent. For example in Figure 5, EDM is not tested on CelebA-64. Furthermore, these methods must be tested in more complex distributions such as ImageNet.
- The training of each pretrained model and the size of the model is not specified in the main paper. Are these models identical and trained for the same amount of time?
- In Collorary 1, the authors show a connection between stability and the parameter $\lambda$ (which should evolve with time?), however it is not clear how this hyperparrameter is chosen. Furthermore, Corollary 1 does not explain the behaviour of $\lambda$-DDIM-II, which performs best.

**Questions:**

- Do the authors have an explanation why some methods outperform CSPS-D for very small numbers of NFE? (Figure 5, up-left plot).
- Have the authors measured the total integration time? If the generation speed is our main objective, then such results should be provided as well.
- How is $m_0$ defined during the training of 'Conjugate Symplectic Euler' and 'Conjugate Velocity Verlet'? That is, what is the generated $m_{\epsilon}$. In my opinion, the paper would be improved if corresponding training algorithms to Algorithms 2 and 3 would be added. On a related note, does the transformation along the position dimensions $x_t$ remain a diffeomorphism in the augmented setting? If not, this could have negative implications about density estimation, and mode coverage in generation.

---

> ### Author Response · Authors · 2023-11-20
> **Response to Reviewer eNRC (Part 1/n)**
>
> We thank the reviewer for their insight into our work and respond to each question below:
>
>
>
> > Combinations between datasets and methods are not consistent. For example in Figure 5, EDM is not tested on CelebA-64. Furthermore, these methods must be tested in more complex distributions such as ImageNet.
>
> Response:
>
> **Combinations between datasets and methods are not consistent. For example in Figure 5, EDM is not tested on CelebA-64.**
>
>
> Response: In Figure 5 (in the main text), we report FID scores for different baseline samplers based on the results in their respective papers or by using pre-trained checkpoints from their official code implementations. Consequently, for a given dataset, we do not include results for a baseline if the corresponding pre-trained diffusion checkpoint is missing or the paper lacks results for that dataset.
>
>
> **Furthermore, these methods must be tested in more complex distributions such as ImageNet.**
>
>
> Since we demonstrate the proposed samplers in the context of PSLD (a recently proposed state-of-the-art diffusion model), due to the absence of pre-trained models for this type of diffusion model, we would need to re-train PSLD models on large datasets like ImageNet. Training large-scale diffusion models is typically computationally prohibitive, and thus in this work, we only consider small-scale datasets for which training fits within our compute limits or datasets for which pre-trained PSLD checkpoints are available. For instance, for training on ImageNet at 64x64 resolution, the authors in [3], train the diffusion model for two weeks on a setup with 32 GPUs which is computationally expensive.
>
> > The training of each pretrained model and the size of the model is not specified in the main paper. Are these models identical and trained for the same amount of time?
>
> Response: We thank the reviewer for pointing this out. For most competitive baseline results presented in Figure 5 (like DEIS, DPM-Solver, etc.), which use the VP-SDE diffusion baseline, the size of the pre-trained diffusion model is around 108M parameters, while the size of the pre-trained PSLD model used to evaluate our proposed sampler is 97M parameters. Both models are trained for approximately the same number of steps for CIFAR-10 (around 800k-1M steps). Therefore, our comparisons are fair in terms of sample quality and time for each score function evaluation. We have added these details in Appendix E.2.
>
>
> >In Collorary 1, the authors show a connection between stability and the parameter λ (which should evolve with time?), however it is not clear how this hyperparrameter is chosen. Furthermore, Corollary 1 does not explain the behaviour of λ-DDIM-II, which performs best.
>
>
>
>
> Response: We would like to point out that the main motivation behind including Theorem 2 is to give intuition about when projecting to the space $\hat{z}_t$ (which is parameterized by the design choice $B_t$) is a good idea. Furthermore, since we employ simplistic choices of $B_t = \lambda I$, we include Corollary 1 to provide further intuition since the expression for $\bar{\Lambda}$ simplifies. With this motivation in mind, we address the reviewer’s question in two parts:
>
>
> **In Collorary 1, the authors show a connection between stability and the parameter λ (which should evolve with time?), however it is not clear how this hyperparrameter is chosen**
>
>
> In this work, the parameter $\lambda$ is chosen at inference using grid search. Specifically, given a step size h, we tune $\lambda$ using grid search to optimize for best sample quality. In principle, while $\lambda$ can be time-dependent, we use a rather simplistic choice by setting $\lambda$ to be a constant scalar across all timesteps. While this might seem overly simplistic (due to the result in Corollary-1, which suggests that $\lambda$ would be rather time-dependent), it works well empirically and significantly reduces the effort required in tuning $\lambda$ during inference.
>
>
> **Furthermore, Corollary 1 does not explain the behaviour of λ-DDIM-II, which performs best.**
>
>
> As stated above, choosing $B_t = \lambda I$ simplifies the expression for $\bar{\Lambda}$. However, the same is not true for the choice of $B_t = \lambda 1$, and thus it is difficult to argue about the better performance of $\lambda$-DDIM-II over $\lambda$-DDIM-I. Therefore, further investigation into different theoretical aspects of Conjugate Integrators remains an exciting direction for future work. We already make this point more explicit in our future directions section at the end of the main text.

---

> ### Author Response · Authors · 2023-11-20
> **Response to Reviewer eNRC (Part 2/n)**
>
> Questions:
>
>
>
> > Do the authors have an explanation why some methods outperform CSPS-D for very small numbers of NFE? (Figure 5, up-left plot).
>
> Response: We thank the reviewer for this question. For a very small number of NFEs, baselines like DPM-Solver, and DEIS outperform our proposed solvers. To intuitively understand this observation, we highlight that we evaluate our proposed samplers on the PSLD diffusion trained with the epsilon prediction objective. As noted in [1], for augmented space models like CLD, the dynamics in the epsilon space are oscillatory in nature, which prohibits taking larger step sizes with these models. On the other hand, baseline samplers in Figure 5, like DPM-Solver and DEIS, are evaluated on VP-SDE, which does not suffer from this problem and thus allows for larger step size. To quantify this intuition, we generalize DEIS and apply it to the PSLD diffusion ODE. The results in Table 1 below show that DEIS applied to PSLD performs significantly worse in comparison to the proposed samplers in this work across all timesteps.
>
> > Have the authors measured the total integration time? If the generation speed is our main objective, then such results should be provided as well.
>
> Response: We assume the reviewer means the integration time for computing the coefficients required for ODE integration. Given a set of sampling timepoints ${t_i}$, since coefficients are shared between all samples, we only need to perform this process once at the start of sampling. Empirically, for our largest budget of 100 time points, numerical integration for computing coefficients takes around 20s on our setup. We have added these results in Appendix B.6
>
> > How is m0 defined during the training of 'Conjugate Symplectic Euler' and 'Conjugate Velocity Verlet'? That is, what is the generated mϵ. In my opinion, the paper would be improved if corresponding training algorithms to Algorithms 2 and 3 would be added. On a related note, does the transformation along the position dimensions xt remain a diffeomorphism in the augmented setting? If not, this could have negative implications about density estimation, and mode coverage in generation.
>
> Response: We assume that by mentioning the training of “Conjugate Symplectic Euler” and “Conjugate Velocity Verlet”, the reviewer intends the training of the PSLD diffusion model since Conjugate Symplectic Euler and Conjugate Velocity Verlet are sampling algorithms and do not require any training. We hope this clarifies any potential misunderstanding. Next, $m_0$ i.e., the auxiliary variables in PSLD, are sampled from a Gaussian distribution $N(0, MI)$ where $M$ is the mass matrix as defined in PSLD. We encourage the reviewer to check the PSLD paper [2] for more details about $m_0$ and its training algorithm.
>
>
> Regarding comments about the diffeomorphism of the transformation in the position space, we would like to point out that as long as the transformation in the augmented space itself is a diffeomorphism, we can invert the dynamics at any time to reliably recover the augmented space $[x_t, m_t]$ and then discard $m_t$ to recover $x_t$. Therefore, ensuring that the transformation remains a diffeomorphism in the $x_t$ dimension becomes moot since the diffusion happens in the joint space of $x_t$ and $m_t$.
>
> **Additional Results**
>
> | Method                        | 30         | 50       | 70        | 100      |
> |-------------------------------|------------|----------|-----------|----------|
> | DEIS (q=1)                    | 200.67     | 56.9     | 20.41     | 7.33     |
> | DEIS (q=2)                    | 165.75     | 8.61     | 3.39      | 2.95     |
> | CSPS-D (Ours)                 | **7.23**       | **2.65**     | **2.34**      | **2.11**     |
>
> **Table 1**: Comparison between different ODE-solvers for the CIFAR-10 dataset at various sampling budgets. DEIS was generalized using the Conjugate Integrator framework and applied to PSLD reverse diffusion ODE. DEIS (q=k) represents DEIS with polynomial extrapolation (where k represents the polynomial order), which is known to improve performance for low NFEs [5]. CSPS-D (our proposed deterministic sampler in this work) outperforms DEIS by a large margin for all sampling budgets.
>
> References:
>
> [1] gDDIM: Generalized denoising diffusion implicit models
>
> [2] A Complete Recipe for Diffusion Generative Models
>
> [3] Elucidating the Design Space of Diffusion-Based Generative Models

---

> > ### Author Response · Authors · 2023-11-23
> > **End of discussion period**
> >
> > Dear Reviewer, Since the discussion period is coming to an end, please let us know if our response clarifies your concerns. We would be happy to answer any outstanding questions (if any).

---

### Official Review · Reviewer_9LJY · 2023-11-12

**Soundness:** 2 fair
**Presentation:** 2 fair
**Contribution:** 2 fair
**Rating:** 5
**Confidence:** 5

**Summary:**

Very sorry for the late review! I hope the authors will find my advice helpful! As the review is late, I think this is more like a piece of advice for the authors about how to improve this idea and submission. Please feel free to tell me if you have concerns about some of my suggestions!

This paper is based on a novel and fundamental idea, which has not been explored previously. I am supervised by the authors when they present their ideas, as it is a so fundamental and elegant formulation while I have not thought of it previously.

The authors propose to study the integral in a projected space, where the new variable has a stable homeomorphism with the original variable. Under this setting the original integral is equivalent to the new integral, while under careful design, the new integral could be much more easy and fast to compute. Almost all accelerating methods, as I can recall, can fit in this formulation, which means it can generally define the theory insights of all acceleration methods. It could be a good idea to simplify and extend the definition in this paper to a more generalized form. While this paper only considers linear projections, I think it is enough to cover most cases.

The authors then split the new integrals into multiple components, which earns further accelerations. The authors give enough discussions about the error analysis. Extensive studies on small datasets are used to fairly evaluate the proposed methods.

**Strengths:**

1. The formulation of the problem is fundamental and sound. Very promising direction of future research

2. Theory analysis about the stability and error is solid.

3. The method can extend to many familiar sampling techniques, providing a unified and novel insight for all of them.

**Weaknesses:**

1. I don't know whether I deviate too much from the author's experiences. In my experience, in most cases, 25-step DPM-solver sampling could produce good enough sampling results for large models like StableDiffusions on large diverse image distribution; and less than 15-step DPM-solver sampling could produce good enough results for small diffusion models in small data domain. Please tell me if you observe a different phenomenon, and I will be happy to test it. Getting back to the subject, are the 100 and 50 NFEs considered by this paper a bit redundant?

2. In eq 1, the authors actually simplify the original diffusion process to linear ones, and replace the f(z_t) function to linear map F_t z_t, so do please kindly mention that in the corresponding place. Otherwise, it may cause confusion. I care about the scope of this simplification, could it be applied to generation diffusion processes? Especially those in large-scale datasets and models? For example,

3. The stability analysis seems to avoid the key points? I would like to know a two-phase conclusion: first, when will matrix A_t stability have an inverse, this is the key to the stable conjuncture; second, under the condition that A_t is inversible and stable, when will eq 7 and 8 as ODE solvers be stable? Theorem 2 seems like a naive application of ODE stability conditions, and gets rid of the key part. A_t is an integral of multiple components, so we may not directly assume it to be stably invertible. But you also do not assume it stable anyway (minimum eigenvalue larger than some positive constant). It would also be helpful if the authors could explain in what reality settings those conditions can be satisfied.

4. Theorem 3 gives error analysis based on h, I have concerns about its error with respect to the original t. After the projection is to solve the original problem, measuring the error in a projected space could be less meaningful. This is the same problem in the DPM-Solver paper, they only analyze errors when lambda is stably transformed from t, but in reality is often not the case. But DPM-Solver performs great generally so I am not criticizing it. Just point out that error analysis with respect to t variable is preferred for typical ode numerical analysis.

5. This work, also reminds me of the DPM-Solvers. DPM-Solver could be viewed as a special case, it also projects the integral to another space, the lambda space in fact. It uses a prediction-correction (PC) method to achieve much higher accuracy in the projected space. So I care about two things: first how the dpm-solver will formulate under your settings and when will your method outperform it? Second, can your method benefit from PC? While this work does not compare with DPM-Solver, considering the huge influences of and similarity with DPM-Solver, it could be better if the authors could compare with it.

6. Improvements seem not to be good enough, especially considering speed. Large datasets and models are also not considered.


Some minor things and suggestions include:
1. I recommend using the ICLR official math notations, including in the original tex file, to express math concepts. For example, $dt$ should be $\mathrm{d}t$, $\bm{z}_t$ is better than $\rv{z}_t$ in an ODE equation (and I also think is better in SDEs), and set A should be $\mathcal{A}$.
2. Remember to add , or . after equations, they are components of your sentences.

**Questions:**

how the dpm-solver will formulate under your settings and when will your method outperform it?

Sorry for the late review, hope you will find those points useful.

---

> ### Author Response · Authors · 2023-11-20
> **Response to Reviewer 9LJY (Part 1/n)**
>
> We thank the reviewer for their insight into our work and respond to each question below:
>
>
>
> > I don't know whether I deviate too much from the author's experiences. In my experience, in most cases, 25-step DPM-solver sampling could produce good enough sampling results for large models like StableDiffusions on large diverse image distribution; and less than 15-step DPM-solver sampling could produce good enough results for small diffusion models in small data domain. Please tell me if you observe a different phenomenon, and I will be happy to test it. Getting back to the subject, are the 100 and 50 NFEs considered by this paper a bit redundant?
>
> Response: For non-augmented diffusion models (i.e., without auxiliary variables) like Stable Diffusion, indeed, DPM-Solver can produce plausible samples in 15-25 steps. However,
>
> 1. Firstly, Stable Diffusion [1] is position space only, i.e., there are no auxiliary variables in its design. In this work, we consider the sampler design for a broader class of diffusion models, and it's not apparent whether DPM-Solver works effectively in such cases. In this work, we propose Conjugate Integrators, which allow us to bring samplers like DEIS [6], DPM-Solver [7] (see our response to your point 5), and DDIM [8] in a common framework. This further allows us to extend these samplers in a principled way to a more broad class of diffusion models like PSLD. This is illustrated in Table 1 below, where we first extend DEIS (which is an exponential-based integrator like DPM-Solver) to PSLD and show that it does not generalize well to diffusions in augmented spaces like PSLD even for NFE=50 and NFE=100, which justifies our use of 100 and 50 NFEs in this work.
> 2. Secondly, for data-space-only diffusions like VP-SDE, it is commonly observed that state-of-the-art ODE-based samplers like DPM-Solver and DEIS converge quickly in fewer steps but saturate to a sub-optimal FID for even smaller datasets like CIFAR-10 (see the Top-left subfigure in Figure 5) and therefore might not be suitable for applications which might allow for a larger sampling budget. On the contrary, though applied to the PSLD diffusion, our proposed sampler in Figure 5 converges to a better FID score of 2.11 on CIFAR-10 (vs. 2.59 for DPM-Solver). Therefore, when evaluating diffusion models, we think it is more reasonable to evaluate sampler quality on a range of sampling budgets (visualized as sampling speed in NFE vs sample quality in FID).
>
> Overall, we think the sampling budgets of 50 and 100 NFEs are **not** redundant for the results presented in this paper.
>
> > In eq 1, the authors actually simplify the original diffusion process to linear ones, and replace the f(z_t) function to linear map F_t z_t, so do please kindly mention that in the corresponding place. Otherwise, it may cause confusion. I care about the scope of this simplification, could it be applied to generation diffusion processes? Especially those in large-scale datasets and models? For example,
>
> Response: Thanks for pointing this out. We have added a brief note in the background specifying that we primarily consider diffusion processes with affine drift, i.e., $f(z_t) = F_t z_t$. To the best of our knowledge,
>
> 1. The assumption of an affine drift is an essential component for obtaining tractable perturbation kernels $p(x_t|x_0)$ needed for training using denoising score matching.
> 2. Secondly, the assumption works well empirically and is employed in most state-of-the-art diffusion models like EDM [3], VP-SDE [4], and PSLD [5]. Therefore, diffusion models with affine drifts are widely adopted for most state-of-the-art diffusion models (including large-scale diffusion models like Stable Diffusion [1], Imagen [2], etc.)

---

> ### Author Response · Authors · 2023-11-20
> **Response to Reviewer 9LJY (Part 2/n)**
>
> > The stability analysis seems to avoid the key points? I would like to know a two-phase conclusion: first, when will matrix A_t stability have an inverse, this is the key to the stable conjuncture; second, under the condition that A_t is inversible and stable, when will eq 7 and 8 as ODE solvers be stable? Theorem 2 seems like a naive application of ODE stability conditions, and gets rid of the key part. A_t is an integral of multiple components, so we may not directly assume it to be stably invertible. But you also do not assume it stable anyway (minimum eigenvalue larger than some positive constant). It would also be helpful if the authors could explain in what reality settings those conditions can be satisfied.
>
> Response: We thank the reviewer for pointing this issue out. We address the question in two parts.
>
>
> **When will A_t be invertible:**
>
>
> Theoretically, since the expression for $A_t$ in Eqn. 6 consists of multiple terms in the integral with the matrices $B_t$, $F_t$, and $C_\text{skip}(t)$ having no specific properties, it's non-trivial to conclude if arbitrary choices of these matrices can ensure invertibility on $A_t$. An alternate (but less elegant) fix would be to update our mapping instead to ensure an invertible transformation from $z_t$ to $\hat{z}_t$ by setting $\hat{z}_t = (A_t + \delta I) z_t$ where $\delta > 0$ is a small constant to ensure non zero eigenvalues at any time t.
>
>
> However, empirically, when using the PSLD diffusion model (as considered in this work), we do not notice any instabilities because of this issue across our experiments with multiple datasets. We have highlighted this aspect in Section 3.1 and Appendix B.5 in the paper.
>
>
> **When will the ODE solvers derived in Eqn. 8 be stable:**
>
> Theorem 2 states when the ODE integrator in Eqn. 8 is stable, provided the mapping $A_t$ is invertible. We have updated Theorem 2 to reflect the assumption on the invertibility of $A_t$.
>
>
>
> >Theorem 3 gives error analysis based on h, I have concerns about its error with respect to the original t. After the projection is to solve the original problem, measuring the error in a projected space could be less meaningful. This is the same problem in the DPM-Solver paper, they only analyze errors when lambda is stably transformed from t, but in reality is often not the case. But DPM-Solver performs great generally so I am not criticizing it. Just point out that error analysis with respect to t variable is preferred for typical ode numerical analysis.
>
> Response: We think there might be some misunderstanding here.
>
>
> Firstly, we would like to clarify that Theorem 3 presents the error analysis for Naive Splitting Integrators (specifically for the Naive Velocity Verlet (NVV) sampler). As presented in Section 3.2, Splitting Integrators are independent of the Conjugate Integrators presented in Section 3.1 and, thus, do not involve a projection step. Therefore, the truncation error in Theorem 3 is measured in the original rather than the projected space. Furthermore, we don’t combine Conjugate Integrators with Splitting Integrators until Section 3.3. To clarify further, in Section 3.3, we numerically solve individual splitting updates using conjugate integrators.
>
>
> Secondly, the error analysis in Theorem 3 is based on $h$, which is the step size during numerical integration. We would like to point out that error analysis based on the integration step size is commonplace in the numerical analysis of ODEs. We hope this clarifies any misunderstandings about the proposed samplers.

---

> ### Author Response · Authors · 2023-11-20
> **Response to Reviewer 9LJY (Part 3/n)**
>
> > This work, also reminds me of the DPM-Solvers. DPM-Solver could be viewed as a special case, it also projects the integral to another space, the lambda space in fact. It uses a prediction-correction (PC) method to achieve much higher accuracy in the projected space. So I care about two things: first how the dpm-solver will formulate under your settings and when will your method outperform it? Second, can your method benefit from PC? While this work does not compare with DPM-Solver, considering the huge influences of and similarity with DPM-Solver, it could be better if the authors could compare with it.
>
> Response: We address this question in multiple parts.
>
>
> **DPM-Solver as a particular case of Conjugate Integrators**
>
>
> We thank the reviewer for their intuitive insight. During the response period, we established a theoretical connection of Conjugate Integrators with DPM-Solver [1]. More specifically, we find that the DPM Solver sampler for the position space-only diffusion models considered in [1] is a particular case of conjugate integrators with the choice of matrix $B_t = 0$. We have updated Proposition 2 in the main text with proof in Appendix B.3 to reflect the same.
>
>
> **When does our method outperform DPM-Solver?**
>
>
> Our method can outperform DPM-Solver (or, more broadly, other exponential integrator-based methods like DEIS) in the following use cases.
>
> 1. **Exponential-based integrators do not generalize automatically to diffusion models in augmented spaces:** Firstly, with the perspective presented in DPM Solver [1] and other exponential-based integrators like DEIS [5], it is not apparent how to generalize these samplers to a broader class of diffusion models i.e. augmented diffusion models (where diffusion is performed in the data space concatenated with some auxiliary variables like in CLD [2] or PSLD [3]) as considered in this work. Theoretically, one of our main contributions is a framework that generalizes ideas in fast diffusion sampling like DDIM [4], DPM-Solver [1], and DEIS [5] under a single unified framework (see Propositions 1,2 in the main text). The benefit of such a framework is that it allows the construction of analogous samplers for different types of diffusion models like CLD and PSLD (which could be helpful in different contexts) and not just position space-only diffusion models (like Stable Diffusion). Moreover, a principled framework like ours allows us to extend beyond exponential integrator-based samplers. The resulting samplers can generalize better to other types of diffusion models.
>
>     To illustrate our point, empirically, we compare the FID scores of DEIS (generalized to PSLD by setting $B_t=0$) with “\lambda-DEIS” (obtained by setting $B_t=\lambda 1$) applied to the PSLD reverse diffusion in Table 1 (in this response) for the CIFAR-10 dataset. Our extension of DEIS i.e. $\lambda$-DEIS, outperforms the former by a large margin. We expect similar results for DPM-Solver since DEIS and DPM-Solver are exponential integrator-based methods and exhibit similar empirical performance in practice [5]. These results demonstrate that the exponential integrator-based framework does not generalize to novel diffusions automatically, therefore justifying the need for a broader framework as proposed in this work._
>
> 2. **DPM Solver saturates in sample quality for higher budgets.** When comparing with DPM Solver (applied to VP-SDE, which is a non-augmented diffusion model), we find that exponential integrator-based methods converge faster but to a sub-optimal FID. In contrast, our proposed sampler (though applied to PSLD) eventually converges to better overall FID scores for slightly larger budgets (FID=2.11 for our sampler vs 2.59 for DPM-Solver). Please refer to the top-left subfigure in Figure 5 in the main text, where we compare with DPM-Solver and other sampler baselines. Therefore, our method is better suited for applications with a slightly larger computing budget at inference.
>
> **Can the proposed samplers be combined with PC?**
>
>
> Yes, our samplers can be extended to incorporate PC-based sampling as well since special cases like DPM-Solver and DEIS can automatically incorporate correction steps.

---

> ### Author Response · Authors · 2023-11-20
> **Response to Reviewer 9LJY (Part 4/n)**
>
> >Improvements seem not to be good enough, especially considering speed. Large datasets and models are also not considered.
>
> Response: We address this comment in two parts:
>
> **Improvements seem not to be good enough, especially considering speed**
>
> We respectfully disagree and would like to re-highlight some important empirical contributions:
>
> In terms of absolute comparisons with state-of-the-art samplers:
>
>
>
> 1. For deterministic samplers proposed in this work, for sampling budgets between 50 and 100 steps, we outperform existing samplers like DEIS and DPM-Solver across all datasets considered in this work. We find that our proposed sampler for PSLD achieves a FID score of 2.11 in 100 steps (compared to 2.57 for the best-performing baseline DEIS) for CIFAR-10 (See Table 1 and Fig. 5 in the main text). We observe similar results for other datasets like CelebA and AFHQv2.
> 2. For Stochastic Samplers proposed in this work, we again outperform state-of-the-art solvers like EDM [6], where our best stochastic sampler achieves a FID score of 2.36 in 100 function evaluations (as compared to 2.63 for the baseline).
>
> In terms of generalizing existing samplers like DPM-Solver, DEIS, etc., and applying them to the PSLD diffusion, our proposed sampler (CSPS-D)  outperforms these methods by a large margin, as illustrated in Table 1 (below in the response).
>
> **Large datasets and models are also not considered**
>
> Since we demonstrate the proposed samplers in the context of PSLD (a recently proposed state-of-the-art diffusion model), due to the absence of pre-trained models for this type of diffusion model, we would need to re-train PSLD models on large datasets like ImageNet. Training large-scale diffusion models is typically computationally prohibitive, and thus in this work, we only consider small-scale datasets for which training fits within our compute budget or datasets for which pre-trained PSLD checkpoints are available. For instance, for training on ImageNet at 64x64 resolution, the authors in [3] train the diffusion model for two weeks on a setup with 32 GPUs which is computationally expensive. Moreover, we believe that the advantages of our sampling framework are already well-demonstrated on smaller-scale datasets considered in this work.
>
> Minor Comments: We thank the reviewer for pointing out these issues, which we will fix in our final revision.
>
> **Questions:**
>
> >how the dpm-solver will formulate under your settings and when will your method outperform it?
>
> We already provide a response to this question in point 5 of our response above.
>
> **Additional Results**
>
> | Method                        | 30         | 50       | 70        | 100      |
> |-------------------------------|------------|----------|-----------|----------|
> | DEIS (q=1)                    | 200.67     | 56.9     | 20.41     | 7.33     |
> | DEIS (q=2)                    | 165.75     | 8.61     | 3.39      | 2.95     |
> | $\lambda$-DEIS (q=2) (Ours)     | 38.85      | 3.88     | 3.13      | 2.82     |
> | CSPS-D (Ours)                 | **7.23**       | **2.65**     | **2.34**      | **2.11**     |
>
> **Table 1**: Comparison between different ODE-solvers for the CIFAR-10 dataset at various sampling budgets. DEIS was generalized using the Conjugate Integrator framework and applied to PSLD reverse diffusion ODE. DEIS (q=k) represents DEIS with polynomial extrapolation (where k represents the polynomial order), which is known to improve performance for low NFEs [5]. CSPS-D (our proposed deterministic sampler in this work) outperforms DEIS by a large margin for all sampling budgets.
>
> **References**
>
> [1] High-Resolution Image Synthesis with Latent Diffusion Models, Rombach et al.
>
> [2] Photorealistic Text-to-Image Diffusion Models with Deep Language Understanding
>
> [3] Elucidating the Design Space of Diffusion-Based Generative Models
>
> [4] Score-Based Generative Modeling through Stochastic Differential Equations
>
> [5] A Complete Recipe for Diffusion Generative Models
>
> [6] Fast Sampling of Diffusion Models with Exponential Integrator
>
> [7] DPM-Solver: A Fast ODE Solver for Diffusion Probabilistic Model Sampling in Around 10 Steps
>
> [8] Denoising Diffusion Implicit Models

---

> ### Comment · Reviewer_9LJY · 2023-11-21
> **Regarding the Stability Issue**
>
> Thank you for the response.
>
> I have an immediate question about the stability issue of eq 70.
>
> Notice that you have ABA^-1 in the integral. The ode will be very unstable (huge numerical error) if A has a small eigenvalues, adding a small delta will not help it, as it still has an error larger than O(1/sigma).
>
> By the way, noting eq 7, if A^-1 or ABA^-1 is infinite, then this sde does not have a solution as it as infinite Lipschitz on z.

---

> ### Author Response · Authors · 2023-11-21
> **Response - Regarding the Stability Issue**
>
> We thank the reviewer for the follow-up questions. There are two aspects to this issue: Theoretical and empirical.
>
> Theoretically, for the transformation $A_t + \delta I$, the minimum eigenvalue will be $1/(\lambda_\text{min} + \delta)$. Thus, with a small enough delta, we can ensure that the overall transformation matrix $A_t + \delta I$ is invertible (on a machine with finite numerical precision). Moreover, since our framework does not require additional training during inference, $\delta$ could even be a hyperparameter tuned during inference (whether this is even required is an empirical question, which we address below) to obtain a stable inverse.
>
> Given that our transformation is invertible (using a $\delta >0$ parameter to stabilize eigenvalues), regarding whether the ODE in Eqn. 7 has a solution, or whether the integrator in Eqn. 8 has significant discretization errors due to small eigenvalues of $A_t^{-1}$, we would like to highlight that both of these cases depend not only on the invertibility of $A_t$ but also on the design parameter $B_t$ which is essential in the design of conjugate integrators. For instance, for $B_t = \lambda I$ (equivalent to $\lambda$-DDIM-I in this work), the expression $A_tB_tA_t^{-1}$ simplifies to $\lambda I$ thus bypassing all problems stated by the reviewer. Thus, the overall stability of both the ODE in Eqn. 7 and our conjugate integrator in Eqn. 8 depend on $B_t$ and not just on the invertibility of $A_t$. A more formal analysis of the stability of the ODE in Eqn. 7 would likely involve analysis of the eigenvalues of the jacobian of the vector field in Eqn. 7 at an equilibrium point. We acknowledge that this could be an interesting direction for future work.
>
> Empirically, we find that for the PSLD ODE, NFE=100 (i.e., 100 discretized time points) with $\delta=0$, $B_t=\lambda 1$ (i.e., $\lambda$-DDIM-II), the minimum eigenvalue of $A_t$ across all time points is around 1.0089 (which is near $t=\epsilon=1e-3$). Therefore, in practice, we do not observe such instabilities and the inverse of the transformation can be easily computed.
>
> We will make some of these points more explicit in our final revision. Please feel free to revert back if you have any additional questions.

---

> > ### Comment · Reviewer_9LJY · 2023-11-22
> >
> > I respectfully disagree.
> >
> > Invertible and stably invertible are different.
> >
> > Since you propose a new integral formula to solve the original problem, and you cannot demonstrate it in diverse datasets. I think a theoretical check of the properties of the new integral is important. Among them, the existence of a unique strong solution, and the stability of numerical ito calculation are the most basic and important.
> >
> > For the existence of the unique strong solution, the authors may want to prove A_t B_t A_T^-1 is globally bounded, d\phi_t is bounded, and d\epsilon is Lipschitz on z_t. Here bounded means smaller than a reasonable constant, so a rigid constant bound is preferred in your theorem.
> >
> > The overall integral time is also important mentioned by the other reviewers. You need to compute A_t^-1 in each iteration. While the authors propose to add delta to A_t, the accumulated error through the computation is still considerable. And the authors did not analyze this.
> >
> > Also, let's talk about the A_t+delta issue. If you choose the delta to be large, then the computation will not be accurate. The error analysis in Th3 will just be meaningless. The accumulation error from the delta itself can dominate the overall error. If you choose delta to be tiny, then it is meaningless as (A_t+\dleta I)^-1 will still be very very large. The numerical computation will be very inaccurate. In the th3 the authors just assume A_t and A_t^-1 is both stably reversible, meaning they all have appropriate eigenvalues, not too small nor too large. This needs to be mentioned in the theorem.
> >
> > Another problem is that delta should be time-aware. A_t and A_t^-1 will evolve with the time. The delta should be appropriately chosen to suit every time step t. The authors many need to analyze what delta_t is preferred to choose and the overall error of using them.

---

> ### Author Response · Authors · 2023-11-22
> **Response (Contd.)**
>
> Thank you for your follow-up. From your response, we sense a couple of misconceptions (please feel free to correct us if we misunderstood something in your response).
>
> 1. Firstly, Theorem 3 states the error analysis for Splitting Integrators. We once again highlight that the error analysis in Theorem 3 is strictly applicable to Splitting Integrators and **not** Conjugate Integrators (which involve a projection step) and, therefore, separate from this discussion. (Perhaps you meant Theorem 2?)
> 2. We specifically deal with ODEs in Conjugate Integrators in this work, not SDEs. Therefore, Ito's calculus is not applicable here, so there is no notion of a strong/weak solution. (Though extension to SDEs can be a potential direction for future work, which we also highlight in the conclusion.)
>
> With these points in mind, let's consider the following aspects:
>
> **Existence of a solution**: This is a good point. Below, we examine the boundedness of different terms in the ODE in Eqn. 7 using matrix norms (Frobenius norms in particular denoted as ||.|| in subsequent discussion):
>
> 1. **Boundedness of $A_t B_t A_t^{-1}$**: It is straightforward to see that $||A_t B_t A_t^{-1}|| \leq \kappa_t ||B_t||$ where $\kappa_t = ||A_t|| ||A_t^{-1}||$ is the time-dependent condition number. Thus, for this matrix to be bounded, we need $\kappa_t ||B_t||$ to be bounded. This implies that the upper bound on the norm of $A_t B_t A_t^{-1}$ is not only dependent on $\kappa_t$ but also on $||B_t||$. Thus, this regularity condition provides a guideline for the end-user to choose $B_t$ appropriately. We will add this regularity condition to the theorem.
>
> 2. **Boundedness of $d\Phi_t/dt$**: Similarly, it can be shown that $||d\Phi_t/dt|| \leq 1/2 ||A_t|| ||G_t||^2 ||C_\text{out}||$. Since the norm of the diffusion coefficient $G_t$ is bounded, the boundedness of the norm of $||d\Phi/dt||$ primarily depends on the norms $||A_t||$ and $||C_\text{out}||$. Thus, our second regularity condition is that the norm of $A_t$ and $C_\text{out}$ must be bounded.
>
> 3. **Boundedness of $\epsilon_{\theta}(z_t, t)$**: This is our last regularity condition which usually holds for modern neural network design.
>
> Therefore, for our vector field in Eqn. 7 to be bounded, we have **three** regularity conditions that we highlight above. We thank the reviewer for pointing out these issues and will include the proposed regularity conditions in the extended statement for our theorem. It is worth noting that we have tried to be generic in our treatment of the proposed regularity conditions. The exact magnitude of these bounds will depend on the specific choices of design matrices like $F_t$, $G_t$, $C_\text{skip}$, $C_\text{out}$ and $B_t$.
>
> Regarding overall integration time, we already provide estimates for computing coefficients $\Phi_t$ in Appendix B.6 (see response to Reviewer eNRC). We will include the total wall clock running time for different NFE budgets in our subsequent revision. Additionally, we would like to highlight that empirically, in our experiments, we always set $\delta=0$.
>
> Regarding the $(A_t + \delta)$ issue, as already shown above, the norm of $||A_tB_tA_t^{-1}||$ depends on $\kappa_t ||B_t||$. Thus, a time-dependent $\delta$ might not be required since we control the norm $||B_t||$ during sampling anyway by tuning $B_t$ as illustrated in this paper. This automatically helps us control the bound on $||A_tB_tA_t^{-1}||$ and allows us to use a small $\delta$ independent of time.
>
> **Regarding stable invertibility of A_t**: As pointed out by the reviewer, we will mention this condition more explicitly in the revised version of our theorem.
>
> We hope our response provides more theoretical intuition and clarifies your concerns.

---

### Author Response · Authors · 2023-11-20
**Response to Common Comments and Concerns**

We thank the reviewers for their insightful comments, which helped us improve the paper. We have uploaded a revision with relevant changes highlighted in blue. Below we address some common questions from reviewers. For other specific questions, we respond individually to all reviewers.

**Evaluation on larger datasets**

This was a common point from Reviewers 8Lyn, eNRC, and 9LJY. While we acknowledge that our work could potentially benefit from evaluation on larger datasets like ImageNet (Reviewer eNRC), we would like to point out that we demonstrate the proposed samplers in the context of PSLD (a recently proposed state-of-the-art diffusion model). Due to the absence of pre-trained models for this type of diffusion model on large-scale datasets like ImageNet, we would need to re-train PSLD models from scratch. Training large-scale diffusion models is typically computationally prohibitive, and thus in this work, we only consider small-scale datasets for which training fits within our compute budget or datasets for which pre-trained PSLD checkpoints are available. For instance, for training on ImageNet at 64x64 resolution (a relatively small resolution), the authors in [4] train the diffusion model for two weeks on a setup with 32 GPUs which is computationally expensive. So experiments on datasets with resolution >512 (as asked by Reviewer 8Lyn) would scale accordingly and would not fit within our compute budget. Similarly, experiments with large-scale text-to-image models (Reviewers 8Lyn, 9LJY) like StableDiffusion are beyond the scope of this work.

**Main Methodological contributions:**

Due to some review misunderstandings, we re-highlight our core methodological contributions for the readers’ convenience.



Motivation: While fast samplers have been developed for non-augmented diffusions (i.e., diffusion models with no auxiliary variables. e.g., VP-SDE [1]), it is unclear if these samplers can be reliably used to accelerate sampling in a broader class of diffusion models namely augmented diffusion models (where diffusion is performed in a joint space of data and momentum variables, e.g., CLD [2], PSLD [3]). To tackle this question, in this work, we introduce two **novel** frameworks for fast sampling in diffusion models, namely:



1. Conjugate Integrators (see Section 3.1): These integrators project diffusion dynamics to a more “friendly” space and perform numerical integration in that space while projecting back to the original space after completion. We show that popular and commonly used sampling methods like DDIM [5], DEIS [6], and DPM-Solver [7], which were initially developed in the context of non-augmented diffusion models (like VP-SDE), can be cast in this framework (Propositions 1 and 2). This has two benefits. Firstly, this allows us to bring these samplers in a unified framework which allows us to generalize these samplers to a broader class of diffusion models like PSLD (see Figure 2). Secondly, we can extend and improve these samplers further (see Figure 2).

    Lastly, we also present a result that illustrates under what conditions projecting to such a space might be a good idea (see Theorem 2)

2. Splitting Integrators (see Section 3.2): These integrators split the dynamics into multiple components to speed up sampling. We show that the naive application of these integrators is sub-optimal for diffusion sampling. Consequently, based on numerical error analysis (see Theorem 3), we make several design updates to naive splitting samplers, greatly enhancing sampling speed in PSLD (see Figure 3). We refer to these as “Reduced Splitting Integrators”.

Lastly, we combine these two frameworks by applying the proposed conjugate integrators to solve each component of our reduced splitting integrators which yields our best-performing **deterministic** and **stochastic** samplers (see Section 3.3). Reviewer 9LJY had some misunderstanding about this aspect.

A few key points:

1. Our framework allows a unified view for constructing deterministic and stochastic samplers for diffusion generative models.
2. All our empirical analysis throughout the paper is done on the PSLD diffusion model to show that the proposed samplers readily apply to augmented diffusion models. Reviewer 8LyN had some misunderstanding about this point.
3. Our sampling framework is completely “training-free,” i.e., we only need a pre-trained diffusion model (in this case, PSLD) during inference, and any of the proposed sampling methods in this work can be applied directly without requiring any additional training. Reviewer eNRC had some misunderstanding about this point.

---

> ### Author Response · Authors · 2023-11-20
> **(Contd.) Response to Common Comments and Concerns**
>
> References
>
> [1] Score-Based Generative Modeling through Stochastic Differential Equations
>
> [2] Score-Based Generative Modeling with Critically-Damped Langevin Diffusion
>
> [3] A Complete Recipe for Diffusion Generative Models
>
> [4] Elucidating the Design Space of Diffusion-Based Generative Models
>
> [5] Denoising Diffusion Implicit Models
>
> [6] Fast Sampling of Diffusion Models with Exponential Integrator
>
> [7] DPM-Solver: A Fast ODE Solver for Diffusion Probabilistic Model Sampling in Around 10 Steps

---

### Meta-Review · Area_Chair_5R9S · 2023-12-04

**Metareview:**

This paper focuses on the numerical simulation of the backward process of pre-trained diffusion generative model. Two numerical approaches, namely Conjugate Integrators and Splitting Integrators, are proposed for accelerated generation speed.

All reviewers and I agree that the ideas are interesting and we appreciate how the authors introduce tools from numerical analysis to help machine learning. Although some concerns were raised about the scalability of the empirical results, the contribution seems enough for me to recommend acceptance.

However, I urge the authors to revise the paper where appropriate, based on the discussion with the reviewers.

**Justification For Why Not Higher Score:**

Reviewers still have some concern, such as empirical results not being provided for larger scale problems.

**Justification For Why Not Lower Score:**

The ideas are interesting.

---

### Decision · Program_Chairs · 2024-01-16

Accept (poster)